# Near-optimal Offline Reinforcement Learning with Linear Representation: Leveraging Variance Information with Pessimism

**Ming Yin**[*]        **Yaqi Duan**[†]        **Mengdi Wang**[†]        **Yu-Xiang Wang**[*]

[*]Department of Computer Science
University of California, Santa Barbara
{ming_yin,yuxiangw}@cs.ucsb.edu

[†]Department of Electrical and Computer Engineering
Princeton University
{yaqid,mengdiw}@princeton.edu

## Abstract

*Offline reinforcement learning*, which seeks to utilize offline/historical data to optimize sequential decision-making strategies, has gained surging prominence in recent studies. Due to the advantage that appropriate function approximators can help mitigate the sample complexity burden in modern reinforcement learning problems, existing endeavors usually enforce powerful function representation models (*e.g.* neural networks) to learn the optimal policies. However, a precise understanding of the statistical limits with function representations, remains elusive, even when such a representation is linear.

Towards this goal, we study the statistical limits of offline reinforcement learning with linear model representations. To derive the tight offline learning bound, we design the *variance-aware pessimistic value iteration* (VAPVI), which adopts the conditional variance information of the value function for time-inhomogeneous episodic linear *Markov decision processes* (MDPs). VAPVI leverages estimated variances of the value functions to reweight the Bellman residuals in the least-square pessimistic value iteration and provides improved offline learning bounds over the best-known existing results (whereas the Bellman residuals are equally weighted by design). More importantly, our learning bounds are expressed in terms of system quantities, which provide natural instance-dependent characterizations that previous results are short of. We hope our results draw a clearer picture of what offline learning should look like when linear representations are provided.

## 1 Introduction

*Offline reinforcement learning* (offline RL or batch RL Lange et al. (2012); Levine et al. (2020)) is the framework for learning a reward-maximizing policy in an unknown environment (*Markov Decision Process* or MDP)[1] using the logged data coming from some behavior policy $\mu$. Function approximations, on the other hand, are well-known for generalization in the standard supervised learning. Offline RL with function representation/approximation, as a result, provides generalization across large state-action spaces for the challenging sequential decision-making problems when no iteration is allowed (as opposed to online learning). This paradigm is crucial to the success of modern RL problems as many deep RL algorithms find their prototypes in the literature of offline RL. For example, Xie and Jiang (2020) provides a view that *Fitted Q-Iteration* (Gordon, 1999; Ernst et al., 2005) can be considered as the theoretical prototype of the deep $Q$-networks algorithm (DQN) Mnih et al. (2015) with neural networks being the function representors. On the empirical side, there are a huge body of deep RL-based algorithms (Mnih et al., 2015; Silver et al., 2017; Fujimoto et al., 2019; Kumar et al., 2019; Wu et al., 2019; Kidambi et al., 2020; Yu et al., 2020; Kumar et al., 2020; Janner et al., 2021; Chen et al., 2021a; Kostrikov et al., 2022) that utilize function approximations to achieve respective successes in the offline regime. However, it is also realized that practical function approximation schemes can be quite sample inefficient (*e.g.* millions of samples are needed for deep $Q$-network to solve certain Atari games Mnih et al. (2015)).

---

[1]The environment could have other forms as well, *e.g. partially-observed MDP* (POMDP) or *non-markovian decision process* (NMDP).

To understand this phenomenon, there are numerous studies consider how to achieve sample efficiency with function approximation from the theoretical side, as researchers find sample efficient algorithms are possible with particular model representations, in either online RL (*e.g.* Yang and Wang (2019; 2020); Modi et al. (2020); Jin et al. (2020); Ayoub et al. (2020); Jiang et al. (2017); Du et al. (2019); Sun et al. (2019); Zanette et al. (2020); Zhou et al. (2021a); Jin et al. (2021a); Du et al. (2021)) or offline RL (*e.g.* Munos (2003); Chen and Jiang (2019); Xie and Jiang (2020); Jin et al. (2021b); Xie et al. (2021a); Min et al. (2021); Duan et al. (2021); Nguyen-Tang et al. (2021); Zanette et al. (2021)).

Among them, the linear MDP model (Yang and Wang, 2020; Jin et al., 2020), where the transition is represented as a linear combinations of the given $d$-dimensional feature, is (arguably) the most studied setting in function approximation and there are plenty of extensions based upon it (*e.g.* generalized linear model (Wang et al., 2021b), reward-free RL (Wang et al., 2020), gap-dependent analysis (He et al., 2021) or generative adversarial learning (Liu et al., 2021)). Given its prosperity, however, there are still unknowns for understanding function representations in RL, especially in the offline case.

- While there are surging researches in showing provable sample efficiency (polynomial sample complexity is possible) under a variety of function approximation schemes, how to improve the sample efficiency for a given class of function representations remains understudied. For instance, given a neural network approximation class, an algorithm that learns the optimal policy with complexity $O(H^{10})$ is far worse than the one that can learn in $O(H^3)$ sample complexity, despite that both algorithms are considered sample efficient. Therefore, how to achieve the optimal/tight sample complexity when function approximation is provided is a valuable question to consider. On the other hand, it is known that tight sample complexity, due to the limit of the existing statistical analysis tools, can be very tough to establish when function representation has a very complicated form. However, does this mean tight analysis is not hopeful even when the representation is linear?

- Second, in the existing analysis of offline RL (with function approximation or simply the tabular MDPs), the learning bounds depend either explicitly on the data-coverage quantities (*e.g.* uniform concentrability coefficients Chen and Jiang (2019); Xie and Jiang (2020), uniform visitation measure Yin et al. (2021); Yin and Wang (2021a) and single concentrability Rashidinejad et al. (2021); Xie et al. (2021b)) or the horizon length $H$ (Jin et al., 2021b; Uehara and Sun, 2021). While those results are valuable as they do not depend on the structure of the particular problem (therefore, remain valid even for pathological MDPs), in practice, the empirical performances of offline reinforcement learning are often far better than those non-adaptive bounds would indicate. Can the learning bounds reflect the nature of individual MDP instances when the MDP model has a certain function representation?

In this work, we think about offline RL from the above two aspects. In particular, we consider the fundamental linear model representations and ask the following question of interest:

> ***Can we achieve the statistical limits for offline RL when models have linear representations?***

## 1.1 RELATED WORKS

**Offline RL with general function representations.** The finite sample analysis of offline RL with function approximation is initially conducted by Fitted $Q$-Iteration (FQI) type algorithms and can be dated back to (Munos, 2003; Szepesvári and Munos, 2005; Antos et al., 2008a;b). Later, Chen and Jiang (2019); Le et al. (2019); Xie and Jiang (2020) follow this line of research and derive the improved learning results. However, owing to the aim for tackling general function approximation, those learning bounds are expressed in terms of the stringent *concentrability coefficients* (therefore, are less adaptive to individual instances) and are usually only *information-theoretical*, due to the computational intractability of the optimization procedure over the general function classes. Other works impose weaker assumptions (*e.g.* partial coverage (Liu et al., 2020; Kidambi et al., 2020; Uehara and Sun, 2021)), and their finite sample analysis are generally suboptimal in terms of $H$ or the effective horizon $(1 - \gamma)^{-1}$.

**Offline RL with tabular models.** For tabular MDPs, tight learning bounds can be achieved under several data-coverage assumptions. For the class of problems with uniform data-visitation measure $d_m$, the near-optimal sample complexity bound has the rate $O(H^3/d_m\epsilon^2)$ for time-inhomogeneous MDPs (Yin et al., 2021) and $O(H^2/d_m\epsilon)$ for time-homogeneous MDPs (Yin and Wang, 2021a; Ren

et al., 2021). Under the single concentrability assumption, the tight rate $O(H^3 SC^\star/\epsilon^2)$ is obtained by Xie et al. (2021b). In particular, the recent study Yin and Wang (2021b) introduces the *intrinsic offline learning bound* that is not only instance-dependent but also subsumes previous optimal results.

**Offline RL with linear model representations.** Recently, there is more focus on studying the provable efficient offline RL under the linear model representations. Jin et al. (2021b) first shows offline RL with linear MDP is provably efficient by *the pessimistic value iteration*. Their analysis deviates from their lower bound by a factor of $d \cdot H$ (check their Theorem 4.4 and 4.6). Later, Xie et al. (2021a) considers function approximation under the Bellman-consistent assumptions, and, when realized to linear MDP setting, improves the sample complexity guarantee of Jin et al. (2021b) by an order $O(d)$ (Theorem 3.2).[2] However, their improvement only holds for finite action space (due to the dependence $\log |\mathcal{A}|$) and by the direct reduction (from Theorem 3.1) their result does not imply a computationally tractable algorithm with the same guarantee. Concurrently, Zanette et al. (2021) considers the Linear Bellman Complete model and designs the *actor-critic* style algorithm that achieves tight result under the assumption that the value function is bounded by 1. While their algorithm is efficient (which is based on solving a sequence of second-order cone programs), the resulting learning bound requires the action space to be finite due to the mirror descent updates in the *Actor* procedure (Agarwal et al., 2021). Besides, assuming the value function to be less than 1 simplifies the challenges in dealing with horizon $H$ since when rescaling their result to $[0, H]$, there is a $H$ factor blow-up, which makes no horizon improvement comparing to Jin et al. (2021b). As a result, none of the existing algorithms can achieve the statistical limit for the well-structured linear MDP model with the general (infinite or continuous) state-action spaces. On the other hand, Wang et al. (2021a); Zanette (2021) study the statistical hardness of offline RL with linear representations by proving the exponential lower bounds. Recently, Foster et al. (2021) shows realizability and concentrability are not sufficient for offline learning when state space is arbitrary large.

**Variance-aware studies.** Talebi and Maillard (2018) first incorporates the variance structure in online tabular MDPs and Zanette and Brunskill (2019) tightens the result. For linear mixture MDPs, Zhou et al. (2021a) first uses variance structure to achieve near-optimal result and the *Weighted OFUL* incorporates the variance structure explicitly in the regret bound. Recently, Variance-awareness is also considered in Zhang et al. (2021) for horizon-free setting and for OPE problem (Min et al., 2021). In particular, We point out that Min et al. (2021) is the first work that uses variance reweighting for policy evaluation in offline RL, which inspires our study for policy optimization problem. The guarantee of Min et al. (2021) strictly improves over Duan et al. (2020) for OPE problem.

## 1.2 Our contribution

In this work, we study offline RL for time-inhomogeneous episodic linear Markov decision processes. Linear MDPs serve as one critical step towards understanding function approximation in RL since: **1.** unlike general function representation, linear MDP representation has the well-structured form by the given feature representors, which makes delicate statistical analysis hopeful; **2.** unlike tabular representation, which only works for finite models, linear MDP provides generalization as it adapts to infinite or continuous state-action spaces. Especially, we design the *variance-aware pessimistic value iteration* (VAPVI, Algorithm 1) which incorporates the conditional variance information of the value function and, by the variance structure, Theorem 3.2 is able to improve over the aforementioned state-of-the-art guarantees. In addition, we further improve the state-action guarantee by designing an even tighter bonus (4). VAPVI-Improved (Theorem 3.3) is near-minimax optimal as indicated by our lower bound (Theorem 3.5). Importantly, the resulting learning bounds from VAPVI/VAPVI-Improved are able to characterize the adaptive nature of individual instances and yield different convergence rates for different problems. Algorithmically, our algorithm builds upon the nice Min et al. (2021) with pessimism as we use the estimated variances to reweight the Bellman residual learning objective so that the (training) samples with high uncertainty get less attention (Section 3). This is the key to obtaining instance-adaptive guarantees.

## 2 Preliminaries

---

[2] This comparison is based on translating their infinite horizon discounted setting to the finite-horizon case.

## 2.1 PROBLEM SETTINGS

**Episodic time-inhomogeneous linear Markov decision process.** A finite-horizon *Markov Decision Process* (MDP) is denoted as $M = (\mathcal{S}, \mathcal{A}, P, r, H, d_1)$ (Sutton and Barto, 2018), where $\mathcal{S}$ is the arbitrary state space and $\mathcal{A}$ is the arbitrary action space which can be infinite or even continuous. A time-inhomogeneous transition kernel $P_h : \mathcal{S} \times \mathcal{A} \mapsto \Delta^{\mathcal{S}}$ ($\Delta^{\mathcal{S}}$ represents a probability simplex) maps each state action$(s_h, a_h)$ to a probability distribution $P_h(\cdot|s_h, a_h)$ and $P_h$ can be different across time. In addition, $r : \mathcal{S} \times A \mapsto \mathbb{R}$ is the mean reward function satisfying $0 \leq r \leq 1$. $d_1$ is the initial state distribution. $H$ is the horizon. A policy $\pi = (\pi_1, \dots, \pi_H)$ assigns each state $s_h \in \mathcal{S}$ a probability distribution over actions according to the map $s_h \mapsto \pi_h(\cdot|s_h) \; \forall h \in [H]$ and induces a random trajectory $s_1, a_1, r_1, \dots, s_H, a_H, r_H, s_{H+1}$ with $s_1 \sim d_1, a_h \sim \pi(\cdot|s_h), s_{h+1} \sim P_h(\cdot|s_h, a_h), \forall h \in [H]$. In particular, we adopts the linear MDP protocol from Jin et al. (2020; 2021b), meaning that the transition kernel and the mean reward function admit linear structures in the feature map.

**Definition 2.1** (Linear MDPs). [3] *An episodic MDP* $(\mathcal{S}, \mathcal{A}, H, P, r)$ *is called a linear MDP with a known (unsigned) feature map* $\phi : \mathcal{S} \times \mathcal{A} \to \mathbb{R}^d$ *if there exist $d$ unknown (unsigned) measures* $\nu_h = (\nu_h^{(1)}, \dots, \nu_h^{(d)})$ *over $\mathcal{S}$ and an unknown vector $\theta_h \in \mathbb{R}^d$ such that*

$$P_h(s' \mid s, a) = \langle \phi(s, a), \nu_h(s') \rangle, \quad r_h(s, a) = \langle \phi(x, a), \theta_h \rangle, \quad \forall s', s \in \mathcal{S}, \; a \in \mathcal{A}, \; h \in [H].$$

*where* $\|\nu_h(\mathcal{S})\|_2 \leq \sqrt{d}$ *and* $\max(\|\phi(s,a)\|_2, \|\theta_h\|_2) \leq 1$ *for all* $h \in [H]$ *and* $\forall s, a \in \mathcal{S} \times \mathcal{A}$. $\|\mu_h(\mathcal{S})\| = \int_{\mathcal{S}} \|\mu_h(s)\| \, ds$.

**V-values and Q-values.** For any policy $\pi$, the $V$-value functions $V_h^\pi(\cdot) \in \mathbb{R}^S$ and Q-value functions $Q_h^\pi(\cdot, \cdot) \in \mathbb{R}^{S \times A}$ are defined as: $V_h^\pi(s) = \mathbb{E}_\pi[\sum_{t=h}^H r_t | s_h = s]$, $Q_h^\pi(s, a) = \mathbb{E}_\pi[\sum_{t=h}^H r_t | s_h, a_h = s, a]$, $\forall s, a, h \in \mathcal{S}, \mathcal{A}, [H]$. The performance measure is defined as $v^\pi := \mathbb{E}_{d_1}[V_1^\pi] = \mathbb{E}_{\pi, d_1}\left[\sum_{t=1}^H r_t\right]$. The Bellman (optimality) equations follow $\forall h \in [H]$: $Q_h^\pi = r_h + P_h V_{h+1}^\pi$, $V_h^\pi = \mathbb{E}_{a \sim \pi_h}[Q_h^\pi]$, $Q_h^\star = r_h + P_h V_{h+1}^\star$, $V_h^\star = \max_a Q_h^\star(\cdot, a)$ (where $Q_h, V_h, P_h$ are vectors). By Definition 2.1, the $Q$-values also admit linear structures, *i.e.* $Q_h^\pi = \langle \phi, w_h^\pi \rangle$ for some $w_h^\pi \in \mathbb{R}^d$ (Lemma H.9). Lastly, for a policy $\pi$, we denote the induced occupancy measure over the state-action space at any time $h \in [H]$ to be: for any $E \subseteq \mathcal{S} \times \mathcal{A}$, $d_h^\pi(E) := \mathbb{E}[(s_h, a_h) \in E | s_1 \sim d_1, a_i \sim \pi(\cdot|s_i), s_i \sim P_{i-1}(\cdot|s_{i-1}, a_{i-1}), 1 \leq i \leq h]$ and $\mathbb{E}_{\pi, h}[f(s, a)] := \int_{\mathcal{S} \times \mathcal{A}} f(s, a) d_h^\pi(s, a) ds da$. Here for notation simplicity we abuse $d_h^\pi(\cdot)$ to denote either probability measure or density function.

**Offline learning setting.** Offline RL requires the agent to learn the policy $\pi$ that maximizes $v^\pi$, provided with the historical data $\mathcal{D} = \{(s_h^\tau, a_h^\tau, r_h^\tau, s_{h+1}^\tau)\}_{\tau \in [K]}^{h \in [H]}$ rolled out from some behavior policy $\mu$. The offline nature requires we cannot change $\mu$ and in particular we do not know the data generating distribution of $\mu$. To sum up, the agent seeks to find a policy $\pi_{\text{alg}}$ such that $v^\star - v^{\pi_{\text{alg}}} \leq \epsilon$ for the given batch data $\mathcal{D}$ and a given targeted accuracy $\epsilon > 0$.

## 2.2 ASSUMPTIONS

It is known that learning a near-optimal policy from the offline data $\mathcal{D}$ cannot be sample efficient without certain data-coverage assumptions (Wang et al., 2021a; Yin and Wang, 2021b). To begin with, we define the population covariance matrix under the behavior policy $\mu$ for all $h \in [H]$:

$$\Sigma_h^p := \mathbb{E}_{\mu, h}\left[\phi(s, a)\phi(s, a)^\top\right], \tag{1}$$

since $\Sigma_h^p$ measure the coverage of state-action space for data $\mathcal{D}$, we make the following assumption.

**Assumption 2.2** (Feature Coverage). *The data distributions $\mu$ satisfy the minimum eigenvalue condition:* $\forall h \in [H]$, $\kappa_h := \lambda_{\min}(\Sigma_h^p) > 0$ *and denote* $\kappa = \min_h \kappa_h$. *Note $\kappa$ is a system-dependent (non-universal) quantity as it is upper bounded by $1/d$ (Assumption 2 in Wang et al. (2021a)).*

We make this assumption for the following reasons. First of all, our offline learning guarantee (Theorem 3.2) provides simultaneously comparison to all the policies, which is stronger than only competing with the optimal policy (whereas relaxed assumption suffices, *e.g.* $\sup_{x \in \mathbb{R}^d} \frac{x \Sigma_{\pi^\star} x^\top}{x \Sigma_\mu x^\top} < \infty$

---

[3]This definition is a standard extension over the tabular MDPs by referencing the similar notions from the bandit literature, *i.e.* from *Multi-armed Bandit* to *Linear Bandit* (Lattimore and Szepesvári, 2020).

(Uehara and Sun, 2021)). As a consequence, the behavior distribution $\mu$ must be able to explore each feature dimension for the result to be valid. Second, even if Assumption 2.2 does not hold, we can always restrict our algorithmic design to the *effective subspan* of $\Sigma_h^p$, which causes the alternative notion of $\kappa := \min_{h \in [H]} \{\kappa_h : s.t. \ \kappa_h = \text{smallest positive eigenvalue at time } h\}$ (see Appendix G.1 for detailed discussions). In this scenario, learning the optimal policy cannot be guaranteed as a constant suboptimality gap needs to be suffered due to the lack of coverage and this is formed as *assumption-free RL* in Yin and Wang (2021b). Lastly, previous works analyzing the linear MDPs impose very similar assumptions, *e.g.* Xie et al. (2021a) Theorem 3.2 where $\Sigma_{\mathcal{D}}^{-1}$ exists and Min et al. (2021) for the OPE problem.

Next, for any function $V_{h+1}(\cdot) \in [0, H-h]$, we define the conditional variance $\sigma_{V_{h+1}} : \mathcal{S} \times \mathcal{A} \to \mathbb{R}_+$ as $\sigma_{V_{h+1}}(s,a)^2 := \max\{1, \text{Var}_{P_h}(V_{h+1})(s,a)\}$.[4] Based on this definition, we can define the variance-involved population covariance matrices as: $\Lambda_h^p := \mathbb{E}_{\mu,h} \left[ \sigma_{V_{h+1}}(s,a)^{-2} \phi(s,a) \phi(s,a)^\top \right]$. In particular, when $V_h = V_h^\star$, we use the notation $\Lambda_h^{\star p}$ instead.

## 3 ALGORITHM

Least square regression is usually considered as one of the "default" tools for handling problems with linear structures (*e.g.* LinUCB algorithm for linear Bandits) and finds its popularity in RL as well since *Least-Square Value Iteration* (LSVI, Jin et al. (2020)) is shown to be provably efficient for linear MDPs, due to that $V_{h+1}(s')$ is an unbiased estimator of $[P_h V_{h+1}](s,a)$. Concretely, it solves the ridge regression problems at each time steps (with $\lambda > 0$ being the regularization parameter):

$$\widehat{w}_h := \operatorname*{argmin}_{w \in \mathbb{R}^d} \lambda \|w\|_2^2 + \sum_{k=1}^{K} \left[ \langle \phi(s_h^k, a_h^k), w \rangle - r_h^k - V_{h+1}(s_{h+1}'^k) \right]^2 \tag{2}$$

and has the closed-form solution $\widehat{w}_h = \Sigma_h^{-1} \sum_{k=1}^{K} \phi(s_h^k, a_h^k)[r_{k,h} + V_{h+1}(s_h'^k)]$ with $\Sigma_h^{-1} = \sum_{k=1}^{K} \phi(s_h^k, a_h^k)\phi(s_h^k, a_h^k)^\top + \lambda I$. In offline RL, this has also been leveraged in *pessimistic value iteration* (Jin et al., 2021b) and *fitted Q-evaluation* (Duan et al., 2020). Nevertheless, LSVI could only yield suboptimal guarantees, as illustrated by the following example.

**Example 3.1.** *Instantiate PEVI (Theorem 4.4 in Jin et al. (2021b)) with $\phi(s,a) = \mathbf{1}_{s,a}$ (i.e. tabular MDPs)[5], by direct calculation the learning bound has the form $O(dH \cdot \sum_{h,s,a} d_h^{\pi^\star}(s,a) \sqrt{\frac{1}{K \cdot d_h^\mu(s,a)}})$ and the optimal result (Yin and Wang (2021b) Theorem 4.1) gives $O(\sum_{h,s,a} d_h^{\pi^\star}(s,a) \sqrt{\frac{\text{Var}_{P_{s,a}}(r+V_{h+1}^\star)}{K \cdot d_h^\mu(s,a)}})$. The former has the horizon dependence $H^2$ and the latter is $H^{3/2}$ by law of total variance.*

**Motivation.** By comparing the above two expressions, it can be seen that PEVI cannot get rid of the explicit $H$ factor due to missing the variance information (*w.r.t* $V^\star$). If we go deeper, one could find that it might not be all that ideal to put equal weights on all the training samples in the least square objective (2), since, unlike linear regression where the randomness coming from one source distribution, we are regressing over a sequence of distributions in RL (*i.e.* each $s_h, a_h$ corresponds to a different distribution $P(\cdot|s_h, a_h)$ and there are possibly infinite many of them). Therefore, conceptually, the sample piece $(s_h, a_h, s_{h+1})$ that has higher variance distribution $P(\cdot|s_h, a_h)$ tends to be less "reliable" than the one $(s_h', a_h', s_{h+1}')$ with lower variance (hence should not have equal weight in (2)). This suggests reweighting scheme might help improve the learning guarantee and reweighting over the variance of the value function stands as a natural choice.

### 3.1 VARIANCE-AWARE PESSIMISTIC VALUE ITERATION

Now we explain our framework that incorporates the variance information. Our design is motivated by previous Zhou et al. (2021a) (for online learning) and Min et al. (2021) (for policy evaluation). By the offline nature, we can use the independent episodic data $\mathcal{D}' = \{(\bar{s}_h^\tau, \bar{a}_h^\tau, \bar{r}_h^\tau, \bar{s}_h^{\tau\prime})\}_{\tau \in [K]}^{h \in [H]}$ (from $\mu$) to estimate the conditional variance of any $V$-values $V_{h+1}$ via the definition $[\text{Var}_h V_{h+1}](s,a) =$

---

[4]The $\max(1, \cdot)$ applied here is for technical reason only. In general, it suffices to think $\sigma_{V_{h+1}}^2 \approx \text{Var}_h V_{h+1}$.
[5]This provides a valid illustration since tabular MDP is a special case of linear MDPs.

$[P_h(V_{h+1})^2](s,a) - ([P_hV_{h+1}](s,a))^2$. For the second order moment, by Definition 2.1, it holds

$$\left[P_hV_{h+1}^2\right](s,a) = \int_{\mathcal{S}} V_{h+1}^2\left(s'\right)\ \mathrm{d}P_h\left(s'\mid s,a\right) = \phi(s,a)^\top \int_{\mathcal{S}} V_{h+1}^2\left(s'\right)\ \mathrm{d}\nu_h\left(s'\right).$$

Denote $\beta_h := \int_{\mathcal{S}} V_{h+1}^2\left(s'\right)\ \mathrm{d}\nu_h\left(s'\right)$, then $P_hV_{h+1}^2 = \langle\phi,\beta_h\rangle$ and we can estimator it via:

$$\bar{\beta}_h = \operatorname*{argmin}_{\beta\in\mathbb{R}^d}\sum_{k=1}^K\left[\left\langle\phi(\bar{s}_h^k,\bar{a}_h^k),\beta\right\rangle - V_{h+1}^2\left(\bar{s}_{h+1}^k\right)\right]^2 + \lambda\|\beta\|_2^2 = \bar{\Sigma}_h^{-1}\sum_{k=1}^K\phi(\bar{s}_h^k,\bar{a}_h^k)V_{h+1}^2\left(\bar{s}_{h+1}^k\right)$$

and, similarly, the first order moment $P_hV_{h+1} := \langle\phi,\theta_h\rangle$ can be estimated via:

$$\bar{\theta}_h = \operatorname*{argmin}_{\theta\in\mathbb{R}^d}\sum_{k=1}^K\left[\left\langle\phi(\bar{s}_h^k,\bar{a}_h^k),\theta\right\rangle - V_{h+1}\left(\bar{s}_{h+1}^k\right)\right]^2 + \lambda\|\theta\|_2^2 = \bar{\Sigma}_h^{-1}\sum_{k=1}^K\phi(\bar{s}_h^k,\bar{a}_h^k)V_{h+1}\left(\bar{s}_{h+1}^k\right)$$

The final estimator is defined as $\widehat{\sigma}_{V_h}^2(\cdot,\cdot) := \max\{1,\widehat{\mathrm{Var}}_hV_{h+1}(\cdot,\cdot)\}$ with $\widehat{\mathrm{Var}}_hV_{h+1}(\cdot,\cdot) = \langle\phi(\cdot,\cdot),\bar{\beta}_h\rangle_{[0,(H-h+1)^2]} - \left[\langle\phi(\cdot,\cdot),\bar{\theta}_h\rangle_{[0,H-h+1]}\right]^2$.[6] In particular, when setting $V_{h+1} = \widehat{V}_{h+1}$, it recovers $\widehat{\sigma}_h$ in Algorithm 1 line 8. Here $\bar{\Sigma}_h = \sum_{\tau=1}^K\phi(\bar{s}_h^\tau,\bar{a}_h^\tau)\phi(\bar{s}_h^\tau,\bar{a}_h^\tau)^\top + \lambda I_d$.

**Variance-weighted LSVI.** The idea of LSVI (2) is based on approximate the Bellman updates: $\mathcal{T}_h(V)(s,a) = r_h(s,a) + (P_hV)(s,a)$. With variance estimator $\widehat{\sigma}_h$ at hand, we can modify (2) to solve the variance-weighted LSVI instead (Line 10 of Algorithm 1)

$$\widehat{w}_h := \operatorname*{argmin}_{w\in\mathbb{R}^d}\lambda\|w\|_2^2 + \sum_{k=1}^K\frac{\left[\langle\phi(s_h^k,a_h^k),w\rangle - r_h^k - \widehat{V}_{h+1}(s_{h+1}'^k)\right]^2}{\widehat{\sigma}_h^2(s_h^k,a_h^k)} = \widehat{\Lambda}_h^{-1}\sum_{k=1}^K\frac{\phi\left(s_h^k,a_h^k\right)\cdot\left[r_h^k + \widehat{V}_{h+1}\left(s_{h+1}^k\right)\right]}{\widehat{\sigma}^2(s_h^k,a_h^k)}$$

where $\widehat{\Lambda}_h = \sum_{k=1}^K\phi(s_h^k,a_h^k)\phi(s_h^k,a_h^k)^\top/\widehat{\sigma}_h^2(s_h^k,a_h^k) + \lambda I_d$. The estimated Bellman update $\widehat{\mathcal{T}}_h$ (acts on $\widehat{V}_{h+1}$) is defined as: $(\widehat{\mathcal{T}}_h\widehat{V}_{h+1})(\cdot,\cdot) = \phi(\cdot,\cdot)^\top\widehat{w}_h$ and the pessimism $\Gamma_h$ is assigned to update $\widehat{Q}_h \approx \widehat{\mathcal{T}}_h\widehat{V}_{h+1} - \Gamma_h$, *i.e.* Bellman update + Pessimism (Line 10-12 in Algorithm 1).

**Tighter Pessimistic Design.** To improve the learning guarantee, we create a tighter penalty design that includes $\widehat{\Lambda}_h^{-1}$ rather than $\bar{\Sigma}_h^{-1}$ and an extra higher order $O(\frac{1}{K})$ term:

$$\Gamma_h \leftarrow O\left(\sqrt{d}\cdot(\phi(\cdot,\cdot)^\top\widehat{\Lambda}_h^{-1}\phi(\cdot,\cdot))^{1/2}\right) + \frac{2H^3\sqrt{d}}{K}$$

Note such a design admits no explicit factor in $H$ in the main term (as opposed to Jin et al. (2021b)) therefore is the key for achieving adaptive/problem-dependent results (as we shall discuss later). The full algorithm VAPVI is stated in Algorithm 1. In particular, we halve the offline data into two independent parts with $\mathcal{D} = \{(s_h^\tau,a_h^\tau,r_h^\tau,s_h^{\tau\prime})\}_{\tau\in[K]}^{h\in[H]}$ and $\mathcal{D}' = \{(\bar{s}_h^\tau,\bar{a}_h^\tau,\bar{r}_h^\tau,\bar{s}_h^{\tau\prime})\}_{\tau\in[K]}^{h\in[H]}$ for different purposes (estimating variance and updating $Q$-values).

## 3.2 MAIN RESULT

We denote quantities $\mathcal{M}_1, \mathcal{M}_2, \mathcal{M}_3, \mathcal{M}_4$ as in the notation list A. Then VAPVI provides the following result. The complete proof is provided in Appendix C.

**Theorem 3.2.** *Let $K$ be the number of episodes. If $K > \max\{\mathcal{M}_1,\mathcal{M}_2,\mathcal{M}_3,\mathcal{M}_4\}$ and $\sqrt{d} > \xi$, where $\xi := \sup_{V\in[0,H],\ s'\sim P_h(s,a),\ h\in[H]}\left|\frac{r_h+V(s')-(\mathcal{T}_hV)(s,a)}{\sigma_V(s,a)}\right|$. Then for any $0 < \lambda < \kappa$, with probability $1-\delta$, for all policy $\pi$ simultaneously, the output $\widehat{\pi}$ of Algorithm 1 satisfies*

$$v^\pi - v^{\widehat{\pi}} \leq \widetilde{O}\big(\sqrt{d}\cdot\sum_{h=1}^H\mathbb{E}_\pi\left[\sqrt{\phi(\cdot,\cdot)^\top\Lambda_h^{-1}\phi(\cdot,\cdot)}\right]\big) + \frac{2H^4\sqrt{d}}{K}$$

*where $\Lambda_h = \sum_{k=1}^K\frac{\phi(s_h^k,a_h^k)\cdot\phi(s_h^k,a_h^k)^\top}{\sigma_{\widehat{V}_{h+1}(s_h^k,a_h^k)}^2} + \lambda I_d$. In particular, we have with probability $1-\delta$,*

$$v^\star - v^{\widehat{\pi}} \leq \widetilde{O}\big(\sqrt{d}\cdot\sum_{h=1}^H\mathbb{E}_{\pi^\star}\left[\sqrt{\phi(\cdot,\cdot)^\top\Lambda_h^{\star-1}\phi(\cdot,\cdot)}\right]\big) + \frac{2H^4\sqrt{d}}{K} \tag{3}$$

*where $\Lambda_h^\star = \sum_{k=1}^K\frac{\phi(s_h^k,a_h^k)\cdot\phi(s_h^k,a_h^k)^\top}{\sigma_{V_{h+1}^\star(s_h^k,a_h^k)}^2} + \lambda I_d$ and $\widetilde{O}$ hides universal constants and the Polylog terms.*

---

[6]The truncation used here is a standard treatment for making the estimator to be within the valid range.

---

**Algorithm 1** Variance-Aware Pessimistic Value Iteration (VAPVI)

1: **Input:** Dataset $\mathcal{D} = \{(s_h^\tau, a_h^\tau, r_h^\tau)\}_{\tau,h=1}^{K,H}$ $\mathcal{D}' = \{(\bar{s}_h^\tau, \bar{a}_h^\tau, \bar{r}_h^\tau)\}_{\tau,h=1}^{K,H}$. Universal constant $C$.
2: **Initialization:** Set $\widehat{V}_{H+1}(\cdot) \leftarrow 0$.
3: **for** $h = H, H-1, \ldots, 1$ **do**
4:     $\diamond$ Phase1: Regular Least-square Value Iteration for conditional variances
5:     Set $\bar{\Sigma}_h \leftarrow \sum_{\tau=1}^{K} \phi(\bar{s}_h^\tau, \bar{a}_h^\tau) \phi(\bar{s}_h^\tau, \bar{a}_h^\tau)^\top + \lambda I$
6:     Set $\bar{\beta}_h \leftarrow \bar{\Sigma}_h^{-1} \sum_{\tau=1}^{K} \phi(\bar{s}_h^\tau, \bar{a}_h^\tau) \cdot \widehat{V}_{h+1}(\bar{s}_{h+1}^\tau)^2$
7:     Set $\bar{\theta}_h \leftarrow \bar{\Sigma}_h^{-1} \sum_{\tau=1}^{K} \phi(\bar{s}_h^\tau, \bar{a}_h^\tau) \cdot \widehat{V}_{h+1}(\bar{s}_{h+1}^\tau)$
8:     Set $\left[\widehat{\mathrm{Var}}_h \widehat{V}_{h+1}\right](\cdot,\cdot) = \left\langle \phi(\cdot,\cdot), \bar{\beta}_h \right\rangle_{[0,(H-h+1)^2]} - \left[\left\langle \phi(\cdot,\cdot), \bar{\theta}_h \right\rangle_{[0,H-h+1]}\right]^2$
9:     Set $\widehat{\sigma}_h(\cdot,\cdot)^2 \leftarrow \max\{1, \widehat{\mathrm{Var}}_{P_h} \widehat{V}_{h+1}(\cdot,\cdot)\}$
10:    $\diamond$ Phase2: Weighted Least-square Value Iteration for pessimistic updates
11:    Set $\widehat{\Lambda}_h \leftarrow \sum_{\tau=1}^{K} \phi(s_h^\tau, a_h^\tau) \phi(s_h^\tau, a_h^\tau)^\top / \widehat{\sigma}^2(s_h^\tau, a_h^\tau) + \lambda \cdot I$,
12:    Set $\widehat{w}_h \leftarrow \widehat{\Lambda}_h^{-1} \left( \sum_{\tau=1}^{K} \phi(s_h^\tau, a_h^\tau) \cdot \left( r_h^\tau + \widehat{V}_{h+1}(s_{h+1}^\tau) \right) / \widehat{\sigma}^2(s_h^\tau, a_h^\tau) \right)$
13:    Set $\Gamma_h(\cdot,\cdot) \leftarrow C\sqrt{d} \cdot \left( \phi(\cdot,\cdot)^\top \widehat{\Lambda}_h^{-1} \phi(\cdot,\cdot) \right)^{1/2} + \frac{2H^3\sqrt{d}}{K}$     (Use $\Gamma_h^I$ for the improved version)
14:    Set $\bar{Q}_h(\cdot,\cdot) \leftarrow \phi(\cdot,\cdot)^\top \widehat{w}_h - \Gamma_h(\cdot,\cdot)$
15:    Set $\widehat{Q}_h(\cdot,\cdot) \leftarrow \min\left\{\bar{Q}_h(\cdot,\cdot), H-h+1\right\}^+$
16:    Set $\widehat{\pi}_h(\cdot \mid \cdot) \leftarrow \arg\max_{\pi_h} \left\langle \widehat{Q}_h(\cdot,\cdot), \pi_h(\cdot \mid \cdot) \right\rangle_{\mathcal{A}}, \widehat{V}_h(\cdot) \leftarrow \max_{\pi_h} \left\langle \widehat{Q}_h(\cdot,\cdot), \pi_h(\cdot \mid \cdot) \right\rangle_{\mathcal{A}}$
17: **end for**
18: **Output:** $\{\widehat{\pi}_h\}_{h=1}^{H}$.

---

Theorem 3.2 provides improvements over the existing best-known results and we now explain it. However, before that, we first discuss about our theorem condition.

**Comparing to Zhou et al. (2021a).** In the online regime, Zhou et al. (2021a) is the first result that achieves optimal regret rate with $O(dH\sqrt{T})$ in the linear (mixture) MDPs. However, this result requires the condition $d \geq H$ (their Theorem 6 and Remark 7). In offline RL, VAPVI only requires a milder condition $\sqrt{d} > \xi$ comparing to $d \geq H$ (since for any fixed $V \in [0,H]$, the standardized quantity $\frac{r+V(s')-(\mathcal{T}_h V)(s,a)}{\sigma_V(s,a)}$ is bounded by constant with high probability, *e.g.* by *chebyshev* inequality), which makes our result apply to a wider range of linear MDPs.

**Comparing to Jin et al. (2021b).** Jin et al. (2021b) first shows *pessimistic value iteration* (PEVI) is provably efficient for Linear MDPs in offline RL. VAPVI improves PEVI over $O(\sqrt{d})$ on the feature dimension, and improves the horizon dependence as $\Lambda_h \succcurlyeq \frac{1}{H^2}\Sigma_h$ implies $\Lambda_h^{-1} \preccurlyeq H^2\Sigma_h^{-1}$. In addition, when instantiate to the tabular case, *i.e.* $\phi(s,a) = \mathbf{1}_{s,a}$, VAPVI gives $O(\sqrt{d} \sum_{h,s,a} d_h^{\pi^\star}(s,a) \sqrt{\frac{\mathrm{Var}_{P_{s,a}}(r+V_{h+1}^\star)}{K \cdot d_h^\mu(s,a)}})$, which enjoys $O(\sqrt{H})$ improvement over PEVI (recall Example 3.1) and the order $O(H^{3/2})$ is tight (check Section G for the detailed derivation).

**Comparing to Xie et al. (2021a).** Their linear MDP guarantee in Theorem 3.2. enjoys the same rate as VAPVI in feature dimension but the horizon dependence is essentially the same as Jin et al. (2021b) (by translating $H \approx O(\frac{1}{1-\gamma})$) therefore is not optimal. The general function approximation scheme in Xie et al. (2021a) provides elegant characterizations for on-support error and off-support error, but the algorithmic framework is information-theoretical only (and the practical version PSPI will not yield the same learning guarantee). Also, due to the use finite function class and policy class, the reduction to linear MDP only works with finite action space. As a comparison, VAPVI has no constraints on any of these.

**Comparing to Zanette et al. (2021).** Concurrently, Zanette et al. (2021) considers offline RL with the linear Bellman complete model, which is more general than linear MDPs and, with the assumption $Q^\pi \leq 1$, their PACLE algorithm provides near-minimax optimal guarantee in this setting. However, when recovering to the standard setting $Q^\pi \in [0,H]$, their bound will rescale by an $H$ factor,[7] which could be suboptimal due to the variance-unawareness. The reason behind this is: when $Q^\pi \leq 1$, lack of variance information encoding will not matter, since in this case $\mathrm{Var}_P(V^\pi) \leq 1$ has constant

---

[7]Check their Footnote 2 in Page 9.

order (therefore will not affect the optimal rate); when $Q^\pi \in [0, H]$, $\mathrm{Var}_P(V^\pi)$ can be as large as $H^2$, effectively leveraging the variance information can help improve the sample efficiency, *e.g.* via *law of total variances*, just like VAPVI does. On the other hand, their guarantee also requires finite action space, due to the mirror descent style analysis. Nevertheless, we do point out Zanette et al. (2021) has improved state-action measure than VAPVI, as $\|E_\pi[\phi(\cdot, \cdot)]\|_{M^{-1}} \leq \mathbb{E}_\pi[\|\phi(\cdot, \cdot)\|_{M^{-1}}]$ by Jensen's inequality and that norm $\|\cdot\|_{M^{-1}}$ is convex for some positive-definite matrix $M$.

**Adaptive characterization and faster convergence.** Comparing to existing works, one major improvement is that the main term for VAPVI $\sqrt{d} \sum_{h=1}^{H} \mathbb{E}_{\pi^\star}\left[\sqrt{\phi(\cdot, \cdot)^\top \Lambda_h^{\star-1} \phi(\cdot, \cdot)}\right]$ admits no explicit dependence on $H$, which provides a more adaptive/instance-dependent characterization. For instance, if we ignore the technical treatment by taking $\lambda = 0$ and $\sigma_h^\star \approx \mathrm{Var}_P(V_{h+1}^\star)$, then for the **partially deterministic systems** (where there are $t$ stochastic $P_h$'s and $H - t$ deterministic $P_h$'s), the main term diminishes to $\sqrt{d} \sum_{i=1}^{t} \mathbb{E}_{\pi^\star}\left[\sqrt{\phi(\cdot, \cdot)^\top \Lambda_{h_i}^{\star-1} \phi(\cdot, \cdot)}\right]$ with $h_i \in \{h : s.t. P_h \text{ is stochastic}\}$ and can be a much smaller quantity when $t \ll H$. Furthermore, for the **fully deterministic system**, VAPVI automatically provides faster convergence rate $O(\frac{1}{K})$ from the higher order term, given that the main term degenerates to 0. Those adaptive/instance-dependent features are not enjoyed by (Xie et al., 2021a; Zanette et al., 2021), as they always provide the standard statistical rate $O(\frac{1}{\sqrt{K}})$ (also check Remark C.9 for a related discussion).

### 3.3 VAPVI-IMPROVED: FURTHER IMPROVEMENT IN STATE-ACTION DIMENSION

Can we further improve the VAPVI? Indeed, by deploying a carefully tuned tighter penalty, we are able to further improve the state-action dependence if the feature is non-negative ($\phi \geq 0$). Concretely, we replace the following $\Gamma_h^I$ in Algorithm 1 instead, and call the algorithm VAPVI-Improved (or VAPVI-I for short). The proof can be found in Appendix D.

$$\Gamma_h^I(s, a) \leftarrow \phi(s, a)^\top \left| \widehat{\Lambda}_h^{-1} \sum_{\tau=1}^{K} \frac{\phi(s_h^\tau, a_h^\tau) \cdot \left(r_h^\tau + \widehat{V}_{h+1}(s_{h+1}^\tau) - \left(\widehat{\mathcal{T}}_h \widehat{V}_{h+1}\right)(s_h^\tau, a_h^\tau)\right)}{\widehat{\sigma}_h^2(s_h^\tau, a_h^\tau)} \right| + \widetilde{O}\left(\frac{H^3 d/\kappa}{K}\right) \quad (4)$$

**Theorem 3.3.** *Suppose the feature is non-negative ($\phi \geq 0$). Let $K$ be the number of episodes. If $K > \max\{\mathcal{M}_1, \mathcal{M}_2, \mathcal{M}_3, \mathcal{M}_4\}$ and $\sqrt{d} > \xi$. Deploying $\Gamma_h^I$ (4) in Algorithm 1. Then for any $0 < \lambda < \kappa$, with probability $1 - \delta$, for all policy $\pi$ simultaneously, the output $\widehat{\pi}$ of Algorithm 1 (VAPVI-I) satisfies*

$$v^\pi - v^{\widehat{\pi}} \leq \widetilde{O}\left(\sqrt{d} \cdot \sum_{h=1}^{H} \sqrt{\mathbb{E}_\pi[\phi(\cdot, \cdot)]^\top \Lambda_h^{-1} \mathbb{E}_\pi[\phi(\cdot, \cdot)]}\right) + \widetilde{O}\left(\frac{H^4 d/\kappa}{K}\right)$$

*In particular, when choosing $\pi = \pi^\star$, the above guarantee holds true with $\Lambda_h^{-1}$ replaced by $\Lambda_h^{\star-1}$. Here $\Lambda_h^{-1}, \Lambda_h^{\star-1}, \xi$ are defined the same as Theorem 3.2.*

Theorem 3.3 maintains nearly all the features of Theorem 3.2 (except higher order term is slightly worse) and the dominate term evolves from $\mathbb{E}_\pi \|\phi\|_{\Lambda_h^{-1}}$ to $\|\mathbb{E}_\pi[\phi]\|_{\Lambda_h^{-1}}$. Clearly, the two bounds differ by the magnitude of Jensen's inequality. To provide a concrete view of how much improvement is made, we check the parameter dependence in the context of tabular MDPs (where we ignore the higher order term for conciseness). In particular, we compare the results under the single-policy concentrability.

**Assumption 3.4** (Rashidinejad et al. (2021); Xie et al. (2021b)). *There exists a optimal policy $\pi^\star$, s.t. $\sup_{h,s,a} d_h^{\pi^\star}(s, a)/d_h^\mu(s, a) := C^\star < \infty$, where $d^\pi$ is the marginal state-action probability under $\pi$.*

In tabular RL, $\phi(s, a) = \mathbf{1}_{s,a}$ and $d = S \cdot A$ ($S, A$ be the finite state, action cardinality), then

$$\text{Theorem 3.2} \rightarrow \sqrt{SA} \sum_{h}^{H} \sum_{s,a} d_h^{\pi^\star}(s, a) \sqrt{\frac{\mathrm{Var}_{P_{s,a}}(r + V_{h+1}^\star)}{K \cdot d_h^\mu(s, a)}} \leq \sqrt{\frac{H^3 C^\star S^2 A}{K}};$$

$$\text{Theorem 3.3} \rightarrow \sqrt{SA} \sum_{h}^{H} \sqrt{\sum_{s,a} d_h^{\pi^\star}(s, a)^2 \frac{\mathrm{Var}_{P_{s,a}}(r + V_{h+1}^\star)}{K \cdot d_h^\mu(s, a)}} \leq \sqrt{\frac{H^3 C^\star S A}{K}}. \quad (5)$$

Theorem 3.3 enjoys a $S$ state improvement over Theorem 3.2 and nearly recovers the minimax rate $\sqrt{\frac{H^3 C^\star S}{K}}$ (Xie et al., 2021b). The detailed derivation can be found in Appendix G. Also, to show our result is near-optimal, we provide the corresponding lower bound. The proof is in Appendix E.

**Theorem 3.5** (Minimax lower bound). *There exist a pair of universal constants $c, c' > 0$ such that given dimension $d$, horizon $H$ and sample size $K > c'd^3$, one can always find a family of linear MDP instances $\mathcal{M}$ such that (where $\Lambda_h^\star = \sum_{k=1}^K \frac{\phi(s_h^k, a_h^k) \cdot \phi(s_h^k, a_h^k)^\top}{\mathrm{Var}_h(V_{h+1}^\star)(s_h^k, a_h^k)}$ satisfies $(\Lambda_h^\star)^{-1}$ exists and $\mathrm{Var}_h(V_{h+1}^\star)(s_h^k, a_h^k) > 0 \; \forall M \in \mathcal{M}$)*

$$\inf_{\widehat{\pi}} \sup_{M \in \mathcal{M}} \mathbb{E}_M \left[ v^\star - v^{\widehat{\pi}} \right] / \left( \sqrt{d} \cdot \sum_{h=1}^H \sqrt{\mathbb{E}_{\pi^\star}[\phi]^\top (\Lambda_h^\star)^{-1} \mathbb{E}_{\pi^\star}[\phi]} \right) \geq c. \tag{6}$$

Theorem 3.5 nearly matches the main term in VAPVI-I (Theorem 3.3) and certifies it is near-optimal.

## 4 Proof Overview

Due to the space constraint, we could only provide a brief overview of the key proving ideas of the theorems. We begin with Theorem 3.2. First, by *the extended value difference lemma* (Lemma H.7), we can convert bounding the suboptimality gap of $v^\star - v^{\widehat{\pi}}$ to bounding $\sum_{h=1}^H 2 \cdot \mathbb{E}_\pi [\Gamma_h(s_h, a_h)]$, given that $|(\mathcal{T}_h \widehat{V}_{h+1} - \widehat{\mathcal{T}}_h \widehat{V}_{h+1})(s,a)| \leq \Gamma_h(s,a)$ for all $s, a, h$. To bound $\mathcal{T}_h \widehat{V}_{h+1} - \widehat{\mathcal{T}}_h \widehat{V}_{h+1}$, by decomposing it reduces to bounding the key quantity

$$\phi(s,a)^\top \widehat{\Lambda}_h^{-1} \left[ \sum_{\tau=1}^K \phi(s_h^\tau, a_h^\tau) \cdot \left( r_h^\tau + \widehat{V}_{h+1}(s_{h+1}^\tau) - \left( \mathcal{T}_h \widehat{V}_{h+1} \right)(s_h^\tau, a_h^\tau) \right) / \widehat{\sigma}_h^2(s_h^\tau, a_h^\tau) \right] \tag{7}$$

The term is treated in two steps. First, we bound the gap of $\left\| \sigma_{\widehat{V}_{h+1}}^2 - \widehat{\sigma}_h^2 \right\|$ so we can convert $\widehat{\sigma}_h^2$ to $\sigma_{\widehat{V}_{h+1}}^2$. Next, since $\mathrm{Var} \left[ r_h^\tau + \widehat{V}_{h+1}(s_{h+1}^\tau) - \left( \mathcal{T}_h \widehat{V}_{h+1} \right)(s_h^\tau, a_h^\tau) \mid s_h^\tau, a_h^\tau \right] \approx \sigma_{\widehat{V}_{h+1}}^2$, therefore by the variance-weighted scheme in (equation 7), we can leverage the recent technical development *Bernstein inequality for self-normalized martingale* (Lemma H.3) for acquiring the tight result, in contrast to the previous treatment of Hoeffding inequality for self-normalized martingale + Covering.[8] For the second part, one needs to further convert $\sigma_{\widehat{V}_{h+1}}^2$ to $\sigma_h^{\star 2}$ ($\Lambda_h^{-1}$ to $\Lambda_h^{\star -1}$) with appropriate concentrations. The proof of Theorem 3.3 is similar but with more complicated computations and relies on using the linear representation of $\phi$ in $\Gamma_h^I$ (4), so that the expectation over $\pi$ is inside the square root by taking expectation over the linear representation at the beginning. The lower bound proof uses a simple modification of Zanette et al. (2021) which consists of the reduction from learning to testing with Assouad's method, and the use of standard information inequalities (*e.g.* from total variation to KL divergence). For completeness, we provide the full proof in Appendix E.

## 5 Discussion and Conclusion

This work studies offline RL with linear MDP representation and contributes *Variance Aware Pessimistic Value Iteration* (VAPVI) which adopts the conditional variance information of the value function. VAPVI uses the estimated variances to reweight the Bellman residuals in the least-square pessimistic value iteration and provides improved offline learning bounds over the existing best-known results. VAPVI-I further improves over VAPVI in the state-action dimension and is near-minimax optimal. One highlight of the theorems is that our learning bounds are expressed in terms of system quantities, which automatically provide natural instance-dependent characterizations that previous results are short of.

On the other hand, while VAPVI/VAPVI-I close the existing gap from previous literature (Jin et al., 2021b; Xie et al., 2021a), the optimal guarantee is in the minimax sense. Although our upper bounds possess instance-dependent characterizations, the lower bound only holds true for a class of hard instances. In this sense, whether "instance-dependent optimality" can be achieved remains elusive in the current linear MDP setting (such a discussion is recently initiated in MAB problems (Xiao et al., 2021)). We leave this as future work.

---

[8]Variance-reweighting in (7) is important, since applying *Bernstein inequality for self-normalized martingale* (Lemma H.3) without variance-reweighting cannot provide any improvement.

ACKNOWLEDGMENTS

The authors would like to thank Quanquan Gu for explaining Min et al. (2021) and introducing a couple of related literatures. Ming Yin would like to thank Zhuoran Yang for the helpful suggestions and Dan Qiao for a careful proofreading. Mengdi Wang gratefully acknowledges funding from Office of Naval Research (ONR) N00014-21-1-2288, Air Force Office of Scientific Research (AFOSR) FA9550-19-1-0203, and NSF 19-589, CMMI-1653435. Yu-Xiang Wang gratefully acknowledges funding from National Science Foundation (NSF) #2007117 and #2003257.

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

# Appendix

## A  Notation List

| | |
|---|---|
| $\Sigma_h^p$ | $\mathbb{E}_{\mu,h}\left[\phi(s,a)\phi(s,a)^\top\right]$ |
| $\Lambda_h^p$ | $\mathbb{E}_{\mu,h}\left[\sigma_{V_{h+1}}(s,a)^{-2}\phi(s,a)\phi(s,a)^\top\right]$ |
| $\kappa$ | $\min_h \lambda_{\min}(\Sigma_h^p)$ |
| $\iota$ | $\min_h \lambda_{\min}(\Lambda_h^p) \geq \kappa/H^2$ for any $V_h$ |
| $\sigma_V^2(s,a)$ | $\max\{1, \mathrm{Var}_{P_h}(V)(s,a)\}$ for any $V$ |
| $\sigma_{V_{h+1}}^2(s,a)$ | $\max\{1, \mathrm{Var}_{P_h}(V_{h+1})(s,a)\}$ |
| $\widehat{\sigma}_{V_h}^2(s,a)$ | $\max\{1, \widehat{\mathrm{Var}}_h V_{h+1}(s,a)\}$ |
| $\bar{\Sigma}_h$ | $\sum_{\tau=1}^{K} \phi(\bar{s}_h^\tau, \bar{a}_h^\tau)\phi(\bar{s}_h^\tau, \bar{a}_h^\tau)^\top + \lambda I_d$ |
| $\widehat{\Lambda}_h$ | $\sum_{k=1}^{K} \phi(s_h^k, a_h^k)\phi(s_h^k, a_h^k)^\top/\widehat{\sigma}_h^2(s_h^k, a_h^k) + \lambda I_d$ |
| $\mathcal{M}_1$ | $\max\{2\lambda, 128\log(2d/\delta), 128H^4\log(2d/\delta)/\kappa^2\}$ |
| $\mathcal{M}_2$ | $\max\{\frac{\lambda^2}{\kappa\log((\lambda+K)H/\lambda\delta)}, 96^2 H^{12}d\log((\lambda+K)H/\lambda\delta)/\kappa^5\}$ |
| $\mathcal{M}_3$ | $\max\left\{512H^4/\kappa^2\log\left(\frac{2d}{\delta}\right), 4\lambda H^2/\kappa\right\}$ |
| $\mathcal{M}_4$ | $12\sqrt{H^4 d\log((\lambda+K)H/\lambda\delta)/\kappa}$ |
| $\delta$ | Failure probability |
| $\xi$ | $\sup_{V\in[0,H],\, s'\sim P_h(s,a),\, h\in[H]} \left|\frac{r_h + V(s') - (\mathcal{T}_h V)(s,a)}{\sigma_V(s,a)}\right|$ |
| $C_{H,d,\kappa,K}$ | $36\sqrt{\frac{H^4 d^3}{\kappa}\log\left(\frac{(\lambda+K)2KdH^2}{\lambda\delta}\right)} + 12\lambda\frac{H^2\sqrt{d}}{\kappa}$ |

## B  Extended Literature Review

### B.1  Linear model representation and its extension in online RL

There are numerous works in online RL that study linear model representations. Yang and Wang (2019; 2020); Jin et al. (2020) propose Linear MDP, which assumes the transition kernel and the reward are linear in given features. Cai et al. (2020); Ayoub et al. (2020); Modi et al. (2020); Zhou et al. (2021b) propose Linear mixture MDP, which assumes the transition probability is a linear combination of some base kernels. Linear Bellman Complete model (Zanette et al., 2020) generalizes linear MDP model by allowing linear functions to approximate the $Q$-function and the function class is closed under the Bellman update. The notion of low Bellman rank (Jiang et al., 2017) subsumes not only linear MDPs but also models including linear quadratic regulator (LQR), Reactive POMDP (Krishnamurthy et al., 2016) and Block MDP (Du et al., 2019). There are also other models, *e.g.* factored MDP (Sun et al., 2019), Bellman Eluder dimension (Jin et al., 2021a) and Bilinear class (Du et al., 2021). With the linear MDP model itself, there are also fruitful extensions, *e.g.* gap-dependent analysis with logarithmic regret (He et al., 2021), low-switching cost RL (Gao et al., 2021), safe RL (Amani et al., 2021), reward-free RL (Wang et al., 2020), generalized linear model (GLM) (Wang et al., 2021b), two-player markov game (Chen et al., 2021b) and generative adversarial learning (Liu et al., 2021). In particular, Zhou et al. (2021a) shows UCRL-VTR+ is near-minimax optimal when feature dimension $d \geq H$.

### B.2  Existing results in offline RL with model representations

**Offline RL with general function representations.** The finite sample analysis of offline RL with function approximation is initially conducted by Fitted $Q$-Iteration (FQI) type algorithms and can be dated back to (Munos, 2003; Szepesvári and Munos, 2005; Antos et al., 2008a;b). Later, Chen and Jiang (2019); Le et al. (2019); Xie and Jiang (2020) follow this line of research and derive the

improved learning results. However, owing to the aim for tackling general function approximation, those learning bounds are expressed in terms of the stringent *concentrability coefficients* (therefore, are less adaptive to individual instances) and are usually only *information-theoretical*, due to the computational intractability of the optimization procedure over the general function classes. Other works impose weaker assumptions (*e.g.* partial coverage (Liu et al., 2020; Kidambi et al., 2020; Uehara and Sun, 2021)), and their finite sample analysis are generally suboptimal in terms of $H$ or the effective horizon $(1 - \gamma)^{-1}$.

**Offline RL with tabular models.** For tabular MDPs, tight learning bounds can be achieved under several data-coverage assumptions. For the class of problems with uniform data-visitation measure $d_m$, the near-optimal sample complexity bound has the rate $O(H^3/d_m\epsilon^2)$ for time-inhomogeneous MDPs (Yin et al., 2021) and $O(H^2/d_m\epsilon)$ for time-homogeneous MDPs (Yin and Wang, 2021a; Ren et al., 2021). Under the single concentrability assumption, the tight rate $O(H^3 SC^\star/\epsilon^2)$ is obtained by Xie et al. (2021b). In particular, the recent study Yin and Wang (2021b) introduces the *intrinsic offline learning bound* that is not only instance-dependent but also subsumes previous optimal results. More recently, Shi et al. (2022) uses model-free approach to achieve minimax rate with a larger $\epsilon$-range.

**Offline RL with linear model representations.** Recently, there are more focus on studying the provable efficient offline RL under the linear model representations. Jin et al. (2021b) first shows offline RL with linear MDP is provably efficient by *the pessimistic value iteration* (PEVI), which is an offline counterpart of LSVI-UCB in Jin et al. (2020). Their analysis deviates from their lower bound by a factor of $d \cdot H$ (check their Theorem 4.4 and 4.6). Later, Xie et al. (2021a) considers function approximation under the Bellman-consistent assumptions, and, when realized to linear MDP setting, improve the sample complexity guarantee of Jin et al. (2021b) by a order $O(d)$ (Theorem 3.2). However, their improvement only holds for finite action space (due to the dependence $\log |\mathcal{A}|$) and by the direct reduction (from Theorem 3.1) their result does not imply a computationally tractable algorithm. In addition, there is no improvement on the horizon dependence. Concurrently, Zanette et al. (2021) considers the Linear Bellman Complete model (which originates from its online version Zanette et al. (2020)) and designs the *actor-critic* style algorithm that achieves tight result under the assumption that the value function is bounded by 1. While their algorithm is efficient (which is based on solving a sequence of second-order cone programs), the resulting learning bound requires the action space to be finite due to the mirror descent/natural policy gradient updates in the *Actor* procedure (Agarwal et al., 2021). Besides, assuming the value function to be less than 1 simplifies the challenges in dealing with horizon $H$ since when rescale their result to $[0, H]$, there is a $H$ factor blow-up, which makes no improvement in the horizon dependence comparing to Jin et al. (2021b). On the other hand, Wang et al. (2021a); Zanette (2021) study the statistical hardness of offline RL with linear representations by proofing the exponential lower bounds. As a result, none of the existing algorithms can achieve the statistical limit for the well-structured linear MDP model with the general (infinite or continuous) state-action spaces in the offline regime.

## C    PROOFS IN SECTION 3.2

Instead of proofing the result for $v^\star - v^{\widehat{\pi}}$, in most parts of the proof we deal with $V_1^\star - V_1^{\widehat{\pi}}$, which is more general.

### C.1    SOME PREPARATIONS

Define the Bellman update error $\zeta_h(s, a) := (\mathcal{T}_h \widehat{V}_{h+1})(s, a) - \widehat{Q}_h(s, a)$ and recall $\widehat{\pi}_h(s) = \arg\max_{\pi_h} \langle \widehat{Q}_h(s, \cdot), \pi_h(\cdot \mid s) \rangle_\mathcal{A}$, then by the direct application of Lemma H.8

$$V_1^\pi(s) - V_1^{\widehat{\pi}}(s) \leq \sum_{h=1}^{H} \mathbb{E}_\pi \left[ \zeta_h(s_h, a_h) \mid s_1 = s \right] - \sum_{h=1}^{H} \mathbb{E}_{\widehat{\pi}} \left[ \zeta_h(s_h, a_h) \mid s_1 = s \right]. \tag{8}$$

The next lemma shows it is sufficient to bound the pessimistic penalty, which is the key in the proof.

**Lemma C.1.** *Suppose with probability $1 - \delta$, it holds for all $h, s, a \in [H] \times \mathcal{S} \times A$ that $|(\mathcal{T}_h \widehat{V}_{h+1} - \widehat{\mathcal{T}}_h \widehat{V}_{h+1})(s, a)| \leq \Gamma_h(s, a)$, then it implies $\forall s, a, h \in \mathcal{S} \times \mathcal{A} \times [H]$, $0 \leq \zeta_h(s, a) \leq 2\Gamma_h(s, a)$.*

*Furthermore, it holds for any policy $\pi$ simultaneously, with probability $1 - \delta$,*

$$V_1^\pi(s) - V_1^{\widehat{\pi}}(s) \leq \sum_{h=1}^H 2 \cdot \mathbb{E}_\pi\left[\Gamma_h(s_h, a_h) \mid s_1 = s\right].$$

*Proof of Lemma C.1.* We first show given $|(\mathcal{T}_h\widehat{V}_{h+1} - \widehat{\mathcal{T}}_h\widehat{V}_{h+1})(s,a)| \leq \Gamma_h(s,a)$, then $0 \leq \zeta_h(s,a) \leq 2\Gamma_h(s,a), \forall s, a, h \in \mathcal{S} \times \mathcal{A} \times [H]$.

**Step1:** we first show $0 \leq \zeta_h(s,a), \forall s, a, h \in \mathcal{S} \times \mathcal{A} \times [H]$.

Indeed, if $\bar{Q}_h(s,a) \leq 0$, then by definition $\widehat{Q}_h(s,a) = 0$ and in this case $\zeta_h(s,a) := (\mathcal{T}_h\widehat{V}_{h+1})(s,a) - \widehat{Q}_h(s,a) = (\mathcal{T}_h\widehat{V}_{h+1})(s,a) \geq 0$; if $\bar{Q}_h(s,a) > 0$, then $\widehat{Q}_h(s,a) \leq \bar{Q}_h(s,a)$ and

$$\begin{aligned}
\zeta_h(s,a) :=& (\mathcal{T}_h\widehat{V}_{h+1})(s,a) - \widehat{Q}_h(s,a) \geq (\mathcal{T}_h\widehat{V}_{h+1})(s,a) - \bar{Q}_h(s,a) \\
=& (\mathcal{T}_h\widehat{V}_{h+1})(s,a) - (\widehat{\mathcal{T}}_h\widehat{V}_{h+1})(s,a) + \Gamma_h(s,a) \geq 0.
\end{aligned}$$

**Step2:** next we show $\zeta_h(s,a) \leq 2\Gamma_h(s,a), \forall s, a, h \in \mathcal{S} \times \mathcal{A} \times [H]$.

Indeed, we have $\widehat{Q}_h(s,a) = \max(\bar{Q}_h(s,a), 0)$ and this is because: $\bar{Q}_h(x,a) = (\widehat{\mathcal{T}}_h\widehat{V}_{h+1})(x,a) - \Gamma_h(x,a) \leq (\mathcal{T}_h\widehat{V}_{h+1})(x,a) \leq H - h + 1$. Therefore, in this case we have:

$$\begin{aligned}
\zeta_h(s,a) :=& (\mathcal{T}_h\widehat{V}_{h+1})(s,a) - \widehat{Q}_h(s,a) \leq (\mathcal{T}_h\widehat{V}_{h+1})(s,a) - \bar{Q}_h(s,a) \\
=& (\mathcal{T}_h\widehat{V}_{h+1})(s,a) - (\widehat{\mathcal{T}}_h\widehat{V}_{h+1})(s,a) + \Gamma_h(s,a) \leq 2 \cdot \Gamma_h(s,a).
\end{aligned}$$

For the last statement, denote $\mathfrak{F} := \{0 \leq \zeta_h(s,a) \leq 2\Gamma_h(s,a), \ \forall s, a, h \in \mathcal{S} \times \mathcal{A} \times [H]\}$. Note conditional on $\mathfrak{F}$, then by equation 8, $V_1^\pi(s) - V_1^{\widehat{\pi}}(s) \leq \sum_{h=1}^H 2 \cdot \mathbb{E}_\pi[\Gamma_h(s_h, a_h) \mid s_1 = s]$ holds for any policy $\pi$ almost surely. Therefore,

$$\begin{aligned}
&\mathbb{P}\left[\forall \pi, \ V_1^\pi(s) - V_1^{\widehat{\pi}}(s) \leq \sum_{h=1}^H 2 \cdot \mathbb{E}_\pi[\Gamma_h(s_h, a_h) \mid s_1 = s].\right] \\
=&\mathbb{P}\left[\forall \pi, \ V_1^\pi(s) - V_1^{\widehat{\pi}}(s) \leq \sum_{h=1}^H 2 \cdot \mathbb{E}_\pi[\Gamma_h(s_h, a_h) \mid s_1 = s]\Big|\mathfrak{F}\right] \cdot \mathbb{P}[\mathfrak{F}] \\
+&\mathbb{P}\left[\forall \pi, \ V_1^\pi(s) - V_1^{\widehat{\pi}}(s) \leq \sum_{h=1}^H 2 \cdot \mathbb{E}_\pi[\Gamma_h(s_h, a_h) \mid s_1 = s]\Big|\mathfrak{F}^c\right] \cdot \mathbb{P}[\mathfrak{F}^c] \\
\geq&\mathbb{P}\left[\forall \pi, \ V_1^\pi(s) - V_1^{\widehat{\pi}}(s) \leq \sum_{h=1}^H 2 \cdot \mathbb{E}_\pi[\Gamma_h(s_h, a_h) \mid s_1 = s]\Big|\mathfrak{F}\right] \cdot \mathbb{P}[\mathfrak{F}] \geq 1 \cdot \mathbb{P}[\mathfrak{F}] \geq 1 - \delta,
\end{aligned}$$

which finishes the proof.

$\square$

## C.2  BOUNDING $\left|(\mathcal{T}_h\widehat{V}_{h+1})(s,a) - (\widehat{\mathcal{T}}_h\widehat{V}_{h+1})(s,a)\right|$.

By Lemma C.1, it remains to bound $|(\mathcal{T}_h\widehat{V}_{h+1})(s,a) - (\widehat{\mathcal{T}}_h\widehat{V}_{h+1})(s,a)|$. Suppose $w_h$ is the coefficient corresponding to the $\mathcal{T}_h\widehat{V}_{h+1}$ (such $w_h$ exists by Lemma H.9), *i.e.* $\mathcal{T}_h\widehat{V}_{h+1} = \phi^\top w_h$, and recall

$(\widehat{\mathcal{T}_h}\widehat{V}_{h+1})(s,a) = \phi(s,a)^\top \widehat{w}_h$, then:

$$\left(\mathcal{T}_h\widehat{V}_{h+1}\right)(s,a) - \left(\widehat{\mathcal{T}_h}\widehat{V}_{h+1}\right)(s,a) = \phi(s,a)^\top (w_h - \widehat{w}_h)$$

$$=\phi(s,a)^\top w_h - \phi(s,a)^\top \widehat{\Lambda}_h^{-1} \left(\sum_{\tau=1}^K \phi(s_h^\tau, a_h^\tau) \cdot \left(r_h^\tau + \widehat{V}_{h+1}\left(s_{h+1}^\tau\right)\right)/\widehat{\sigma}_h^2(s_h^\tau, a_h^\tau)\right)$$

$$=\underbrace{\phi(s,a)^\top w_h - \phi(s,a)^\top \widehat{\Lambda}_h^{-1} \left(\sum_{\tau=1}^K \phi(s_h^\tau, a_h^\tau) \cdot \left(\mathcal{T}_h\widehat{V}_{h+1}\right)(s_h^\tau, a_h^\tau)/\widehat{\sigma}_h^2(s_h^\tau, a_h^\tau)\right)}_{\text{(i)}}$$

$$+\underbrace{\phi(s,a)^\top \widehat{\Lambda}_h^{-1} \left(\sum_{\tau=1}^K \phi(s_h^\tau, a_h^\tau) \cdot \left(r_h^\tau + \widehat{V}_{h+1}\left(s_{h+1}^\tau\right) - \left(\mathcal{T}_h\widehat{V}_{h+1}\right)(s_h^\tau, a_h^\tau)\right)/\widehat{\sigma}_h^2(s_h^\tau, a_h^\tau)\right)}_{\text{(ii)}}.$$

$$(9)$$

The term (i) is dealt by the following lemma.

**Lemma C.2.** *Recall $\kappa$ in Assumption 2.2. Suppose $K \geq \max\left\{512H^4/\kappa^2 \log\left(\frac{2d}{\delta}\right), 4\lambda H^2/\kappa\right\}$, then with probability $1-\delta$, for all $s,a,h \in \mathcal{S} \times \mathcal{A} \times [H]$*

$$\left|\phi(s,a)^\top w_h - \phi(s,a)^\top \widehat{\Lambda}_h^{-1} \left(\sum_{\tau=1}^K \phi(s_h^\tau, a_h^\tau) \cdot \left(\mathcal{T}_h\widehat{V}_{h+1}\right)(s_h^\tau, a_h^\tau)/\widehat{\sigma}^2(s_h^\tau, a_h^\tau)\right)\right| \leq \frac{2\lambda H^3\sqrt{d}/\kappa}{K}.$$

*Proof.* Recall $\mathcal{T}_h\widehat{V}_{h+1} = \phi^\top w_h$ and apply Lemma H.6, we obtain with probability $1-\delta$, for all $s,a,h \in \mathcal{S} \times \mathcal{A} \times [H]$,

$$\phi(s,a)^\top w_h - \phi(s,a)^\top \widehat{\Lambda}_h^{-1} \left(\sum_{\tau=1}^K \phi(s_h^\tau, a_h^\tau) \cdot \left(\mathcal{T}_h\widehat{V}_{h+1}\right)(s_h^\tau, a_h^\tau)/\widehat{\sigma}^2(s_h^\tau, a_h^\tau)\right)$$

$$=\phi(s,a)^\top w_h - \phi(s,a)^\top \widehat{\Lambda}_h^{-1} \left(\sum_{\tau=1}^K \phi(s_h^\tau, a_h^\tau) \cdot \phi(s_h^\tau, a_h^\tau)^\top w_h/\widehat{\sigma}^2(s_h^\tau, a_h^\tau)\right)$$

$$=\phi(s,a)^\top w_h - \phi(s,a)^\top \widehat{\Lambda}_h^{-1}\left(\widehat{\Lambda}_h - \lambda I\right) w_h = \lambda \cdot \phi(s,a)^\top \widehat{\Lambda}_h^{-1} w_h$$

$$\leq \lambda \|\phi(s,a)\|_{\widehat{\Lambda}_h^{-1}} \cdot \|w_h\|_{\widehat{\Lambda}_h^{-1}} \leq \frac{\lambda}{K} \|\phi(s,a)\|_{(\tilde{\Lambda}_h^p)^{-1}} \cdot \|w_h\|_{(\tilde{\Lambda}_h^p)^{-1}}$$

$$\leq \frac{\lambda}{K} 1 \cdot \sqrt{\left\|(\tilde{\Lambda}_h^p)^{-1}\right\|} \cdot 2H\sqrt{d} \cdot \sqrt{\left\|(\tilde{\Lambda}_h^p)^{-1}\right\|}$$

where $\tilde{\Lambda}_h^p := \mathbb{E}_{\mu,h}\left[\widehat{\sigma}_h(s,a)^{-2}\phi(s,a)\phi(s,a)^\top\right]$ and the second inequality is by Lemma H.6 (with $\phi' = \phi/\widehat{\sigma}_h$ and $\|\phi/\widehat{\sigma}_h\| \leq \|\phi\| \leq 1 := C$) and the third inequality uses $\sqrt{a^\top \cdot A \cdot a} \leq \sqrt{\|a\|_2 \|A\|_2 \|a\|_2} = \|a\|_2 \sqrt{\|A\|_2}$ with $a$ to be either $\phi$ or $w_h$. Moreover, $\lambda_{\min}(\tilde{\Lambda}_h^p) \geq \kappa/\max_{h,s,a} \widehat{\sigma}_h(s,a)^2 \geq \kappa/H^2$ implies $\left\|(\tilde{\Lambda}_h^p)^{-1}\right\| \leq H^2/\kappa$, therefore for all $s,a,h \in \mathcal{S} \times \mathcal{A} \times [H]$, with probability $1-\delta$

$$\left|\phi(s,a)^\top w_h - \phi(s,a)^\top \widehat{\Lambda}_h^{-1} \left(\sum_{\tau=1}^K \phi(s_h^\tau, a_h^\tau) \cdot \left(\mathcal{T}_h\widehat{V}_{h+1}\right)(s_h^\tau, a_h^\tau)/\widehat{\sigma}^2(s_h^\tau, a_h^\tau)\right)\right| \leq \frac{2\lambda H^3\sqrt{d}/\kappa}{K}.$$

$\square$

For term (ii), denote: $x_\tau = \frac{\phi(s_h^\tau, a_h^\tau)}{\widehat{\sigma}(s_h^\tau, a_h^\tau)}$, $\quad \eta_\tau = \left( r_h^\tau + \widehat{V}_{h+1}\left(s_{h+1}^\tau\right) - \left(\mathcal{T}_h \widehat{V}_{h+1}\right)(s_h^\tau, a_h^\tau) \right) / \widehat{\sigma}(s_h^\tau, a_h^\tau)$,
then by Cauchy inequality it follows

$$
\left| \phi(s,a)^\top \widehat{\Lambda}_h^{-1} \left( \sum_{\tau=1}^{K} \phi(s_h^\tau, a_h^\tau) \cdot \left( r_h^\tau + \widehat{V}_{h+1}\left(s_{h+1}^\tau\right) - \left(\mathcal{T}_h \widehat{V}_{h+1}\right)(s_h^\tau, a_h^\tau) \right) / \widehat{\sigma}_h^2(s_h^\tau, a_h^\tau) \right) \right|
$$

$$
\leq \sqrt{\phi(s,a)^\top \widehat{\Lambda}_h^{-1} \phi(s,a)} \cdot \| \sum_{\tau=1}^{K} x_\tau \eta_\tau \|_{\widehat{\Lambda}_h^{-1}}
\tag{10}
$$

### C.2.1 ANALYZING THE TERM $\sqrt{\phi(s,a)\widehat{\Lambda}_h^{-1}\phi(s,a)}$

Recall (in Theorem 3.2) the estimated $\widehat{\Lambda}_h = \sum_{\tau=1}^{K} \phi\left(s_h^\tau, a_h^\tau\right) \phi\left(s_h^\tau, a_h^\tau\right)^\top / \widehat{\sigma}^2(s_h^\tau, a_h^\tau) + \lambda \cdot I$ and
$\Lambda_h = \sum_{\tau=1}^{K} \phi(s_h^\tau, a_h^\tau)^\top \phi(s_h^\tau, a_h^\tau) / \sigma_{\widehat{V}_{h+1}}^2(s_h^\tau, a_h^\tau) + \lambda I$. Then we have the following lemma to
control the term $\sqrt{\phi(s,a)\widehat{\Lambda}_h^{-1}\phi(s,a)}$.

**Lemma C.3.** *Denote the quantities* $C_1 = \max\{2\lambda, 128\log(2d/\delta), 128H^4\log(2d/\delta)/\kappa^2\}$ *and*
$C_2 = \max\{\frac{\lambda^2}{\kappa \log((\lambda+K)H/\lambda\delta)}, 96^2 H^{12} d\log((\lambda+K)H/\lambda\delta)/\kappa^5\}$. *Suppose the number of episode*
$K$ *satisfies* $K > \max\{C_1, C_2\}$, *then with probability* $1 - \delta$,

$$
\sqrt{\phi(s,a)\widehat{\Lambda}_h^{-1}\phi(s,a)} \leq 2\sqrt{\phi(s,a)\Lambda_h^{-1}\phi(s,a)}, \quad \forall s,a \in \mathcal{S} \times \mathcal{A}.
$$

*Proof of Lemma C.3.* By definition $\sqrt{\phi(s,a)\widehat{\Lambda}_h^{-1}\phi(s,a)} = \|\phi(s,a)\|_{\widehat{\Lambda}_h^{-1}}$. Then denote

$$
\widehat{\Lambda}_h' = \frac{1}{K}\widehat{\Lambda}_h, \quad \Lambda_h' = \frac{1}{K}\Lambda_h,
$$

where $\Lambda_h = \sum_{\tau=1}^{K} \phi(s_h^\tau, a_h^\tau)^\top \phi(s_h^\tau, a_h^\tau) / \sigma_{\widehat{V}_{h+1}}^2(s_h^\tau, a_h^\tau) + \lambda I$. Under the condition of $K$, by
Lemma C.7, with probability $1 - \delta$

$$
\left\| \widehat{\Lambda}_h' - \Lambda_h' \right\| \leq \sup_{s,a} \left\| \frac{\phi(s,a)\phi(s,a)^\top}{\widehat{\sigma}_h^2(s,a)} - \frac{\phi(s,a)\phi(s,a)^\top}{\sigma_{\widehat{V}_{h+1}}^2(s,a)} \right\|
$$

$$
\leq \sup_{s,a} \left| \frac{\widehat{\sigma}_h^2(s,a) - \sigma_{\widehat{V}_{h+1}}^2(s,a)}{\widehat{\sigma}_h^2(s,a)\sigma_{\widehat{V}_{h+1}}^2(s,a)} \right| \cdot \|\phi(s,a)\|^2 \leq \sup_{s,a} \left| \frac{\widehat{\sigma}_h^2(s,a) - \sigma_{\widehat{V}_{h+1}}^2(s,a)}{1} \right| \cdot 1 \tag{11}
$$

$$
\leq 12\sqrt{\frac{H^4 d}{\kappa K}\log\left(\frac{(\lambda+K)H}{\lambda\delta}\right)} + 12\lambda\frac{H^2\sqrt{d}}{\kappa K}.
$$

Next by Lemma H.5 (with $\phi$ to be $\phi/\sigma_{\widehat{V}_{h+1}}$ and $C = 1$), it holds with probability $1 - \delta$,

$$
\left\| \Lambda_h' - \left( \mathbb{E}_{\mu,h}[\phi(s,a)\phi(s,a)^\top / \sigma_{\widehat{V}_{h+1}}^2(s,a)] + \frac{\lambda}{K}I_d \right) \right\| \leq \frac{4\sqrt{2}}{\sqrt{K}}\left( \log\frac{2d}{\delta} \right)^{1/2}.
$$

Therefore by *Weyl's spectrum theorem* and the condition $K > \max\{2\lambda, 128\log(2d/\delta), 128H^4\log(2d/\delta)/\kappa^2\}$, the above implies

$$
\begin{aligned}
\|\Lambda_h'\| = \lambda_{\max}(\Lambda_h') &\leq \lambda_{\max}\left(\mathbb{E}_{\mu,h}[\phi(s,a)\phi(s,a)^\top/\sigma_{\widehat{V}_{h+1}}^2(s,a)]\right) + \frac{\lambda}{K} + \frac{4\sqrt{2}}{\sqrt{K}}\left(\log\frac{2d}{\delta}\right)^{1/2} \\
&= \left\|\mathbb{E}_{\mu,h}[\phi(s,a)\phi(s,a)^\top/\sigma_{\widehat{V}_{h+1}}^2(s,a)]\right\|_2 + \frac{\lambda}{K} + \frac{4\sqrt{2}}{\sqrt{K}}\left(\log\frac{2d}{\delta}\right)^{1/2} \\
&\leq \|\phi(s,a)\|^2 + \frac{\lambda}{K} + \frac{4\sqrt{2}}{\sqrt{K}}\left(\log\frac{2d}{\delta}\right)^{1/2} \leq 1 + \frac{\lambda}{K} + \frac{4\sqrt{2}}{\sqrt{K}}\left(\log\frac{2d}{\delta}\right)^{1/2} \leq 2,
\end{aligned}
$$

$$
\begin{aligned}
\lambda_{\min}(\Lambda_h') &\geq \lambda_{\min}\left(\mathbb{E}_{\mu,h}[\phi(s,a)\phi(s,a)^\top/\sigma_{\widehat{V}_{h+1}}^2(s,a)]\right) + \frac{\lambda}{K} - \frac{4\sqrt{2}}{\sqrt{K}}\left(\log\frac{2d}{\delta}\right)^{1/2} \\
&\geq \lambda_{\min}\left(\mathbb{E}_{\mu,h}[\phi(s,a)\phi(s,a)^\top/\sigma_{\widehat{V}_{h+1}}^2(s,a)]\right) - \frac{4\sqrt{2}}{\sqrt{K}}\left(\log\frac{2d}{\delta}\right)^{1/2} \\
&\geq \frac{\kappa}{H^2} - \frac{4\sqrt{2}}{\sqrt{K}}\left(\log\frac{2d}{\delta}\right)^{1/2} \geq \frac{\kappa}{2H^2}.
\end{aligned}
$$

Hence with probability $1 - \delta$, $\|\Lambda_h'\| \leq 2$ and $\left\|\Lambda_h'^{-1}\right\| = 1/\lambda_{\min}(\Lambda_h') \leq 2H^2/\kappa$. Similarly, one can show $\left\|\widehat{\Lambda}_h'^{-1}\right\| \leq 2H^2/\kappa$ with high probability.

Now apply Lemma H.4 to $\widehat{\Lambda}_h'$ and $\Lambda_h'$ and a union bound, we obtain with probability $1 - \delta$, for all $s, a$

$$
\begin{aligned}
\|\phi(s,a)\|_{\widehat{\Lambda}_h'^{-1}} &\leq \left[1 + \sqrt{\left\|\Lambda_h'^{-1}\right\|\|\Lambda_h'\| \cdot \left\|\widehat{\Lambda}_h'^{-1}\right\| \cdot \left\|\widehat{\Lambda}_h' - \Lambda_h'\right\|}\right] \cdot \|\phi(s,a)\|_{\Lambda_h'^{-1}} \\
&\leq \left[1 + \sqrt{\frac{2H^2}{\kappa}\cdot 1 \cdot \frac{2H^2}{\kappa}\cdot\left\|\widehat{\Lambda}_h' - \Lambda_h'\right\|}\right] \cdot \|\phi(s,a)\|_{\Lambda_h'^{-1}} \\
&\leq \left[1 + \sqrt{\frac{48H^4}{\kappa^2}\left(\sqrt{\frac{H^4d}{\kappa K}\log\left(\frac{(\lambda+K)H}{\lambda\delta}\right)} + \lambda\frac{H^2\sqrt{d}}{\kappa K}\right)}\right] \cdot \|\phi(s,a)\|_{\Lambda_h'^{-1}} \\
&\leq \left[1 + \sqrt{\frac{96H^4}{\kappa^2}\sqrt{\frac{H^4d}{\kappa K}\log\left(\frac{(\lambda+K)H}{\lambda\delta}\right)}}\right] \cdot \|\phi(s,a)\|_{\Lambda_h'^{-1}} \leq 2\|\phi(s,a)\|_{\Lambda_h'^{-1}}
\end{aligned}
$$

where the third inequality uses equation 11 and the last and the second last inequality use $K > \max\{\frac{\lambda^2}{\kappa\log((\lambda+K)H/\lambda\delta)}, 96^2H^{12}d\log((\lambda+K)H/\lambda\delta)/\kappa^5\}$. Note the above is equivalent to $\sqrt{\phi(s,a)\widehat{\Lambda}_h^{-1}\phi(s,a)} \leq 2\sqrt{\phi(s,a)\Lambda_h^{-1}\phi(s,a)}$ by multiplying $1/\sqrt{K}$ on both sides.

$\square$

### C.2.2 Analyzing the term $\|\sum_{\tau=1}^K x_\tau\eta_\tau\|_{\widehat{\Lambda}^{-1}}$

**Lemma C.4.** *Recall $x_\tau = \frac{\phi(s_h^\tau,a_h^\tau)}{\widehat{\sigma}(s_h^\tau,a_h^\tau)}$ and $\eta_\tau = \left(r_h^\tau + \widehat{V}_{h+1}\left(s_{h+1}^\tau\right) - \left(\mathcal{T}_h\widehat{V}_{h+1}\right)(s_h^\tau,a_h^\tau)\right)/\widehat{\sigma}(s_h^\tau,a_h^\tau)$. Let $C_{H,d,\kappa,K} := 36\sqrt{\frac{H^4d^3}{\kappa}\log\left(\frac{(\lambda+K)2KdH^2}{\lambda\delta}\right)} + 12\lambda\frac{H^2\sqrt{d}}{\kappa}$ and denote*

$$
\xi := \sup_{V\in[0,H],\ s'\sim P_h(s,a),\ h\in[H]}\left|\frac{r_h + V\left(s'\right) - \left(\mathcal{T}_hV\right)(s,a)}{\sigma_V(s,a)}\right|.
$$

If $K \geq 4C_{H,d,\kappa,K}^2$ and $K \geq \widetilde{O}(H^6 d/\kappa)$, then with probability $1 - \delta$,

$$\left\| \sum_{\tau=1}^{K} x_\tau \eta_\tau \right\|_{\widehat{\Lambda}^{-1}} \leq 16\sqrt{d \log\left(1 + \frac{K}{\lambda d}\right) \cdot \log\left(\frac{4K^2}{\delta}\right)} + 4\xi \log\left(\frac{4K^2}{\delta}\right) \leq \widetilde{O} \max\left\{\sqrt{d}, \xi\right\},$$

where $\widetilde{O}$ absorbs the constants and Polylog terms.

*Proof of Lemma C.4.* By construction, we have $\|x_\tau\| \leq \|\phi/\widehat{\sigma}\| \leq 1$ and by Lemma C.7, with probability $1 - \delta/3$,

$$\left\| \sigma_{\widehat{V}_{h+1}} - \widehat{\sigma}_h \right\|_\infty = \sup_{s,a} \frac{\left| \sigma_{\widehat{V}_{h+1}}^2(s,a) - \widehat{\sigma}_h^2(s,a) \right|}{\left| \sigma_{\widehat{V}_{h+1}}(s,a) + \widehat{\sigma}_h(s,a) \right|} \leq \frac{1}{2} \left\| \sigma_{\widehat{V}_{h+1}}^2 - \widehat{\sigma}_h^2 \right\|_\infty \leq C_{H,d,\kappa,K} \sqrt{\frac{1}{K}}$$

Therefore, when $K \geq 4C_{H,d,\kappa,K}^2$, $C_{H,d,\kappa,K}\sqrt{\frac{1}{K}} \leq 1/2 \leq \sigma_{\widehat{V}_{h+1}}(s_h^\tau, a_h^\tau)/2$ and hence

$$|\eta_\tau| \leq \left| \frac{r_h^\tau + \widehat{V}_{h+1}\left(s_{h+1}^\tau\right) - \left(\mathcal{T}_h \widehat{V}_{h+1}\right)(s_h^\tau, a_h^\tau)}{\sigma_{\widehat{V}_{h+1}}(s_h^\tau, a_h^\tau) - \frac{C_{H,d,\kappa,K}}{K^{1/2}}} \right| \leq 2 \left| \frac{r_h^\tau + \widehat{V}_{h+1}\left(s_{h+1}^\tau\right) - \left(\mathcal{T}_h \widehat{V}_{h+1}\right)(s_h^\tau, a_h^\tau)}{\sigma_{\widehat{V}_{h+1}}(s_h^\tau, a_h^\tau)} \right|$$

$$\leq 2 \sup_{V \in [0,H], \, s' \sim P_h(s,a)} \left| \frac{r + V(s') - (\mathcal{T}_h V)(s,a)}{\sigma_V(s,a)} \right| := \xi.$$

Next, for a fixed function $V$, we define the Bellman error as $\mathcal{B}_h(V)(s,a) = r_h + V(s') - (\mathcal{T}_h V)(s,a)$, then

$$\mathrm{Var}\left[\eta_\tau | \mathcal{F}_{\tau-1}\right] = \frac{\mathrm{Var}\left[r_h^\tau + \widehat{V}_{h+1}\left(s_{h+1}^\tau\right) - \left(\mathcal{T}_h \widehat{V}_{h+1}\right)(s_h^\tau, a_h^\tau) \Big| \mathcal{F}_{\tau-1}\right]}{\widehat{\sigma}^2(s_h^\tau, a_h^\tau)}$$

$$= \frac{\mathrm{Var}\left[\mathcal{B}_h \widehat{V}_{h+1}(s_h^\tau, a_h^\tau) - \mathcal{B}_h V_{h+1}^\star(s_h^\tau, a_h^\tau) + \mathcal{B}_h V_{h+1}^\star(s_h^\tau, a_h^\tau) \Big| \mathcal{F}_{\tau-1}\right]}{\widehat{\sigma}^2(s_h^\tau, a_h^\tau)}$$

$$\leq \frac{\mathrm{Var}\left[\mathcal{B}_h V_{h+1}^\star(s_h^\tau, a_h^\tau) \big| \mathcal{F}_{\tau-1}\right] + 8H \left\| \mathcal{B}_h \widehat{V}_{h+1} - \mathcal{B}_h V_{h+1}^\star \right\|_\infty}{\widehat{\sigma}^2(s_h^\tau, a_h^\tau)}$$

$$\leq \frac{\mathrm{Var}\left[\mathcal{B}_h V_{h+1}^\star(s_h^\tau, a_h^\tau) \big| \mathcal{F}_{\tau-1}\right] + 16H \left\| \widehat{V}_{h+1} - V_{h+1}^\star \right\|_\infty}{\widehat{\sigma}^2(s_h^\tau, a_h^\tau)}$$

$$\leq \frac{\mathrm{Var}\left[\mathcal{B}_h V_{h+1}^\star(s_h^\tau, a_h^\tau) \big| \mathcal{F}_{\tau-1}\right] + \widetilde{O}(\frac{H^3\sqrt{d}}{\sqrt{\kappa K}})}{\widehat{\sigma}^2(s_h^\tau, a_h^\tau)}$$

$$= \frac{\mathrm{Var}\left[\mathcal{B}_h V_{h+1}^\star(s_h^\tau, a_h^\tau) \big| s_h^\tau, a_h^\tau\right] + \widetilde{O}(\frac{H^3\sqrt{d}}{\sqrt{\kappa K}})}{\widehat{\sigma}^2(s_h^\tau, a_h^\tau)}$$

$$= \frac{\mathrm{Var}_{V_{h+1}^\star}(s_h^\tau, a_h^\tau) + \widetilde{O}(\frac{H^3\sqrt{d}}{\sqrt{\kappa K}})}{\widehat{\sigma}^2(s_h^\tau, a_h^\tau)} \leq \frac{2\mathrm{Var}_{V_{h+1}^\star}(s_h^\tau, a_h^\tau) + \widetilde{O}(\frac{H^3\sqrt{d}}{\sqrt{\kappa K}})}{\sigma^{\star 2}(s_h^\tau, a_h^\tau)} \leq 2 + \frac{\widetilde{O}(\frac{H^3\sqrt{d}}{\sqrt{\kappa K}})}{\sigma^{\star 2}(s_h^\tau, a_h^\tau)}$$

$$\leq \widetilde{O}(1)$$

where the first inequality is by Lemma H.11, the second inequality is by $\mathcal{T}_h$ is non-expansive, the third inequality is by Lemma C.8, the next equality is by Markovian property, and the fourth inequality is by Lemma C.7 and Lemma C.10. The fifth inequality uses definition $\sigma_{h,V}(s,a)^2 := \max\{1, \mathrm{Var}_{P_h}(V)(s,a)\}$ and the last one is by condition $K \geq \widetilde{O}(H^6 d/\kappa)$ and $\sigma_{h,V^\star}(s,a)^2 := \max\{1, \mathrm{Var}_{P_h}(V^\star)(s,a)\} \geq 1$. Thus, by Bernstein inequality for self-normalized martingale

(Lemma H.3),[9] with probability $1 - \delta$,

$$\left\| \sum_{\tau=1}^{K} x_\tau \eta_\tau \right\|_{\widehat{\Lambda}^{-1}} \leq \widetilde{O}\left( \sqrt{d \log\left(1 + \frac{K}{\lambda d}\right) \cdot \log\left(\frac{4K^2}{\delta}\right)} \right) + 4\xi \log\left(\frac{4K^2}{\delta}\right) \leq \widetilde{O} \max\left\{ \sqrt{d}, \xi \right\}$$

where $\widetilde{O}$ absorbs the constants and Polylog terms. $\qquad\square$

Recall $\mathcal{M}_1, \mathcal{M}_2, \mathcal{M}_3, \mathcal{M}_4$ in List A. Based on the above results, we have the following key lemma:

**Lemma C.5.** *Assume* $K > \max\{\mathcal{M}_1, \mathcal{M}_2, \mathcal{M}_3, \mathcal{M}_4\}$, *for any* $0 < \lambda < \kappa$, *suppose* $\sqrt{d} > \xi$, *where* $\xi := \sup_{V \in [0, H], \, s' \sim P_h(s,a), \, h \in [H]} \left| \frac{r_h + V(s') - (\mathcal{T}_h V)(s,a)}{\sigma_V(s,a)} \right|$. *Then with probability* $1 - \delta$, *for all* $h, s, a \in [H] \times \mathcal{S} \times \mathcal{A}$,

$$\left| (\mathcal{T}_h \widehat{V}_{h+1} - \widehat{\mathcal{T}}_h \widehat{V}_{h+1})(s, a) \right| \leq \widetilde{O}\left( \sqrt{d} \sqrt{\phi(s,a) \Lambda_h^{-1} \phi(s,a)} \right) + \frac{2H^3 \sqrt{d}}{K},$$

*where* $\Lambda_h = \sum_{\tau=1}^{K} \phi(s_h^\tau, a_h^\tau)^\top \phi(s_h^\tau, a_h^\tau) / \sigma_{\widehat{V}_{h+1}}^2(s_h^\tau, a_h^\tau) + \lambda I$ *and* $\widetilde{O}$ *absorbs the universal constants and Polylog terms.*

*Proof of Lemma C.5.* Combing equation 9, Lemma C.2, equation 10, Lemma C.3 and C.4 and a union bound to finish the proof. $\qquad\square$

## C.3   Proof of the first part of Theorem 3.2

**Theorem C.6** (First part of Theorem 3.2). *Let* $K$ *be the number of episodes. Suppose* $\sqrt{d} > \xi$, *where* $\xi := \sup_{V \in [0,H], \, s' \sim P_h(s,a), \, h \in [H]} \left| \frac{r_h + V(s') - (\mathcal{T}_h V)(s,a)}{\sigma_V(s,a)} \right|$ *and* $K > \max\{\mathcal{M}_1, \mathcal{M}_2, \mathcal{M}_3, \mathcal{M}_4\}$[10]. *Then for any* $0 < \lambda < \kappa$, *with probability* $1 - \delta$, *for all policy* $\pi$ *simultaneously, the output* $\widehat{\pi}$ *of Algorithm 1 satisfies*

$$v^\pi - v^{\widehat{\pi}} \leq \widetilde{O}\left( \sqrt{d} \cdot \sum_{h=1}^{H} \mathbb{E}_\pi \left[ \left( \phi(\cdot, \cdot)^\top \Lambda_h^{-1} \phi(\cdot, \cdot) \right)^{1/2} \right] \right) + \frac{2H^4 \sqrt{d}}{K}$$

*where* $\Lambda_h = \sum_{\tau=1}^{K} \frac{\phi(s_h^\tau, a_h^\tau) \cdot \phi(s_h^\tau, a_h^\tau)^\top}{\sigma_{\widehat{V}_{h+1}}^2(s_h^\tau, a_h^\tau)} + \lambda I_d$ *and* $\widetilde{O}$ *absorbs the universal constants and the Polylog terms.*

*Proof of Theorem C.6.* Combing Lemma C.1 and Lemma C.5, we directly have with probability $1 - \delta$, for all policy $\pi$ simultaneously,

$$V_1^\pi(s) - V_1^{\widehat{\pi}}(s) \leq \widetilde{O}\left( \sqrt{d} \cdot \sum_{h=1}^{H} \mathbb{E}_\pi \left[ \left( \phi(\cdot, \cdot)^\top \Lambda_h^{-1} \phi(\cdot, \cdot) \right)^{1/2} \Big| s_1 = s \right] \right) + \frac{2H^4 \sqrt{d}}{K}, \qquad (12)$$

now take the initial distribution $d_1$ on both sides to get the stated result. $\qquad\square$

## C.4   Two Intermediate results

The next two lemmas provide intermediate results in finishing the whole proofs.

---

[9]To be rigorous, Lemma H.3 needs to be modified since the absolute value bound and the variance bound here are in the high probability sense. However, this will not affect the validity of the result as the weaker version can also be obtained (see Chung and Lu (2006) and a related discussion in Yin et al. (2021) Remark E.7.) To make the proof more readable, we do not include them here to avoid over-technicality.

[10]The definition of $\mathcal{M}_i$ is in List A.

### C.4.1 BOUNDING THE VARIANCE

**Lemma C.7.** *Recall the definition* $\widehat{\sigma}_h(\cdot,\cdot)^2 = \max\{1, \widehat{\mathrm{Var}}_{P_h}\widehat{V}_{h+1}(\cdot,\cdot)\} + 1$ *and* $\sigma_{\widehat{V}_{h+1}}(\cdot,\cdot)^2 :=$ $\max\{1, \mathrm{Var}_{P_h}\widehat{V}_{h+1}(\cdot,\cdot)\} + 1$. *Moreover,* $[\widehat{\mathrm{Var}}_h\widehat{V}_{h+1}](\cdot,\cdot) = \langle\phi(\cdot,\cdot), \bar{\beta}_h\rangle_{[0,(H-h+1)^2]} - [\langle\phi(\cdot,\cdot), \bar{\theta}_h\rangle_{[0,H-h+1]}]^2$ *(where* $\bar{\beta}_h$ *and* $\bar{\theta}_h$ *are defined in Algorithm 1). Let* $K \geq \max\left\{512(1/\kappa)^2 \log\left(\frac{4Hd}{\delta}\right), 4\lambda/\kappa\right\}$, *then with probability* $1 - \delta$,

$$\sup_h ||\widehat{\sigma}_h^2 - \sigma_{\widehat{V}_{h+1}}^2||_\infty \leq 36\sqrt{\frac{H^4 d^3}{\kappa K}\log\left(\frac{(\lambda+K)2KdH^2}{\lambda\delta}\right)} + 12\lambda\frac{H^2\sqrt{d}}{\kappa K}.$$

*Proof.* **Step1:** we first show for all $h, s, a \in [H] \times \mathcal{S} \times \mathcal{A}$, with probability $1 - \delta$

$$\left|\langle\phi(s,a), \bar{\beta}_h\rangle_{[0,(H-h+1)^2]} - \mathbb{P}_h(\widehat{V}_{h+1})^2(s,a)\right| \leq 12\sqrt{\frac{H^4 d^3}{\kappa K}\log\left(\frac{(\lambda+K)2KdH^2}{\lambda\delta}\right)} + 4\lambda\frac{H^2\sqrt{d}}{\kappa K}.$$

**Proof of Step1.** Note

$$\left|\langle\phi(s,a), \bar{\beta}_h\rangle_{[0,(H-h+1)^2]} - \mathbb{P}_h(\widehat{V}_{h+1})^2(s,a)\right| \leq \left|\langle\phi(s,a), \bar{\beta}_h\rangle - \mathbb{P}_h(\widehat{V}_{h+1})^2(s,a)\right|$$

$$= \left|\phi(s,a)^\top \bar{\Sigma}_h^{-1} \sum_{\tau=1}^K \phi(\bar{s}_h^\tau, \bar{a}_h^\tau) \cdot \widehat{V}_{h+1}(\bar{s}_{h+1}^\tau)^2 - \mathbb{P}_h(\widehat{V}_{h+1})^2(s,a)\right|$$

$$= \left|\phi(s,a)^\top \bar{\Sigma}_h^{-1} \sum_{\tau=1}^K \phi(\bar{s}_h^\tau, \bar{a}_h^\tau) \cdot \widehat{V}_{h+1}(\bar{s}_{h+1}^\tau)^2 - \phi(s,a)^\top \int_{\mathcal{S}} (\widehat{V}_{h+1})^2(s')d\nu_h(s')\right|$$

$$= \left|\phi(s,a)^\top \bar{\Sigma}_h^{-1} \sum_{\tau=1}^K \phi(\bar{s}_h^\tau, \bar{a}_h^\tau) \cdot \widehat{V}_{h+1}(\bar{s}_{h+1}^\tau)^2 - \phi(s,a)^\top \bar{\Sigma}_h^{-1}(\sum_{\tau=1}^K \phi(\bar{s}_h^\tau, \bar{a}_h^\tau)\phi(\bar{s}_h^\tau, \bar{a}_h^\tau)^\top + \lambda I)\int_{\mathcal{S}} (\widehat{V}_{h+1})^2(s')d\nu_h(s')\right|$$

$$\leq \underbrace{\left|\phi(s,a)^\top \bar{\Sigma}_h^{-1} \sum_{\tau=1}^K \phi(\bar{s}_h^\tau, \bar{a}_h^\tau) \cdot \left(\widehat{V}_{h+1}(\bar{s}_{h+1}^\tau)^2 - \mathbb{P}_h(\widehat{V}_{h+1})^2(\bar{s}_h^\tau, \bar{a}_h^\tau)\right)\right|}_{\textcircled{1}} + \underbrace{\lambda\left|\phi(s,a)^\top \bar{\Sigma}_h^{-1} \int_{\mathcal{S}} (\widehat{V}_{h+1})^2(s')d\nu_h(s')\right|}_{\textcircled{2}}$$

For $\textcircled{2}$, since $K \geq \max\left\{512(1/\kappa)^2 \log\left(\frac{4Hd}{\delta}\right), 4\lambda/\kappa\right\}$, by Lemma H.6 and a union bound over $h \in [H]$, with probability $1 - \delta$ for all $h, s, a \in [H] \times \mathcal{S} \times \mathcal{A}$,

$$\textcircled{2} \leq \lambda \|\phi(s,a)\|_{\bar{\Sigma}_h^{-1}} \left\|\int_{\mathcal{S}} (\widehat{V}_{h+1})^2(s')d\nu_h(s')\right\|_{\bar{\Sigma}_h^{-1}}$$

$$\leq \lambda \frac{2}{\sqrt{K}} \|\phi(s,a)\|_{(\Sigma_h^p)^{-1}} \frac{2}{\sqrt{K}} \left\|\int_{\mathcal{S}} (\widehat{V}_{h+1})^2(s')d\nu_h(s')\right\|_{(\Sigma_h^p)^{-1}} \leq 4\lambda\left\|(\Sigma_h^p)^{-1}\right\| \frac{H^2\sqrt{d}}{K} \leq 4\lambda\frac{H^2\sqrt{d}}{\kappa K}. \tag{13}$$

For $\textcircled{1}$, we have

$$\textcircled{1} \leq \|\phi(s,a)\|_{\bar{\Sigma}_h^{-1}} \left\|\sum_{\tau=1}^K \phi(\bar{s}_h^\tau, \bar{a}_h^\tau) \cdot \left(\widehat{V}_{h+1}(\bar{s}_{h+1}^\tau)^2 - \mathbb{P}_h(\widehat{V}_{h+1})^2(\bar{s}_h^\tau, \bar{a}_h^\tau)\right)\right\|_{\bar{\Sigma}_h^{-1}} \tag{14}$$

**Bounding using covering.** Note for any fix $V_{h+1}$, we can define $x_\tau = \phi(\bar{s}_h^\tau, \bar{a}_h^\tau)$ ($\|\phi\|_2 \leq 1$) and $\eta_\tau = V_{h+1}(\bar{s}_{h+1}^\tau)^2 - \mathbb{P}_h(V_{h+1})^2(\bar{s}_h^\tau, \bar{a}_h^\tau)$ is $H^2$-subgaussian, by Lemma H.2 (where $t = K$ and $L = 1$) with probability $1 - \delta$,

$$\left\|\sum_{\tau=1}^K \phi(\bar{s}_h^\tau, \bar{a}_h^\tau) \cdot \left(V_{h+1}(\bar{s}_{h+1}^\tau)^2 - \mathbb{P}_h(V_{h+1})^2(\bar{s}_h^\tau, \bar{a}_h^\tau)\right)\right\|_{\bar{\Sigma}_h^{-1}} \leq \sqrt{8H^4 \cdot \frac{d}{2}\log\left(\frac{\lambda+K}{\lambda\delta}\right)}$$

let $\mathcal{N}_h(\epsilon)$ be the minimal $\epsilon$-cover (with respect the supremum norm) of $\mathcal{V}_h := \{V_h : V_h(\cdot) = \max_{a\in\mathcal{A}}\left\{\min\{\phi(s,a)^\top\theta - C_1\sqrt{d \cdot \phi(\cdot,\cdot)^\top\widehat{\Lambda}_h^{-1}\phi(\cdot,\cdot)} - C_2, H - h + 1\}^+\}\right\}$. That is, for any

$V \in \mathcal{V}_h$, there exists a value function $V' \in \mathcal{N}_h(\epsilon)$ such that $\sup_{s \in \mathcal{S}} |V(s) - V'(s)| < \epsilon$. Now by a union bound, we obtain with probability $1 - \delta$

$$\sup_{V_{h+1} \in \mathcal{N}_{h+1}(\epsilon)} \left\| \sum_{\tau=1}^{K} \phi(\bar{s}_h^\tau, \bar{a}_h^\tau) \cdot \left( V_{h+1}(\bar{s}_{h+1}^\tau)^2 - \mathbb{P}_h(V_{h+1})^2(\bar{s}_h^\tau, \bar{a}_h^\tau) \right) \right\|_{\bar{\Sigma}_h^{-1}} \leq \sqrt{8H^4 \cdot \frac{d}{2} \log\left( \frac{\lambda + K}{\lambda \delta} |\mathcal{N}_{h+1}(\epsilon)| \right)}$$

which implies

$$\left\| \sum_{\tau=1}^{K} \phi(\bar{s}_h^\tau, \bar{a}_h^\tau) \cdot \left( \widehat{V}_{h+1}(\bar{s}_{h+1}^\tau)^2 - \mathbb{P}_h(\widehat{V}_{h+1})^2(\bar{s}_h^\tau, \bar{a}_h^\tau) \right) \right\|_{\bar{\Sigma}_h^{-1}}$$

$$\leq \sqrt{8H^4 \cdot \frac{d}{2} \log\left( \frac{\lambda + K}{\lambda \delta} |\mathcal{N}_{h+1}(\epsilon)| \right)} + 4H^2 \sqrt{\epsilon^2 K^2 / \lambda}$$

choosing $\epsilon = d\sqrt{\lambda}/K$, applying Lemma B.3 of Jin et al. (2021b)[11] to the covering number $\mathcal{N}_{h+1}(\epsilon)$ w.r.t. $\mathcal{V}_{h+1}$, we can further bound above by

$$\leq \sqrt{8H^4 \cdot \frac{d^3}{2} \log\left( \frac{\lambda + K}{\lambda \delta} 2dHK \right)} + 4H^2\sqrt{d^2} \leq 6\sqrt{H^4 \cdot d^3 \log\left( \frac{\lambda + K}{\lambda \delta} 2dHK \right)}$$

Apply a union bound for $h \in [H]$, we have with probability $1 - \delta$, for all $h \in [H]$,

$$\left\| \sum_{\tau=1}^{K} \phi(\bar{s}_h^\tau, \bar{a}_h^\tau) \cdot \left( \widehat{V}_{h+1}(\bar{s}_{h+1}^\tau)^2 - \mathbb{P}_h(\widehat{V}_{h+1})^2(\bar{s}_h^\tau, \bar{a}_h^\tau) \right) \right\|_{\bar{\Sigma}_h^{-1}} \leq 6\sqrt{H^4 d^3 \log\left( \frac{(\lambda + K)2KdH^2}{\lambda \delta} \right)} \tag{15}$$

and similar to ②, with probability $1 - \delta$ for all $h, s, a \in [H] \times \mathcal{S} \times \mathcal{A}$,

$$\|\phi(s,a)\|_{\bar{\Sigma}_h^{-1}} \leq \frac{2 \left\| (\Sigma_h^p)^{-1} \right\|^{1/2}}{\sqrt{K}} \leq \frac{2}{\sqrt{\kappa K}}. \tag{16}$$

Combing equation 13, equation 14, equation 15 and equation 16 we obtain with probability $1 - \delta$ for all $h, s, a \in [H] \times \mathcal{S} \times \mathcal{A}$,

$$\left| \langle \phi(s,a), \bar{\beta}_h \rangle_{[0,(H-h+1)^2]} - \mathbb{P}_h(\widehat{V}_{h+1})^2(s,a) \right| \leq 12\sqrt{\frac{H^4 d^3}{\kappa K} \log\left( \frac{(\lambda + K)2KdH^2}{\lambda \delta} \right)} + 4\lambda \frac{H^2\sqrt{d}}{\kappa K}.$$

**Step2:** we show for all $h, s, a \in [H] \times \mathcal{S} \times \mathcal{A}$, with probability $1 - \delta$

$$\left| \langle \phi(s,a), \bar{\theta}_h \rangle_{[0,H-h+1]} - \mathbb{P}_h(\widehat{V}_{h+1})(s,a) \right| \leq 12\sqrt{\frac{H^2 d^3}{\kappa K} \log\left( \frac{(\lambda + K)2KdH^2}{\lambda \delta} \right)} + 4\lambda \frac{H\sqrt{d}}{\kappa K}. \tag{17}$$

The proof of Step2 follows nearly the identical way as Step1 except $\widehat{V}_h^2$ is replaced by $\widehat{V}_h$.

**Step3:** We prove $\sup_h \|\widehat{\sigma}_h^2 - \sigma_{\widehat{V}_h}^2\|_\infty \leq 36\sqrt{\frac{H^4 d^3}{\kappa K} \log\left( \frac{(\lambda+K)2KdH^2}{\lambda \delta} \right)} + 12\lambda \frac{H^2\sqrt{d}}{\kappa K}$.

**Proof of Step3.** By equation 17,

$$\left| \left[ \langle \phi(\cdot,\cdot), \bar{\theta}_h \rangle_{[0,H-h+1]} \right]^2 - \left[ \mathbb{P}_h(\widehat{V}_{h+1})(s,a) \right]^2 \right|$$

$$= \left| \langle \phi(s,a), \bar{\theta}_h \rangle_{[0,H-h+1]} + \mathbb{P}_h(\widehat{V}_{h+1})(s,a) \right| \cdot \left| \langle \phi(s,a), \bar{\theta}_h \rangle_{[0,H-h+1]} - \mathbb{P}_h(\widehat{V}_{h+1})(s,a) \right|$$

$$\leq 2H \cdot \left| \langle \phi(s,a), \bar{\theta}_h \rangle_{[0,H-h+1]} - \mathbb{P}_h(\widehat{V}_{h+1})(s,a) \right| \leq 24\sqrt{\frac{H^4 d^3}{\kappa K} \log\left( \frac{(\lambda + K)2KdH^2}{\lambda \delta} \right)} + 8\lambda \frac{H^2\sqrt{d}}{\kappa K}.$$

---

[11]Note the same result in Jin et al. (2021b) applies even though we have an extra constant $C_2$.

Combining this with Step1 we receive $\forall h, s, a \in [H] \times \mathcal{S} \times \mathcal{A}$, with probability $1 - \delta$

$$\left| \widehat{\mathrm{Var}}_h \widehat{V}_{h+1}(s,a) - \mathrm{Var}_{P_h} \widehat{V}_{h+1}(s,a) \right| \leq 36 \sqrt{\frac{H^4 d^3}{\kappa K} \log \left( \frac{(\lambda + K) 2K d H^2}{\lambda \delta} \right)} + 12\lambda \frac{H^2 \sqrt{d}}{\kappa K}.$$

Finally, by the non-expansiveness of operator $\max\{1, \cdot\}$, we have the stated result.

$\square$

### C.4.2 A CRUDE BOUND ON $\sup_h \|V_h^\star - \widehat{V}_h\|_\infty$.

**Lemma C.8.** *Define* $\widehat{\sigma}_h(s,a) = \sqrt{\max\left\{1, \widehat{\mathrm{Var}}_{P_h} \widehat{V}_{h+1}(s,a)\right\} + 1}$, *if* $K \geq \max\{\mathcal{M}_1, \mathcal{M}_2, \mathcal{M}_3, \mathcal{M}_4\}$ *and* $K > C \cdot H^4 \kappa^2$, *then with probability at least* $1 - \delta$,

$$\sup_h \left\| V_h^\star - \widehat{V}_h \right\|_\infty \leq \widetilde{O} \left( \frac{H^2 \sqrt{d}}{\sqrt{\kappa K}} \right).$$

*Proof.* **Step1:** We show with probability at least $1 - \delta$, $\sup_h \left\| V_h^\star - V_h^{\widehat{\pi}} \right\|_\infty \leq \widetilde{O} \left( \frac{H^2 \sqrt{d}}{\sqrt{\kappa K}} \right)$.

Indeed, combing Lemma C.1 and Lemma C.5, similar to the proof of Theorem C.6, we directly have with probability $1 - \delta$, for all policy $\pi$ simultaneously, and for all $s \in \mathcal{S}, h \in [H]$

$$V_h^\pi(s) - V_h^{\widehat{\pi}}(s) \leq \widetilde{O} \left( \sqrt{d} \cdot \sum_{t=h}^H \mathbb{E}_\pi \left[ \left( \phi(\cdot, \cdot)^\top \Lambda_t^{-1} \phi(\cdot, \cdot) \right)^{1/2} \Big| s_h = s \right] \right) + \frac{2H^4 \sqrt{d}}{K}, \qquad (18)$$

Next, since $K \geq \max\left\{ 512 (1/\kappa)^2 \log \left( \frac{4Hd}{\delta} \right), 4\lambda/\kappa \right\}$, by Lemma H.6 and a union bound over $h \in [H]$, with probability $1 - \delta$

$$\sup_{s,a} \|\phi(s,a)\|_{\widehat{\Lambda}_h^{-1}} \leq \frac{2}{\sqrt{K}} \sup_{s,a} \|\phi(s,a)\|_{\Lambda_h^{p-1}} \leq \frac{2H}{\sqrt{\kappa K}}, \quad \forall h \in [H].$$

Lastly, taking $\pi = \pi^\star$ in equation 18 to obtain

$$\begin{aligned}
0 \leq V_h^{\pi^\star}(s) - V_h^{\widehat{\pi}}(s) \leq & \widetilde{O} \left( \sqrt{d} \cdot \sum_{t=h}^H \mathbb{E}_{\pi^\star} \left[ \left( \phi(\cdot, \cdot)^\top \Lambda_t^{-1} \phi(\cdot, \cdot) \right)^{1/2} \Big| s_h = s \right] \right) + \frac{2H^4 \sqrt{d}}{K} \\
\leq & \widetilde{O} \left( \frac{H^2 \sqrt{d}}{\sqrt{\kappa K}} \right) + \frac{2H^4 \sqrt{d}}{K}.
\end{aligned} \qquad (19)$$

This implies by using the condition $K > C \cdot H^4 \kappa^2$, we finish the proof of Step1.

**Step2:** We show with probability $1 - \delta$, $\sup_h \left\| \widehat{V}_h - V_h^{\widehat{\pi}} \right\|_\infty \leq \widetilde{O} \left( \frac{H^2 \sqrt{d}}{\sqrt{\kappa K}} \right)$.

Indeed, applying Extended Value Difference Lemma H.7 for $\pi = \pi' = \widehat{\pi}$, then with probability $1 - \delta$, for all $s, h$

$$\begin{aligned}
\left| \widehat{V}_h(s) - V_h^{\widehat{\pi}}(s) \right| &= \left| \sum_{t=h}^H \mathbb{E}_{\widehat{\pi}} \left[ \widehat{Q}_h(s_h, a_h) - \left( \mathcal{T}_h \widehat{V}_{h+1} \right)(s_h, a_h) \Big| s_h = s \right] \right| \\
&\leq \sum_{t=h}^H \left\| (\widehat{\mathcal{T}}_h \widehat{V}_{h+1} - \mathcal{T}_h \widehat{V}_{h+1})(s,a) \right\| + \|\Gamma_h(s,a)\| \\
&\leq \widetilde{O} \left( H\sqrt{d} \left\| \sqrt{\phi(s,a) \Lambda_h^{-1} \phi(s,a)} \right\| \right) + \frac{4H^4 \sqrt{d}}{K} \leq \widetilde{O} \left( \frac{H^2 \sqrt{d}}{\sqrt{\kappa K}} \right)
\end{aligned}$$

where the second inequality uses Lemma C.5[12] and the last inequality follows the same procedure as Step1.

**Step3:** Combine Step1 and Step2, by triangular inequality and a union bound we finish the proof of the lemma.

$\square$

**Remark C.9.** *Note as an intermediate calculation, equation 19 ensures a learning bound with order* $\widetilde{O}(\frac{H^2\sqrt{d}}{\sqrt{\kappa K}})$. *Here, the convergence rate is the standard statistical rate* $\frac{1}{\sqrt{K}}$ *and the* $H^2$ *dependence is loose. However, the feature dependence* $\sqrt{d/\kappa}$ *is roughly tight, since, in the well-explored case (Assumption 2 of Wang et al. (2021a)),* $\kappa = 1/d$ *and the* $\sqrt{d/\kappa} = \sqrt{d^2}$ *recovers the optimal feature dependence* $dH\sqrt{T}$ *in the online setting (Zhou et al., 2021a). If* $\kappa \ll 1/d$, *then doing offline learning requires sample size proportional to* $d/\kappa$, *which reveals offline RL is harder when the exploration of behavior policy is insufficient. When* $\kappa = 0$, *learning the optimal policy accurately cannot be guaranteed even if the sample/episode size* $K \to \infty$.

## C.5 PROOF OF THE SECOND PART OF THEOREM 3.2

**Lemma C.10.** *Recall* $\widehat{\sigma}_h = \sqrt{\max\left\{1, \widehat{\operatorname{Var}}_{P_h}\widehat{V}_{h+1}\right\} + 1}$ *and* $\sigma_h^\star = \sqrt{\max\left\{1, \operatorname{Var}_{P_h}V_{h+1}^\star\right\} + 1}$.
*Let* $K \geq \max\left\{512(1/\kappa)^2 \log\left(\frac{4Hd}{\delta}\right), 4\lambda/\kappa\right\}$ *and* $K \geq \max\{\mathcal{M}_1, \mathcal{M}_2, \mathcal{M}_3, \mathcal{M}_4\}$, *then with probability* $1 - \delta$,

$$\sup_h \|\widehat{\sigma}_h^2 - \sigma_h^{\star 2}\|_\infty \leq \widetilde{O}\left(\frac{H^3\sqrt{d}}{\sqrt{\kappa K}}\right).$$

*Proof.* By definition and the non-expansiveness of $\max\{1, \cdot\} + 1$, we have

$$\left\|\sigma_{\widehat{V}_{h+1}}^2 - \sigma_h^{\star 2}\right\|_\infty \leq \left\|\operatorname{Var}\widehat{V}_{h+1} - \operatorname{Var}V_{h+1}^\star\right\|_\infty$$
$$\leq \left\|\mathbb{P}_h\left(\widehat{V}_{h+1}^2 - V_{h+1}^{\star 2}\right)\right\|_\infty + \left\|(\mathbb{P}_h\widehat{V}_{h+1})^2 - (\mathbb{P}_h V_{h+1}^\star)^2\right\|_\infty$$
$$\leq \left\|\widehat{V}_{h+1}^2 - V_{h+1}^{\star 2}\right\|_\infty + \left\|(\mathbb{P}_h\widehat{V}_{h+1} + \mathbb{P}_h V_{h+1}^\star)(\mathbb{P}_h\widehat{V}_{h+1} - \mathbb{P}_h V_{h+1}^\star)\right\|_\infty$$
$$\leq 2H\left\|\widehat{V}_{h+1} - V_{h+1}^\star\right\|_\infty + 2H\left\|\mathbb{P}_h\widehat{V}_{h+1} - \mathbb{P}_h V_{h+1}^\star\right\|_\infty \leq \widetilde{O}\left(\frac{H^3\sqrt{d}}{\sqrt{\kappa K}}\right).$$

with probability $1 - \delta$ for all $h \in [H]$, where the last inequality comes from Lemma C.8. Combining this with Lemma C.7, we have the stated result. $\square$

**Lemma C.11.** *Denote the quantities* $C_1 = \max\{2\lambda, 128\log(2d/\delta), 128H^4\log(2d/\delta)/\kappa^2\}$ *and* $C_2 = \max\{\frac{\lambda^2}{\kappa\log((\lambda+K)H/\lambda\delta)}, 96^2 H^{12}d\log((\lambda + K)H/\lambda\delta)/\kappa^5\}$. *Suppose the number of episode* $K$ *satisfies* $K > \max\{C_1, C_2\}$, *then with probability* $1 - \delta$,

$$\sqrt{\phi(s,a)\Lambda_h^{-1}\phi(s,a)} \leq 2\sqrt{\phi(s,a)\Lambda_h^{\star-1}\phi(s,a)}, \quad \forall s, a \in \mathcal{S} \times \mathcal{A},$$

*Proof of Lemma C.11.* By definition $\sqrt{\phi(s,a)\Lambda_h^{-1}\phi(s,a)} = \|\phi(s,a)\|_{\Lambda_h^{-1}}$. Then denote

$$\Lambda_h' = \frac{1}{K}\Lambda_h, \quad \Lambda_h^{\star'} = \frac{1}{K}\Lambda_h^\star,$$

---

[12]To be absolutely rigorous, we cannot directly apply Lemma C.5 here since the crude bound has already been used in Lemma C.4. However, this can be resolved completely by first deriving an even cruder bound for $\sup_h \|V_h^\star - \widehat{V}_h\|_\infty$ that has $1/\sqrt{K}$ rate without using Lemma C.5 (which we call it Lemma C.8*), and we can use Lemma C.8* to show a similar result Lemma C.5*. Finally, we can use Lemma C.5* here to finish the proof of this Lemma C.8. However, we avoid explicitly doing this to prevent over-technicality.

where $\Lambda_h = \sum_{\tau=1}^{K} \phi(s_h^\tau, a_h^\tau)^\top \phi(s_h^\tau, a_h^\tau)/\sigma_{V_{h+1}^\star}^2(s_h^\tau, a_h^\tau) + \lambda I$. Under the condition of $K$, by Lemma C.10, with probability $1 - \delta$

$$
\begin{aligned}
\left\| \Lambda_h^{\star'} - \Lambda_h' \right\| &\leq \sup_{s,a} \left\| \frac{\phi(s,a)\phi(s,a)^\top}{\sigma_h^{\star 2}(s,a)} - \frac{\phi(s,a)\phi(s,a)^\top}{\sigma_{\widehat{V}_{h+1}}^2(s,a)} \right\| \\
&\leq \sup_{s,a} \left| \frac{\sigma_h^{\star 2}(s,a) - \sigma_{\widehat{V}_{h+1}}^2(s,a)}{\sigma_h^{\star 2}(s,a)\sigma_{\widehat{V}_{h+1}}^2(s,a)} \right| \cdot \|\phi(s,a)\|^2 \leq \sup_{s,a} \left| \frac{\sigma_h^{\star 2}(s,a) - \sigma_{\widehat{V}_{h+1}}^2(s,a)}{1} \right| \cdot 1 \qquad (20) \\
&\leq \widetilde{O}\left( \frac{H^3\sqrt{d}}{\sqrt{\kappa K}} \right).
\end{aligned}
$$

Next by Lemma H.5 (with $\phi$ to be $\phi/\sigma_{V_{h+1}^\star}$ and $C = 1$), it holds with probability $1 - \delta$,

$$
\left\| \Lambda_h^{\star'} - \left( \mathbb{E}_{\mu,h}[\phi(s,a)\phi(s,a)^\top/\sigma_{V_{h+1}^\star}^2(s,a)] + \frac{\lambda}{K}I_d \right) \right\| \leq \frac{4\sqrt{2}}{\sqrt{K}}\left( \log \frac{2d}{\delta} \right)^{1/2}.
$$

Therefore by *Weyl's spectrum theorem* and the condition $K > \max\{2\lambda, 128\log(2d/\delta), 128H^4\log(2d/\delta)/\kappa^2\}$, the above implies

$$
\begin{aligned}
\left\| \Lambda_h^{\star'} \right\| &= \lambda_{\max}(\Lambda_h^{\star'}) \leq \lambda_{\max}\left( \mathbb{E}_{\mu,h}[\phi(s,a)\phi(s,a)^\top/\sigma_{V_{h+1}^\star}^2(s,a)] \right) + \frac{\lambda}{K} + \frac{4\sqrt{2}}{\sqrt{K}}\left( \log \frac{2d}{\delta} \right)^{1/2} \\
&\leq \left\| \mathbb{E}_{\mu,h}[\phi(s,a)\phi(s,a)^\top/\sigma_{V_{h+1}^\star}^2(s,a)] \right\| + \frac{\lambda}{K} + \frac{4\sqrt{2}}{\sqrt{K}}\left( \log \frac{2d}{\delta} \right)^{1/2} \\
&\leq \|\phi(s,a)\|^2 + \frac{\lambda}{K} + \frac{4\sqrt{2}}{\sqrt{K}}\left( \log \frac{2d}{\delta} \right)^{1/2} \leq 1 + \frac{\lambda}{K} + \frac{4\sqrt{2}}{\sqrt{K}}\left( \log \frac{2d}{\delta} \right)^{1/2} \leq 2, \\
\lambda_{\min}(\Lambda_h^{\star'}) &\geq \lambda_{\min}\left( \mathbb{E}_{\mu,h}[\phi(s,a)\phi(s,a)^\top/\sigma_{V_{h+1}^\star}^2(s,a)] \right) + \frac{\lambda}{K} - \frac{4\sqrt{2}}{\sqrt{K}}\left( \log \frac{2d}{\delta} \right)^{1/2} \\
&\geq \lambda_{\min}\left( \mathbb{E}_{\mu,h}[\phi(s,a)\phi(s,a)^\top/\sigma_{V_{h+1}^\star}^2(s,a)] \right) - \frac{4\sqrt{2}}{\sqrt{K}}\left( \log \frac{2d}{\delta} \right)^{1/2} \\
&\geq \frac{\kappa}{H^2} - \frac{4\sqrt{2}}{\sqrt{K}}\left( \log \frac{2d}{\delta} \right)^{1/2} \geq \frac{\kappa}{2H^2}.
\end{aligned}
$$

Hence with probability $1 - \delta$, $\left\| \Lambda_h^{\star'} \right\| \leq 2$ and $\left\| \Lambda_h^{\star'-1} \right\| = 1/\lambda_{\min}(\Lambda_h^{\star'}) \leq 2H^2/\kappa$. Similarly, $\left\| \Lambda_h'^{-1} \right\| \leq 2H^2/\kappa$ with high probability.

Now apply Lemma H.4 to $\Lambda_h^{\star'}$ and $\Lambda_h'$ and a union bound, we obtain with probability $1 - \delta$, for all $s, a$

$$
\begin{aligned}
\|\phi(s,a)\|_{\Lambda_h'^{-1}} &\leq \left[ 1 + \sqrt{\left\| \Lambda_h^{\star'-1} \right\| \left\| \Lambda_h^{\star'} \right\| \cdot \left\| \Lambda_h'^{-1} \right\| \cdot \left\| \Lambda_h^{\star'} - \Lambda_h' \right\|} \right] \cdot \|\phi(s,a)\|_{\Lambda_h^{\star'-1}} \\
&\leq \left[ 1 + \sqrt{\frac{2H^2}{\kappa} \cdot 1 \cdot \frac{2H^2}{\kappa} \cdot \left\| \Lambda_h^{\star'} - \Lambda_h' \right\|} \right] \cdot \|\phi(s,a)\|_{\Lambda_h^{\star'-1}} \\
&\leq \left[ 1 + \sqrt{\frac{H^4}{\kappa^2}\left[ \widetilde{O}\left( \frac{H^3\sqrt{d}}{\sqrt{\kappa K}} \right) \right]} \right] \cdot \|\phi(s,a)\|_{\Lambda_h^{\star'-1}} \leq 2\|\phi(s,a)\|_{\Lambda_h^{\star'-1}}
\end{aligned}
$$

where the third inequality uses equation 20 and the last inequality uses $K > \max\{\frac{\lambda^2}{\kappa\log((\lambda+K)H/\lambda\delta)}, 96^2 H^{12}d\log((\lambda + K)H/\lambda\delta)/\kappa^5\}$. The claimed result follows straightforwardly by multiplying $1/\sqrt{K}$ on both sides of the above.

$\square$

*Proof of Theorem 3.2.* The first part of the theorem has been shown in Theorem C.6. For the second part, apply Theorem C.6 with $\pi = \pi^\star$, then with probability $1 - \delta$,

$$v^{\pi^\star} - v^{\widehat{\pi}} \leq \widetilde{O}\left(\sqrt{d} \cdot \sum_{h=1}^{H} \mathbb{E}_{\pi^\star}\left[\left(\phi(\cdot, \cdot)^\top \Lambda_h^{-1} \phi(\cdot, \cdot)\right)^{1/2}\right]\right) + \frac{2H^4\sqrt{d}}{K},$$

Now apply Lemma C.11 and a union bound, with probability $1 - \delta$,

$$0 \leq v^\star - v^{\widehat{\pi}} \leq \widetilde{O}\left(\sqrt{d} \cdot \sum_{h=1}^{H} \mathbb{E}_{\pi^\star}\left[\left(\phi(\cdot, \cdot)^\top \Lambda_h^{\star-1} \phi(\cdot, \cdot)\right)^{1/2}\right]\right) + \frac{2H^4\sqrt{d}}{K}.$$

$\square$

# D  PROOF OF THEOREM 3.3

First of all, we show the following lemma.

**Lemma D.1.** *Suppose* $K > \max\{\mathcal{M}_1, \mathcal{M}_2, \mathcal{M}_3, \mathcal{M}_4\}$*. Plug*

$$\Gamma_h^I(s, a) \leftarrow \phi(s, a)^\top \left|\widehat{\Lambda}_h^{-1} \sum_{\tau=1}^{K} \frac{\phi(s_h^\tau, a_h^\tau) \cdot \left(r_h^\tau + \widehat{V}_{h+1}(s_{h+1}^\tau) - \left(\widehat{\mathcal{T}}_h \widehat{V}_{h+1}\right)(s_h^\tau, a_h^\tau)\right)}{\widehat{\sigma}_h^2(s_h^\tau, a_h^\tau)}\right| + \widetilde{O}\left(\frac{H^3 d/\kappa}{K}\right)$$

*in Algorithm 1 and let $\mathcal{T}_h$ be the Bellman operator and $\widehat{\mathcal{T}}_h$ be the approximated Bellman operator. Then we have with probability $1 - \delta$:*

$$|(\mathcal{T}_h \widehat{V}_{h+1} - \widehat{\mathcal{T}}_h \widehat{V}_{h+1})(s, a)| \leq \Gamma_h^I(s, a), \quad \forall s, a \in \mathcal{S} \times \mathcal{A}.$$

*Proof of Lemma D.1.* Suppose $w_h$ is the coefficient corresponding to the $\mathcal{T}_h \widehat{V}_{h+1}$ (such $w_h$ exists by Lemma H.9), *i.e.* $\mathcal{T}_h \widehat{V}_{h+1} = \phi^\top w_h$, and recall $(\widehat{\mathcal{T}}_h \widehat{V}_{h+1})(s, a) = \phi(s, a)^\top \widehat{w}_h$, then:

$$\left(\mathcal{T}_h \widehat{V}_{h+1}\right)(s, a) - \left(\widehat{\mathcal{T}}_h \widehat{V}_{h+1}\right)(s, a) = \phi(s, a)^\top (w_h - \widehat{w}_h)$$

$$= \phi(s, a)^\top w_h - \phi(s, a)^\top \widehat{\Lambda}_h^{-1}\left(\sum_{\tau=1}^{K} \phi(s_h^\tau, a_h^\tau) \cdot \left(r_h^\tau + \widehat{V}_{h+1}(s_{h+1}^\tau)\right)/\widehat{\sigma}_h^2(s_h^\tau, a_h^\tau)\right)$$

$$= \underbrace{\phi(s, a)^\top w_h - \phi(s, a)^\top \widehat{\Lambda}_h^{-1}\left(\sum_{\tau=1}^{K} \phi(s_h^\tau, a_h^\tau) \cdot \left(\mathcal{T}_h \widehat{V}_{h+1}\right)(s_h^\tau, a_h^\tau)/\widehat{\sigma}_h^2(s_h^\tau, a_h^\tau)\right)}_{(i)}$$

$$+ \underbrace{\phi(s, a)^\top \widehat{\Lambda}_h^{-1}\left(\sum_{\tau=1}^{K} \phi(s_h^\tau, a_h^\tau) \cdot \left(r_h^\tau + \widehat{V}_{h+1}(s_{h+1}^\tau) - \left(\widehat{\mathcal{T}}_h \widehat{V}_{h+1}\right)(s_h^\tau, a_h^\tau)\right)/\widehat{\sigma}_h^2(s_h^\tau, a_h^\tau)\right)}_{(ii)}$$

$$+ \underbrace{\phi(s, a)^\top \widehat{\Lambda}_h^{-1}\left(\sum_{\tau=1}^{K} \phi(s_h^\tau, a_h^\tau) \cdot \left(\left(\widehat{\mathcal{T}}_h \widehat{V}_{h+1}\right)(s_h^\tau, a_h^\tau) - \left(\mathcal{T}_h \widehat{V}_{h+1}\right)(s_h^\tau, a_h^\tau)\right)/\widehat{\sigma}_h^2(s_h^\tau, a_h^\tau)\right)}_{(iii)}$$

$$\tag{21}$$

For term (i), by Lemma C.2 it is bounded by $\frac{2\lambda H^3 \sqrt{d}/\kappa}{K}$ with probability $1 - \delta/2$.[13]

For term (ii), it is bounded by

$$\phi(s, a)^\top \left|\widehat{\Lambda}_h^{-1} \sum_{\tau=1}^{K} \frac{\phi(s_h^\tau, a_h^\tau) \cdot \left(r_h^\tau + \widehat{V}_{h+1}(s_{h+1}^\tau) - \left(\widehat{\mathcal{T}}_h \widehat{V}_{h+1}\right)(s_h^\tau, a_h^\tau)\right)}{\widehat{\sigma}_h^2(s_h^\tau, a_h^\tau)}\right|.$$

---

[13]Note Here Lemma C.2 still applies even if the $\Gamma_h$ changes since it works for all $\widehat{V}_h \in [0, H]$ so that $\|w_h\|_2 \leq 2H\sqrt{d}$ and the truncation (Line 13 in Algorithm 1) guarantees this.

For term (iii), by Cauchy inequality

$$\phi(s,a)^\top \widehat{\Lambda}_h^{-1} \left( \sum_{\tau=1}^K \phi\left(s_h^\tau, a_h^\tau\right) \cdot \left(\left(\widehat{\mathcal{T}}_h \widehat{V}_{h+1}\right)\left(s_h^\tau, a_h^\tau\right) - \left(\mathcal{T}_h \widehat{V}_{h+1}\right)\left(s_h^\tau, a_h^\tau\right)\right) / \widehat{\sigma}_h^2(s_h^\tau, a_h^\tau) \right)$$

$$\leq \|\phi(s,a)\|_{\widehat{\Lambda}_h^{-1}} \cdot \left\| \sum_{\tau=1}^K \phi\left(s_h^\tau, a_h^\tau\right) \cdot \left(\left(\widehat{\mathcal{T}}_h \widehat{V}_{h+1}\right)\left(s_h^\tau, a_h^\tau\right) - \left(\mathcal{T}_h \widehat{V}_{h+1}\right)\left(s_h^\tau, a_h^\tau\right)\right) / \widehat{\sigma}_h^2(s_h^\tau, a_h^\tau) \right\|_{\widehat{\Lambda}_h^{-1}}$$

$$\leq \frac{2H}{\sqrt{\kappa K}} \cdot \left\| \sum_{\tau=1}^K \phi\left(s_h^\tau, a_h^\tau\right) \cdot \left(\left(\widehat{\mathcal{T}}_h \widehat{V}_{h+1}\right)\left(s_h^\tau, a_h^\tau\right) - \left(\mathcal{T}_h \widehat{V}_{h+1}\right)\left(s_h^\tau, a_h^\tau\right)\right) / \widehat{\sigma}_h^2(s_h^\tau, a_h^\tau) \right\|_{\widehat{\Lambda}_h^{-1}}$$

$$\leq \frac{2H}{\sqrt{\kappa K}} \cdot \widetilde{O}\left(\frac{H^2 \sqrt{d/\kappa}}{\sqrt{K}}\right) \cdot \sqrt{d} = \widetilde{O}\left(\frac{H^3 d/\kappa}{K}\right)$$

where the first inequality is by Lemma H.6 (with $\phi' = \phi/\widehat{\sigma}_h$ and $\|\phi/\widehat{\sigma}_h\| \leq \|\phi\| \leq 1 := C$) and the third inequality uses $\sqrt{a^\top \cdot A \cdot a} \leq \sqrt{\|a\|_2 \|A\|_2 \|a\|_2} = \|a\|_2 \sqrt{\|A\|_2}$ with $a$ to be either $\phi$ or $w_h$. Moreover, $\lambda_{\min}(\tilde{\Lambda}_h^p) \geq \kappa/\max_{h,s,a} \widehat{\sigma}_h(s,a)^2 \geq \kappa/H^2$ implies $\left\|(\tilde{\Lambda}_h^p)^{-1}\right\| \leq H^2/\kappa$.

The second inequality is true by denoting $x_\tau = \phi(s_h^\tau, a_h^\tau)/\widehat{\sigma}(s_h^\tau, a_h^\tau)$ and

$$\eta_\tau = \left(\left(\widehat{\mathcal{T}}_h \widehat{V}_{h+1}\right)\left(s_h^\tau, a_h^\tau\right) - \left(\mathcal{T}_h \widehat{V}_{h+1}\right)\left(s_h^\tau, a_h^\tau\right)\right) / \widehat{\sigma}_h(s_h^\tau, a_h^\tau)$$

and use Lemma H.10 as the condition for applying Lemma H.2. By collecting those three terms together we have the result. □

## D.1 PROOF OF THEOREM 3.3

*Proof.* Use Lemma D.1 as the condition for Lemma C.1 and average over initial distribution $d_1$, we obtain with probability $1 - \delta$,

$$v^\pi - v^{\widehat{\pi}} \leq$$

$$\sum_{h=1}^H \mathbb{E}_{\pi_h} [\phi(s,a)]^\top \left| \widehat{\Lambda}_h^{-1} \sum_{\tau=1}^K \frac{\phi\left(s_h^\tau, a_h^\tau\right) \cdot \left(r_h^\tau + \widehat{V}_{h+1}\left(s_{h+1}^\tau\right) - \left(\widehat{\mathcal{T}}_h \widehat{V}_{h+1}\right)\left(s_h^\tau, a_h^\tau\right)\right)}{\widehat{\sigma}_h^2(s_h^\tau, a_h^\tau)} \right| + \widetilde{O}\left(\frac{H^4 d/\kappa}{K}\right)$$

$$\tag{22}$$

Denote $A_h := \sum_{\tau=1}^K \frac{\phi(s_h^\tau, a_h^\tau) \cdot \left(r_h^\tau + \widehat{V}_{h+1}\left(s_{h+1}^\tau\right) - (\mathcal{T}_h \widehat{V}_{h+1})(s_h^\tau, a_h^\tau)\right)}{\widehat{\sigma}_h^2(s_h^\tau, a_h^\tau)}$, then

$$\mathbb{E}_{\pi_h} [\phi(s,a)]^\top \left| \widehat{\Lambda}_h^{-1} \sum_{\tau=1}^K \frac{\phi\left(s_h^\tau, a_h^\tau\right) \cdot \left(r_h^\tau + \widehat{V}_{h+1}\left(s_{h+1}^\tau\right) - \left(\widehat{\mathcal{T}}_h \widehat{V}_{h+1}\right)\left(s_h^\tau, a_h^\tau\right)\right)}{\widehat{\sigma}_h^2(s_h^\tau, a_h^\tau)} \right|$$

$$\leq \mathbb{E}_{\pi_h} [\phi]^\top \cdot \left| \widehat{\Lambda}_h^{-1} A_h \right| + \mathbb{E}_{\pi_h} [\phi]^\top \left| \widehat{\Lambda}_h^{-1} \sum_{\tau=1}^K \frac{\phi\left(s_h^\tau, a_h^\tau\right) \cdot \left(\mathcal{T}_h \widehat{V}_{h+1}\left(s_h^\tau, a_h^\tau\right) - \widehat{\mathcal{T}}_h \widehat{V}_{h+1}\left(s_h^\tau, a_h^\tau\right)\right)}{\widehat{\sigma}_h^2(s_h^\tau, a_h^\tau)} \right|$$

For the second term, it can be bounded similar to term (iii) in Lemma D.1 and for the first term we have the following:

$$\mathbb{E}_{\pi_h} [\phi]^\top \cdot \left| \widehat{\Lambda}_h^{-1} A_h \right| = \mathbb{E}_{\pi_h} [\phi]^\top \cdot \widehat{\Lambda}_h^{-1} \cdot \widehat{\Lambda}_h \left| \widehat{\Lambda}_h^{-1} A_h \right| \leq \|\mathbb{E}_{\pi_h}[\phi]\|_{\widehat{\Lambda}_h^{-1}} \cdot \left\| \widehat{\Lambda}_h |\widehat{\Lambda}_h^{-1} A_h| \right\|_{\widehat{\Lambda}_h^{-1}}$$

$$\leq \|\mathbb{E}_{\pi_h}[\phi]\|_{\widehat{\Lambda}_h^{-1}} \cdot \||A_h\||_{\widehat{\Lambda}_h^{-1}} \leq \widetilde{O}(\sqrt{d}) \|\mathbb{E}_{\pi_h}[\phi]\|_{\widehat{\Lambda}_h^{-1}} \leq \widetilde{O}(\sqrt{d} \|\mathbb{E}_{\pi_h}[\phi]\|_{\Lambda_h^{-1}}),$$

where the first inequality uses Cauchy's inequality, the second inequality uses $\widehat{\Lambda}_h$ is coordinate-wise positive (since we assume here $\phi \geq 0$), the third inequality is identical to the analysis in Section C.2.2 and the fourth inequality is identical to the analysis in Section C.2.1 with $\phi$ replaced by $\mathbb{E}[\phi]$. Plug this back to equation 22 we finish the proof for the first part. For the second part, converting $\Lambda_h^{-1}$ to $\Lambda_h^{\star -1}$ is identical to Section C.5. This finishes the proof. □

# E    PROOF OF MINIMAX LOWER BOUND THEOREM 3.5

The proof follows the lower bound proof of Zanette et al. (2021). For completeness, we provide all the details in below.

## E.1    CONSTRUCTION

Similar to the proof of [Zanette et al. (2021), Theorem 2], we construct a family of MDPs, each parameterized by a Boolean vector $u = (u_1, \ldots, u_H)$ with each $u_h \in \{-1, +1\}^{d-2}$ for $h \in [H]$. The MDPs share the same transition kernel and are only different in the reward observations.

**State space:**  At each time step $h$, there are two states $\mathcal{S} = \{+1, -1\}$.

**Action space:**  The action space $\mathcal{A} = \{-1, 0, +1\}^{d-2}$.

**Feature map:**  The feature map $\phi : \mathcal{S} \times \mathcal{A} \mapsto \mathbb{R}^d$ is given by

$$\phi(+1, a) = \begin{pmatrix} \frac{a}{\sqrt{2d}} \\ \frac{1}{\sqrt{2}} \\ 0 \end{pmatrix} \in \mathbb{R}^d, \qquad \phi(-1, a) = \begin{pmatrix} \frac{a}{\sqrt{2d}} \\ 0 \\ \frac{1}{\sqrt{2}} \end{pmatrix} \in \mathbb{R}^d.$$

The construction ensures the condition $\|\phi(s, a)\|_2 \leq 1$ for any $(s, a) \in \mathcal{S} \times \mathcal{A}$.

**Transition kernel:**  The transition probability $P_h(s' \mid s, a)$ is independent of action $a$. In other words, the Markov decision process reduces to a homogeneous Markov chain with transition matrix

$$\mathbf{P} = \begin{pmatrix} \frac{1}{2} & \frac{1}{2} \\ \frac{1}{2} & \frac{1}{2} \end{pmatrix} \in \mathbb{R}^2.$$

By letting

$$\nu_h(+1) = \nu_h(-1) = \begin{pmatrix} \mathbf{0}_{d-2} \\ \frac{1}{\sqrt{2}} \\ \frac{1}{\sqrt{2}} \end{pmatrix} \in \mathbb{R}^d,$$

we have $P_h(s' \mid s, a) = \langle \phi(s, a), \nu_h(s') \rangle$ to be a valid probability transition.

**Reward observations:**  For any MDP $M_u$, at each times step $h$, the reward follows a Gaussian distribution with

$$R_{u,h}(s, a) \sim \mathcal{N}\left( \frac{s}{\sqrt{6}} + \frac{\delta}{\sqrt{2d}} \langle a, u_h \rangle, 1 \right),$$

where $\delta \in \left[0, \frac{1}{\sqrt{3d}}\right]$ determines to what extent the MDP models are different from each other. The mean reward function satisfies $r_{u,h}(s, a) = \langle \phi(s, a), \theta_{u,h} \rangle$ with

$$\theta_{u,h} = \begin{pmatrix} \delta u_h \\ \frac{1}{\sqrt{3}} \\ -\frac{1}{\sqrt{3}} \end{pmatrix} \in \mathbb{R}^d.$$

**Offline data collection Scheme:**  The dataset $\mathcal{D} = \{(s_h^\tau, a_h^\tau, r_h^\tau, s_{h+1}^\tau)\}_{\tau \in [K]}^{h \in [H]}$ consist of $K$ i.i.d. trajectories. All the trajectories initiate from uniform distribution. We take a behavior policy $\mu(\cdot \mid s)$ that is independent of state $s$. Let $\{e_1, e_2, \ldots, e_{d-2}\}$ be the canonical bases of $\mathbb{R}^{d-2}$ and $\mathbf{0}_{d-2} \in \mathbb{R}^{d-2}$ be the zero vector. The behavior policy $\mu$ is set as

$$\mu(e_j \mid s) = \frac{1}{d} \quad \text{for any } j \in [d-2] \qquad \text{and} \qquad \mu(\mathbf{0}_{d-2} \mid s) = \frac{2}{d}.$$

## E.2    OVERVIEW OF PROOF

The proof of the theorem is based on Assouad's method, where we first reduce the problem to binary hypothesis tests (Lemmas E.1 and E.2) and then connect the testing error to the uncertainty quantity in the upper bound (Lemma E.3).

**Lemma E.1** (Reduction to testing). *There exists a universal constant $c_1 > 0$ such thata*

$$\inf_{\widehat{\pi}} \max_{u \in \mathcal{U}} \mathbb{E}_u\left[V_u^\star - V_u^{\widehat{\pi}}\right] \geq c_1\,\delta\sqrt{d}\,H \min_{u,u' \in \mathcal{U}: D_H(u';u)=1} \inf_{\psi}\left[\mathbb{P}_u(\psi \neq u) + \mathbb{P}_{u'}(\psi \neq u')\right], \quad (23)$$

*where $\widehat{\pi}$ denotes the output of any algorithm that maps from observations to an estimated policy. $\psi$ is any test function for parameter $u$ and $D_H$ is the hamming distance.*

**Lemma E.2.** *There exists a universal constant $c_2 > 0$ such that when taking $\delta := \frac{c_2\,d}{\sqrt{K}}$, we have*

$$\min_{u,u' \in \mathcal{U}: D_H(u';u)=1} \inf_{\psi}\left[\mathbb{P}_u(\psi \neq u) + \mathbb{P}_{u'}(\psi \neq u')\right] \geq \frac{1}{2}. \quad (24)$$

When $K \gtrsim d^3$, $\delta := \frac{c_2\,d}{\sqrt{K}}$ ensures that $\delta \leq 1/\sqrt{3d}$. Combining Lemmas E.1 and E.2 yields a lower bound

$$\inf_{\widehat{\pi}} \max_{u \in \mathcal{U}} \mathbb{E}_u\left[V_u^\star - V_u^{\widehat{\pi}}\right] \geq c\,\frac{d\sqrt{d}H}{\sqrt{K}}, \quad (25)$$

where $c > 0$ is a universal constant. We then use the following Lemma E.3 to connect the above lower bound to the uncertainty term $\sqrt{d} \cdot \sum_{h=1}^{H} \sqrt{\mathbb{E}_{\pi^\star}[\phi]^\top (\Lambda_h^\star)^{-1}\mathbb{E}_{\pi^\star}[\phi]}$ for the chosen linear MDP instances class $\mathcal{M}$.

**Lemma E.3.** *There exists a universal constant $c_3 > 0$ such that for all $M \in \mathcal{M}$,*

$$\sum_{h=1}^{H} \sqrt{\mathbb{E}_{\pi^\star}[\phi]^\top (\Lambda_h^\star)^{-1}\mathbb{E}_{\pi^\star}[\phi]} \leq c_3\,\frac{d\,H}{\sqrt{K}}. \quad (26)$$

Plugging inequality equation E.3 into the bound equation 25, we obtain the minimax lower bound equation 6 in the statement of theorem.

### E.3    REDUCTION TO TESTING VIA ASSOUAD'S METHOD

*Proof of Lemma E.1.* For any index vector $u = (u_1, \ldots, u_H) \in \mathcal{U} = \{-1, +1\}^{(d-2) \times H}$, the optimal policy for MDP instance $M_u$ is simply

$$\pi_h^\star(\cdot) = u_h \qquad \text{for } h \in [H].$$

Similar to the proof of Lemma 9 in Zanette et al. (2021), we can show that the value suboptimality of policy $\pi$ on MDP $M_u$ is given by

$$V_u^\star - V_u^\pi = \frac{\delta}{\sqrt{2d}} \sum_{h=1}^{H} \left\|u_h - \mathbb{E}_\pi[a_h]\right\|_1.$$

Define $u^\pi = (u_1^\pi, \ldots, u_H^\pi)$ with $u_h^\pi := \text{sign}\left(\mathbb{E}_\pi[a_h]\right)$, then the $\ell_1$-norm is lower bounded as

$$\left\|u_h - \mathbb{E}_\pi[a_h]\right\|_1 \geq D_H(u_h^\pi; u_h),$$

where $D_H(\cdot; \cdot)$ denotes the Hamming distance. It follows that

$$V_u^\star - V_u^\pi \geq \frac{\delta}{\sqrt{2d}} D_H(u^\pi; u). \quad (27)$$

We then apply Assouad's method (Lemma 2.12 in Sampson and Guttorp (1992)) and obtain that

$$\inf_{\hat{u} \in \mathcal{U}} \max_{u \in \mathcal{U}} \mathbb{E}_u\left[D_H(\hat{u}; u)\right] \geq \frac{(d-2)H}{2} \min_{u,u' \in \mathcal{U}: D_H(u';u)=1} \inf_{\psi}\left[\mathbb{P}_u(\psi \neq u) + \mathbb{P}_{u'}(\psi \neq u')\right], \quad (28)$$

where $\psi$ is any test functions mapping from observations to $\{u, u'\}$. Combining inequalities equation 27 and equation 28, we finish the proof. $\qquad\square$

### E.4 LOWER BOUND ON THE TESTING ERROR

*Proof of Lemma E.2.* The proof of Lemma E.2 is similar to that of Lemma 10 in Zanette et al. (2021). We first apply Theorem 2.12 in Sampson and Guttorp (1992) to lower bound the testing error using Kullback–Leibler divergence and obtain

$$
\min_{u,u'\in\mathcal{U}:D_H(u';u)=1}\ \inf_{\psi}\ \big[\mathbb{P}_u(\psi\neq u)+\mathbb{P}_{u'}(\psi\neq u')\big]\geq 1-\Big(\tfrac{1}{2}\max_{u,u'\in\mathcal{U}:D_H(u';u)=1}D_{\mathrm{KL}}(\mathcal{Q}_u\|\mathcal{Q}_{u'})\Big)^{1/2}.
\tag{29}
$$

It only remains to estimate $D_{\mathrm{KL}}(\mathcal{Q}_u\|\mathcal{Q}_{u'})$.

The probability density $\mathcal{Q}_u$ takes the form

$$
\mathcal{Q}_u(\mathcal{D})=\prod_{k=1}^{K}\xi_1(s_1^k)\prod_{h=1}^{H}\mu\big(a_h^k\mid s_h^k\big)\big[R_{u,h}(s_h^k,a_h^k)\big](r_h^k)\,\mathbb{P}_h(s_{h+1}^k\mid s_h^k,a_h^k)
$$

where $\xi_1=\big[\tfrac{1}{2},\tfrac{1}{2}\big]$ is the initial distribution. It follows that

$$
\begin{aligned}
D_{KL}(\mathcal{Q}_u\|\mathcal{Q}_{u'}) &= \mathbb{E}_u\big[\log(\mathcal{Q}_u/\mathcal{Q}_{u'})\big]\\
&= K\cdot\sum_{h=1}^{H}\mathbb{E}_u\Big[\log\big(\big[R_{u,h}(s_h^1,a_h^1)\big](r_h^1)/\big[R_{u',h}(s_h^1,a_h^1)\big](r_h^1)\big)\Big]\\
&= \frac{K}{d}\sum_{j=1}^{d-2}D_{\mathrm{KL}}\Big(\mathcal{N}\big(\tfrac{\delta}{\sqrt{2d}}\langle e_j,u_h\rangle,1\big)\,\Big\|\,\mathcal{N}\big(\tfrac{\delta}{\sqrt{2d}}\langle e_j,u_h'\rangle,1\big)\Big).
\end{aligned}
$$

If we take $\delta=\frac{c_2\,d}{\sqrt{K}}$, then inequality equation 29 ensures inequality equation 24, as claimed in the statement of the lemma.

$\square$

### E.5 CONNECTION TO THE UNCERTAINTY TERM

*Proof of Lemma E.3.* We first calculate the explicit form of the inverse of variance-rescaled covariance matrix $\Lambda_h^{\star,p}$. For each time step $h\in[H]$, the value function $V_{u,h+1}^{\star}$ takes the form

$$
V_{u,h+1}^{\star}=\mathbb{E}_{\pi^{\star}}r_{u,h+1}+\big(\mathbb{P}_{h+1}^{\pi^{\star}}V_{u,h+2}^{\star}\big).
$$

Since $\big(\mathbb{P}_{h+1}V_{u,h+2}^{\star}\big)(+1)=\big(\mathbb{P}_{h+1}V_{u,h+2}^{\star}\big)(-1)$ and $r_{u,h+1}(+1,a)-r_{u,h+1}(-1,a)=2/\sqrt{6}$, we have

$$
\mathrm{Var}_{P_h}(V_{u,h+1}^{\star})(+1,a)=\mathrm{Var}_{P_h}(\mathbb{E}_{\pi^{\star}}r_{u,h+1})(+1,a)=\frac{1}{6}.
$$

Similarly,

$$
\mathrm{Var}_{P_h}(V_{u,h+1}^{\star})(-1,a)=\mathrm{Var}_{P_h}(V_{u,h+1}^{\star})(+1,a)=\frac{1}{6}.
$$

By routine calculation, we find that the population-level rescaled covariance matrix takes the form

$$
\Lambda_h^{\star,p}=\frac{3K}{2}\begin{pmatrix}\frac{2}{d^2}\mathbf{I}_{d-2} & \frac{1}{d\sqrt{d}}\mathbf{1}_{(d-2)\times 2}\\ \frac{1}{d\sqrt{d}}\mathbf{1}_{2\times(d-2)} & \mathbf{I}_2\end{pmatrix}\in\mathbb{R}^{d\times d}
$$

for any $h\in[H]$. Applying Gaussian elimination on $\Lambda_h^{\star,p}$, we have

$$
(\Lambda_h^{\star,p})^{-1}=\frac{2}{3K}\begin{pmatrix}\frac{d^2}{2}\big\{\mathbf{I}_{d-2}+\frac{1}{d-2}\mathbf{1}_{(d-2)\times(d-2)}\big\} & -\frac{d\sqrt{d}}{2(d-2)}\mathbf{1}_{(d-2)\times 2}\\ -\frac{d\sqrt{d}}{2(d-2)}\mathbf{1}_{2\times(d-2)} & \frac{1}{d-2}\begin{pmatrix}d-1 & 1\\ 1 & d-1\end{pmatrix}\end{pmatrix}.
$$

For each time step $h \in [H]$, we have (by Jensen's inequality)

$$\sqrt{\mathbb{E}_{\pi^\star}[\phi]^\top (\Lambda_h^\star)^{-1} \mathbb{E}_{\pi^\star}[\phi]} \leq \frac{1}{2} \big\|\phi(+1, u_h)\big\|_{(\Lambda_h^{\star,P})^{-1}} + \frac{1}{2} \big\|\phi(-1, u_h)\big\|_{(\Lambda_h^{\star,P})^{-1}}.$$

Recall that by our construction,

$$\phi(+1, u_h) = \begin{pmatrix} \frac{u_h}{\sqrt{2d}} \\ \frac{1}{\sqrt{2}} \\ 0 \end{pmatrix} \in \mathbb{R}^d, \qquad \phi(-1, u_h) = \begin{pmatrix} \frac{u_h}{\sqrt{2d}} \\ 0 \\ \frac{1}{\sqrt{2}} \end{pmatrix} \in \mathbb{R}^d.$$

It follows that

$$\big\|\phi(+1, u_h)\big\|^2_{(\Lambda_h^{\star,P})^{-1}} = \big\|\phi(-1, u_h)\big\|^2_{(\Lambda_h^{\star,P})^{-1}}$$

$$= \frac{2}{3K}\left\{ \frac{d}{4} u_h^\top \big\{ \mathbf{I}_{d-2} + \tfrac{1}{d-2}\mathbf{1}_{(d-2)\times(d-2)} \big\} u_h - \frac{d}{2(d-2)} \mathbf{1}_{d-2}^\top u_h + \frac{d-1}{2(d-2)} \right\}$$

$$= \frac{2}{3K}\left\{ \frac{d^2}{4} + \frac{d}{4(d-2)}\big(1 - \mathbf{1}_{d-2}^\top u_h\big)^2 + \frac{1}{4} \right\}$$

$$\leq \frac{2}{3K}\left\{ \frac{d^2}{4} + \frac{d(d-1)^2}{4(d-2)} + \frac{1}{4} \right\} = \frac{2}{3K}\left\{ \frac{d^2}{2} + \frac{d-1}{2(d-2)} \right\} \lesssim d^2/K.$$

Therefore,

$$\sqrt{\mathbb{E}_{\pi^\star}[\phi]^\top (\Lambda_h^\star)^{-1} \mathbb{E}_{\pi^\star}[\phi]} \lesssim d/\sqrt{K}.$$

Taking the summation over $h \in [H]$, we obtain the bound equation 26 as claimed in the lemma statement.

$\square$

# F  A NUMERICAL SIMULATION

## F.1  A LINEAR MDP CONSTRUCTION

We consider a synthetic linear MDP example that is similar to Min et al. (2021) but with some modifications for the offline learning task. The MDP instance we use consists of $|\mathcal{S}| = 2$ states and $|\mathcal{A}| = 100$ actions, and feature dimension $d = 100$. We set $\mathcal{S} = \{0, 1\}$ and $\mathcal{A} = \{0, 1, \ldots, 99\}$ respectively. For each action $a \in \{0, 1, \ldots, 99\}$, we use binary encoding to obtain a vector $\mathbf{a} \in \mathbb{R}^8$ using its binary representation (*i.e.* each coordinate is either $0$ or $1$). we interchangebly use $a$ and and its vector representation $\mathbf{a}$ for the ease of explanation. We first define $\delta(s, a) = \begin{cases} 1 & \text{if } \mathbf{1}\{s = 0\} = \mathbf{1}\{a = 0\} \\ 0 & \text{otherwise} \end{cases}$, then the non-stationary linear MDP is specified by the following configuration

- Feature mapping:
$$\phi(s, a) = \big(\mathbf{a}^\top, \delta(s, a), 1 - \delta(s, a)\big)^\top \in \mathbb{R}^{10}$$

- The true measure $\nu_h$
$$\boldsymbol{\nu}_h(s) = (0, \ldots, 0, (1 - s) \oplus \alpha_h, s \oplus \alpha_h),$$
where $\{\alpha_h\}_{h \in [H]}$ is a sequence of integers taking values $0$ or $1$ and $\oplus$ is the standard XOR operator. We define
$$\theta_h \equiv (0, \ldots, 0, r, 1 - r) \in \mathbb{R}^{10}$$
with the choice of $r = 0.9$. The transition follows $P_h(s'|s, a) = \langle \phi(s, a), \nu_h(s') \rangle$ and the mean reward function $r_h(s, a) = \langle \phi(s, a), \theta_h \rangle$.

- Behavior policy: always choose action $a = 0$ with probability $p$, and other actions uniformly with probability $(1 - p)/99$. The initial distribution chooses $s = 0$ and $s = 1$ with equal probability $1/2$. We use $p = 0.6$.

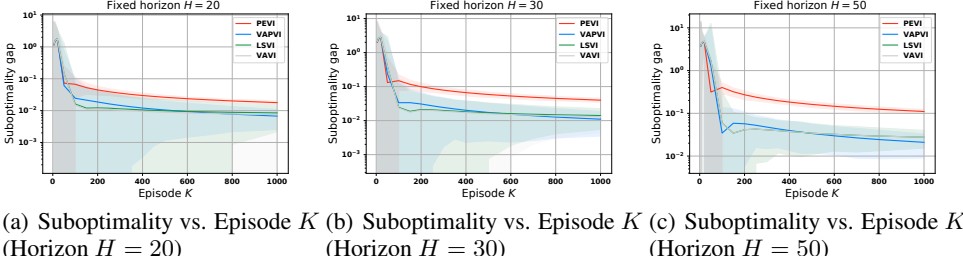

(a) Suboptimality vs. Episode $K$  (b) Suboptimality vs. Episode $K$  (c) Suboptimality vs. Episode $K$
(Horizon $H = 20$)          (Horizon $H = 30$)          (Horizon $H = 50$)

Figure 1: Comparison between PEVI and VAPVI in the non-stationary linear MDP instance described above. In each figure, y-axis denotes suboptimality gap $v^\star - v^{\widehat{\pi}}$, x-axis denotes number of episodes $K$. The problem horizons are fixed to be $H = 20, 30, 50$. The solid line denotes the average suboptimality gap over 50 trials and the error bar area is the corresponding standard deviation. The range of $K$ is from 5 to 1000.

## F.2 EMPIRICAL COMPARISON BETWEEN PEVI AND VAPVI ON THE CONSTRUCTED LINEAR MDP

We compare *Pessimistic Value Iteration* (PEVI) in Jin et al. (2021b) and our VAPVI Algorithm 1 in Figure 1, with horizon to be $H = 20, 30, 50$. In addition, we add the non-pessimistic version for both algorithms, *i.e.* least-square value iteration (LSVI) and variance-aware value iteration (VAVI). The true optimal value $v^\star$ is computed via value iteration using the underlying transition kernels. For the empirical validation of VAPVI, we do not split the data and, in particular, in all the methods we choose $\lambda = 0.01$ (instead of $\lambda = 1$ used in theory (Jin et al., 2021b) which causes over-regularization in the simulation).

We can observe VAPVI outperforms PEVI and the gap becomes larger when horizon $H$ increases. One main reason for this to happen is due to the bonus used in PEVI (Jin et al., 2021b)

$$O\left[dH \cdot \left(\phi(\cdot,\cdot)^\top \Sigma_h^{-1} \phi(\cdot,\cdot)\right)^{1/2}\right]$$

is overly pessimistic comparing to our

$$O\left[\sqrt{d} \cdot \left(\phi(\cdot,\cdot)^\top \Lambda_h^{-1} \phi(\cdot,\cdot)\right)^{1/2}\right]$$

when $H$ becomes larger and this could potentially make the learning less accurate. In addition, both non-pessimistic algorithms exhibit similar accuracy, and this is partially owing to our truncation scheme $\widehat{\sigma}_h(\cdot,\cdot)^2 \leftarrow \max\{1, \widehat{\mathrm{Var}}_{P_h} \widehat{V}_{h+1}(\cdot,\cdot)\}$ so $\widehat{\sigma}_h(\cdot,\cdot)^2$ will just be 1 when the estimated variance is small. Lastly, variance-aware pessimism eventually outperforms non-pessimism algorithms when sample size is large and this might come from that the pessimistic bonus is estimated more accurately when more samples are collected.

## G  SOME MISSING DERIVATIONS AND DISCUSSIONS

### G.1  REGARDING COVERAGE ASSUMPTION

Now we discuss the feature coverage assumption. Indeed, even if Assumption 2.2 is not satisfied, we can still learn in the effective subspan of $\Sigma_h^p := \mathbb{E}_{\mu,h}\left[\phi(s,a)\phi(s,a)^\top\right]$. Concretely, since $\Sigma_h^p$ is symmetric, by orthogonal decomposition we have $\Sigma_h^p = Z_h \Lambda Z_h^\top$, where $Z_h$ (can be estimated using the samples for practical purpose) consists of orthogonal basis and $\Lambda$ consists of eigenvalues of $\Sigma_h^p$ in the diagonal. Suppose we do not have a full coverage, i.e.

$$\Lambda = \mathrm{diag}[\lambda_1, \lambda_2, ..., \lambda_{d'}, 0, ..., 0] \quad \text{with} \quad d' < d,$$

then we can create transformed features $\phi_h'(s,a) = Z_h \cdot \phi_h(s,a)$, and then

$$\mathbb{E}_{\mu,h}\left[\phi_h'(s,a)\phi_h'(s,a)^\top\right] = \Lambda = \mathrm{diag}[\lambda_1, \lambda_2, ..., \lambda_{d'}, 0, ..., 0].$$

Then we can do learning w.r.t. the truncated features $\phi_h'|_{1:d'}$'s instead of the original $\phi$. It reduces to the weaker notion of $\kappa := \min_{h \in [H]}\{\kappa_h : s.t. \ \kappa_h = \text{smallest positive eigenvalue at time } h\}$.

## G.2 DERIVATION OF EQUATION 5

When reducing Theorem 3.2,3.3 to the tabular case, set $\phi(s,a) = \mathbf{1}_{s,a}$, $d = SA$, $\lambda = 0$, and recall by Assumption 3.4 (let's assume $\pi^\star$ is a deterministic policy as it always exists in tabular MDP) $C^\star := \sup_{h,s,a} d_h^{\pi^\star}(s,a)/d_h^\mu(s,a)$, then for Theorem 3.2

$$\sqrt{d} \cdot \sum_{h=1}^{H} \mathbb{E}_{\pi^\star}\left[\sqrt{\phi(\cdot,\cdot)^\top \Lambda_h^{\star-1} \phi(\cdot,\cdot)}\right] = \sqrt{d} \cdot \sum_{h=1}^{H} \sum_{s,a} d_h^{\pi^\star}(s,a)\sqrt{\mathbf{1}_{s,a}^\top \Lambda_h^{\star-1} \mathbf{1}_{s,a}}$$

$$= \sqrt{SA} \cdot \sum_{h=1}^{H} \sum_{s,a} d_h^{\pi^\star}(s,a)\sqrt{\mathbf{1}_{s,a}^\top \mathrm{diag}\left\{\frac{\mathrm{Var}_{P_{\cdot,\cdot}}(V_{h+1}^\star)}{n_{h,\cdot,\cdot}}\right\}\mathbf{1}_{s,a}}$$

$$= \sqrt{SA} \cdot \sum_{h=1}^{H} \sum_{s,a} d_h^{\pi^\star}(s,a)\sqrt{\frac{\mathrm{Var}_{P_{s,a}}(V_{h+1}^\star)}{n_{h,s,a}}} \quad n_{h,s,a} := \sum_{\tau=1}^{K}\mathbf{1}[s_h^\tau, a_h^\tau = s,a]$$

$$\lesssim \sqrt{SA} \cdot \sum_{h=1}^{H} \sum_{s,a} d_h^{\pi^\star}(s,a)\sqrt{\frac{\mathrm{Var}_{P_{s,a}}(V_{h+1}^\star)}{K \cdot d_h^\mu(s,a)}} \leq \sqrt{SAC^\star/K} \cdot \sum_{h=1}^{H} \sum_{s,a} \sqrt{d_h^{\pi^\star}(s,a)\mathrm{Var}_{P_{s,a}}(V_{h+1}^\star)}$$

$$= \sqrt{SAC^\star/K} \cdot \sum_{h=1}^{H} \sum_{s} \sqrt{d_h^{\pi^\star}(s,\pi^\star(s))\mathrm{Var}_{P_{s,\pi^\star(s)}}(V_{h+1}^\star)}$$

$$\leq \sqrt{SAC^\star/K} \cdot \sum_{h=1}^{H} \sqrt{S \cdot \sum_{s} d_h^{\pi^\star}(s,\pi^\star(s))\mathrm{Var}_{P_{s,\pi^\star(s)}}(V_{h+1}^\star)}$$

$$\leq \sqrt{S^2AC^\star/K} \cdot \sqrt{H \sum_{h=1}^{H} \sum_{s} d_h^{\pi^\star}(s,\pi^\star(s))\mathrm{Var}_{P_{s,\pi^\star(s)}}(V_{h+1}^\star)}$$

$$= \sqrt{S^2AC^\star/K} \cdot \sqrt{H \cdot \sum_{h=1}^{H} \mathbb{E}_{\pi_h^\star}[\mathrm{Var}_{P_{(\cdot,\cdot)}}(V_{h+1}^\star)]} \leq \sqrt{H^3 S^2 AC^\star/K}$$

where the first inequality is by Chernoff bound and the last one is by Lemma 3.4. of Yin and Wang (2020) (Law of total variances). The rest of them are from Cauchy's inequality. Similarly, for Theorem 3.3, we also have

$$\sqrt{d} \cdot \sum_{h=1}^{H} \sqrt{\mathbb{E}_{\pi^\star}[\phi]^\top \Lambda_h^{\star-1} \mathbb{E}_{\pi^\star}[\phi]} = \sqrt{d} \cdot \sum_{h=1}^{H} \sqrt{\mathrm{Vec}\{d^{\pi^\star}\}\Lambda_h^{\star-1}\mathrm{Vec}\{d^{\pi^\star}\}}$$

$$= \sqrt{d} \cdot \sum_{h=1}^{H} \sqrt{\mathrm{Vec}\{d^{\pi^\star}\}\mathrm{diag}\left\{\frac{\mathrm{Var}_{P_{\cdot,\cdot}}(V_{h+1}^\star)}{n_{h,\cdot,\cdot}}\right\}\mathrm{Vec}\{d^{\pi^\star}\}}$$

$$= \sqrt{SA} \cdot \sum_{h=1}^{H} \sqrt{\sum_{s,a} d_h^{\pi^\star}(s,a)^2 \frac{\mathrm{Var}_{P_{s,a}}(V_{h+1}^\star)}{n_{h,s,a}}}$$

$$\lesssim \sqrt{SA} \cdot \sum_{h=1}^{H} \sqrt{\sum_{s,a} d_h^{\pi^\star}(s,a)^2 \frac{\mathrm{Var}_{P_{s,a}}(V_{h+1}^\star)}{K \cdot d_h^\mu(s,a)}}$$

$$\leq \sqrt{SAC^\star/K} \cdot \sum_{h=1}^{H} \sqrt{\sum_{s,a} d_h^{\pi^\star}(s,a)\mathrm{Var}_{P_{s,a}}(V_{h+1}^\star)}$$

$$= \sqrt{SAC^\star/K} \cdot \sum_{h=1}^{H} \sqrt{\sum_{s} d_h^{\pi^\star}(s,\pi^\star(s))\mathrm{Var}_{P_{s,\pi^\star(s)}}(V_{h+1}^\star)}$$

$$\leq \sqrt{SAC^\star/K} \cdot \sqrt{H \cdot \sum_{h=1}^{H} \mathbb{E}_{\pi_h^\star}[\mathrm{Var}_{P_{(\cdot,\cdot)}}(V_{h+1}^\star)]} \leq \sqrt{H^3 SAC^\star/K}.$$

# H AUXILIARY LEMMAS

**Lemma H.1** (Matrix McDiarmid inequality / Matrix Chernoff bound (Tropp, 2012)). *Let $z_k$, $k = 1, \ldots, K$ be independent random vectors in $\mathbb{R}^d$, and let $H$ be a mapping that maps $K$ vectors to a $d \times d$ symmetric matrix. Assume there exists a sequence of fixed symmetric matrices $\{A_k\}_{k \in [K]}$ such that for $z_k, z_k'$ ranges over all possible values for each $k \in [K]$, it holds*

$$(H(z_1, \ldots, z_k, \ldots, z_K) - H(z_1, \ldots, z_k', \ldots, z_K))^2 \preceq A_k^2.$$

*Define $\sigma^2 := \left\| \sum_k A_k^2 \right\|$. Then for any $t > 0$,*

$$\mathbb{P}\left\{ \|H(z_1, \ldots, z_K) - \mathbb{E}H(z_1, \ldots, z_K)\| \geq t \right\} \leq d \cdot \exp\left(\frac{-t^2}{8\sigma^2}\right)$$

**Lemma H.2** (Hoeffding inequality for self-normalized martingales (Abbasi-Yadkori et al., 2011)). *Let $\{\eta_t\}_{t=1}^\infty$ be a real-valued stochastic process. Let $\{\mathcal{F}_t\}_{t=0}^\infty$ be a filtration, such that $\eta_t$ is $\mathcal{F}_t$-measurable. Assume $\eta_t$ also satisfies $\eta_t$ given $\mathcal{F}_{t-1}$ is zero-mean and $R$-subgaussian, i.e.*

$$\forall \lambda \in \mathbb{R}, \quad \mathbb{E}\left[e^{\lambda \eta_t} \mid \mathcal{F}_{t-1}\right] \leq e^{\lambda^2 R^2/2}$$

*Let $\{x_t\}_{t=1}^\infty$ be an $\mathbb{R}^d$-valued stochastic process where $x_t$ is $\mathcal{F}_{t-1}$ measurable and $\|x_t\| \leq L$. Let $\Lambda_t = \lambda I_d + \sum_{s=1}^t x_s x_s^\top$. Then for any $\delta > 0$, with probability $1 - \delta$, for all $t > 0$,*

$$\left\| \sum_{s=1}^t x_s \eta_s \right\|_{\Lambda_t^{-1}}^2 \leq 8R^2 \cdot \frac{d}{2} \log\left(\frac{\lambda + tL}{\lambda \delta}\right).$$

**Lemma H.3** (Bernstein inequality for self-normalized martingales (Zhou et al., 2021a)). *Let $\{\eta_t\}_{t=1}^\infty$ be a real-valued stochastic process. Let $\{\mathcal{F}_t\}_{t=0}^\infty$ be a filtration, such that $\eta_t$ is $\mathcal{F}_t$-measurable. Assume $\eta_t$ also satisfies*

$$|\eta_t| \leq R, \mathbb{E}\left[\eta_t \mid \mathcal{F}_{t-1}\right] = 0, \mathbb{E}\left[\eta_t^2 \mid \mathcal{F}_{t-1}\right] \leq \sigma^2.$$

*Let $\{x_t\}_{t=1}^\infty$ be an $\mathbb{R}^d$-valued stochastic process where $x_t$ is $\mathcal{F}_{t-1}$ measurable and $\|x_t\| \leq L$. Let $\Lambda_t = \lambda I_d + \sum_{s=1}^t x_s x_s^\top$. Then for any $\delta > 0$, with probability $1 - \delta$, for all $t > 0$,*

$$\left\| \sum_{s=1}^t \mathbf{x}_s \eta_s \right\|_{\mathbf{\Lambda}_t^{-1}} \leq 8\sigma \sqrt{d \log\left(1 + \frac{tL^2}{\lambda d}\right) \cdot \log\left(\frac{4t^2}{\delta}\right)} + 4R \log\left(\frac{4t^2}{\delta}\right)$$

**Lemma H.4** (Converting the variance under the matrix norm). *Let $\Lambda_1$ and $\Lambda_2 \in \mathbb{R}^{d \times d}$ are two positive semi-definite matrices. Then:*

$$\left\|\Lambda_1^{-1}\right\| \leq \left\|\Lambda_2^{-1}\right\| + \left\|\Lambda_1^{-1}\right\| \cdot \left\|\Lambda_2^{-1}\right\| \cdot \|\Lambda_1 - \Lambda_2\|$$

*and*

$$\|\phi\|_{\Lambda_1^{-1}} \leq \left[1 + \sqrt{\left\|\Lambda_2^{-1}\right\| \|\Lambda_2\| \cdot \left\|\Lambda_1^{-1}\right\| \cdot \|\Lambda_1 - \Lambda_2\|}\right] \cdot \|\phi\|_{\Lambda_2^{-1}}.$$

*for all $\phi \in \mathbb{R}^d$.*

*Proof.* For the first part, note

$$\left\|\Lambda_1^{-1}\right\| \leq \left\|\Lambda_2^{-1}\right\| + \left\|\Lambda_1^{-1} - \Lambda_2^{-1}\right\| \leq \left\|\Lambda_2^{-1}\right\| + \left\|\Lambda_2^{-1}\right\| \|\Lambda_1 - \Lambda_2\| \left\|\Lambda_1^{-1}\right\|$$

For the second one,

$$\|\phi\|_{\Lambda_1^{-1}} = \sqrt{\phi^\top \Lambda_1^{-1} \phi} = \sqrt{\phi^\top \left(\Lambda_1^{-1} - \Lambda_2^{-1}\right) \phi + \phi^\top \Lambda_2^{-1} \phi}$$

$$= \sqrt{\phi^\top \Lambda_2^{-1/2} \left(\Lambda_2^{1/2} \Lambda_1^{-1} \Lambda_2^{1/2} - I + I\right) \Lambda_2^{-1/2} \phi} \leq \sqrt{\|\phi\|_{\Lambda_2^{-1}} \cdot \left(1 + \left\|\Lambda_2^{1/2} \Lambda_1^{-1} \Lambda_2^{1/2} - I\right\|\right) \|\phi\|_{\Lambda_2^{-1}}}$$

$$\leq \left(1 + \left\|\Lambda_2^{1/2} \Lambda_1^{-1} \Lambda_2^{1/2} - I\right\|^{1/2}\right) \cdot \|\phi\|_{\Lambda_2^{-1}} = \left(1 + \left\|\Lambda_2^{1/2} \Lambda_1^{-1} \left(\Lambda_2 - \Lambda_1\right) \Lambda_2^{-1} \Lambda_2^{1/2}\right\|^{1/2}\right) \cdot \|\phi\|_{\Lambda_2^{-1}}$$

$$\leq \left(1 + \sqrt{\|\Lambda_2\| \left\|\Lambda_1^{-1}\right\| \left\|\Lambda_2^{-1}\right\| \|\Lambda_1 - \Lambda_2\|}\right) \cdot \|\phi\|_{\Lambda_2^{-1}}$$

$\square$

**Lemma H.5** (Lemma H.4 of Min et al. (2021)). *let $\phi : \mathcal{S} \times \mathcal{A} \to \mathbb{R}^d$ satisfies $\|\phi(s,a)\| \leq C$ for all $s, a \in \mathcal{S} \times \mathcal{A}$. For any $K > 0, \lambda > 0$, define $\bar{G}_K = \sum_{k=1}^{K} \phi(s_k, a_k)\phi(s_k, a_k)^\top + \lambda I_d$ where $(s_k, a_k)$'s are i.i.d samples from some distribution $\nu$. Then with probability $1 - \delta$,*

$$\left\| \frac{\bar{G}_K}{K} - \mathbb{E}_\nu \left[ \frac{\bar{G}_K}{K} \right] \right\| \leq \frac{4\sqrt{2}C^2}{\sqrt{K}} \left( \log \frac{2d}{\delta} \right)^{1/2} .$$

*Proof of Lemma H.5.* For completeness, we provide the proof of Lemma H.5. Let $x_k = \phi(s_k, a_k)$. Denote $\widetilde{\Sigma}_h$ as the matrix obtained by replacing the k-th vector $x_k$ in $\widehat{\Sigma}_h$ by $\widetilde{x}_k$ and leaving the rest $K - 1$ vectors unchanged. Then

$$\left( \frac{\widehat{\Sigma}_h}{K} - \frac{\widetilde{\Sigma}_h}{K} \right)^2 = \left( \frac{x_k x_k^\top - \widetilde{x}_k \widetilde{x}_k^\top}{K} \right) \preceq \frac{1}{K^2} \left( 2 x_k x_k^\top x_k x_k^\top + 2 \widetilde{x}_k \widetilde{x}_k^\top \widetilde{x}_k \widetilde{x}_k^\top \right) \preceq \frac{4C^4}{K^2} I_d := A_k^2.$$

Notice that $\left\| \sum_k^K A_k^2 \right\| = \frac{4C^4}{K}$, by Lemma H.1 we have the result.

$\square$

**Lemma H.6** (Lemma H.5. of Min et al. (2021)). *Let $\phi : \mathcal{S} \times \mathcal{A} \to \mathbb{R}^d$ be a bounded function s.t. $\|\phi\|_2 \leq C$. Define $\bar{G}_K = \sum_{k=1}^{K} \phi(s_k, a_k)\phi(s_k, a_k)^\top + \lambda I_d$ where $(s_k, a_k)$'s are i.i.d samples from some distribution $\nu$. Let $G = \mathbb{E}_\nu[\phi(s,a)\phi(s,a)^\top]$. Then for any $\delta \in (0, 1)$, if $K$ satisfies*

$$K \geq \max \left\{ 512 C^4 \left\| \mathbf{G}^{-1} \right\|^2 \log \left( \frac{2d}{\delta} \right), 4\lambda \left\| \mathbf{G}^{-1} \right\| \right\} .$$

*Then with probability at least $1 - \delta$, it holds simultaneously for all $u \in \mathbb{R}^d$ that*

$$\|u\|_{\bar{G}_K^{-1}} \leq \frac{2}{\sqrt{K}} \|u\|_{G^{-1}} .$$

**Lemma H.7** (Extended Value Difference (Section B.1 in Cai et al. (2020))). *Let $\pi = \{\pi_h\}_{h=1}^{H}$ and $\pi' = \{\pi'_h\}_{h=1}^{H}$ be two arbitrary policies and let $\{\widehat{Q}_h\}_{h=1}^{H}$ be any given Q-functions. Then define $\widehat{V}_h(s) := \langle \widehat{Q}_h(s, \cdot), \pi_h(\cdot \mid s) \rangle$ for all $s \in \mathcal{S}$. Then for all $s \in \mathcal{S}$,*

$$\widehat{V}_1(s) - V_1^{\pi'}(s) = \sum_{h=1}^{H} \mathbb{E}_{\pi'} \left[ \langle \widehat{Q}_h(s_h, \cdot), \pi_h(\cdot \mid s_h) - \pi'_h(\cdot \mid s_h) \rangle \mid s_1 = s \right] \tag{30}$$
$$+ \sum_{h=1}^{H} \mathbb{E}_{\pi'} \left[ \widehat{Q}_h(s_h, a_h) - \left( \mathcal{T}_h \widehat{V}_{h+1} \right)(s_h, a_h) \mid s_1 = s \right]$$

*where $(\mathcal{T}_h V)(\cdot, \cdot) := r_h(\cdot, \cdot) + (P_h V)(\cdot, \cdot)$ for any $V \in \mathbb{R}^S$.*

*Proof.* Denote $\xi_h = \widehat{Q}_h - \mathcal{T}_h \widehat{V}_{h+1}$. For any $h \in [H]$, we have

$$\begin{aligned}
\widehat{V}_h - V_h^{\pi'} &= \langle \widehat{Q}_h, \pi_h \rangle - \langle Q_h^{\pi'}, \pi'_h \rangle \\
&= \langle \widehat{Q}_h, \pi_h - \pi'_h \rangle + \langle \widehat{Q}_h - Q_h^{\pi'}, \pi'_h \rangle \\
&= \langle \widehat{Q}_h, \pi_h - \pi'_h \rangle + \langle P_h(\widehat{V}_{h+1} - V_{h+1}^{\pi'}) + \xi_h, \pi'_h \rangle \\
&= \langle \widehat{Q}_h, \pi_h - \pi'_h \rangle + \langle P_h(\widehat{V}_{h+1} - V_{h+1}^{\pi'}), \pi'_h \rangle + \langle \xi_h, \pi'_h \rangle
\end{aligned}$$

recursively apply the above for $\widehat{V}_{h+1} - V_{h+1}^{\pi'}$ and use the $\mathbb{E}_{\pi'}$ notation (instead of the inner product of $P_h, \pi'_h$) we can finish the prove of this lemma. $\square$

**Lemma H.8.** *Let $\widehat{\pi} = \{\widehat{\pi}_h\}_{h=1}^{H}$ and $\widehat{Q}_h(\cdot, \cdot)$ be the arbitrary policy and Q-function and also $\widehat{V}_h(s) = \langle \widehat{Q}_h(s, \cdot), \widehat{\pi}_h(\cdot \mid s) \rangle \ \forall s \in \mathcal{S}$. and $\zeta_h(s, a) := (\mathcal{T}_h \widehat{V}_{h+1})(s, a) - \widehat{Q}_h(s, a)$ (element-wisely)*

*to be the Bellman update error. Then for any arbitrary $\pi$, we have*

$$V_1^\pi(s) - V_1^{\widehat{\pi}}(s) = \sum_{h=1}^H \mathbb{E}_\pi \left[ \zeta_h(s_h, a_h) \mid s_1 = s \right] - \sum_{h=1}^H \mathbb{E}_{\widehat{\pi}} \left[ \zeta_h(s_h, a_h) \mid s_1 = s \right]$$

$$+ \sum_{h=1}^H \mathbb{E}_\pi \left[ \langle \widehat{Q}_h(s_h, \cdot), \pi_h(\cdot|s_h) - \widehat{\pi}_h(\cdot|s_h) \rangle \mid s_1 = x \right]$$

*where the expectation are taken over $s_h, a_h$.*

*Proof.* Note the gap can be rewritten as

$$V_1^\pi(s) - V_1^{\widehat{\pi}}(s) = V_1^\pi(s) - \widehat{V}_1(s) + \widehat{V}_1(s) - V_1^{\widehat{\pi}}(s).$$

By Lemma H.7 with $\pi = \widehat{\pi}$, $\pi' = \pi$, we directly have

$$V_1^\pi(s) - \widehat{V}_1(s) = \sum_{h=1}^H \mathbb{E}_\pi \left[ \zeta_h(s_h, a_h) \mid s_1 = s \right] + \sum_{h=1}^H \mathbb{E}_\pi \left[ \langle \widehat{Q}_h(s_h, \cdot), \pi_h(\cdot|s_h) - \widehat{\pi}_h(\cdot|s_h) \rangle \mid s_1 = s \right]$$

$$(31)$$

Next apply Lemma H.7 again with $\pi = \pi' = \widehat{\pi}$, we directly have

$$\widehat{V}_1(s) - V_1^{\widehat{\pi}}(s) = - \sum_{h=1}^H \mathbb{E}_{\widehat{\pi}} \left[ \zeta_h(s_h, a_h) \mid s_1 = s \right]. \tag{32}$$

Combine the above two results we prove the stated result. $\qquad\square$

**Lemma H.9.** *For a linear MDP, for any $0 \leq V(\cdot) \leq H$, then there exists a $w_h \in \mathbb{R}^d$ s.t. $\mathcal{T}_h V = \langle \phi, w_h \rangle$ and $\|w_h\|_2 \leq 2H\sqrt{d}$ for all $h \in [H]$. Here $\mathcal{T}_h(V)(s,a) = r_h(x,a) + (P_h V)(s,a)$. Similarly, for any $\pi$, there exists $w_h^\pi \in \mathbb{R}^d$, such that $Q_h^\pi = \langle \phi, w_h^\pi \rangle$ with $\|w_h^\pi\|_2 \leq 2(H-h+1)\sqrt{d}$.*

*Proof.* By definition,

$$\mathcal{T}_h V = r_h + (P_h V) = \langle \phi, \theta_h \rangle + \langle \phi, \int_{\mathcal{S}} V(s) d\nu_h(s) \rangle$$

$$\Rightarrow w_h = \theta_h + \int_{\mathcal{S}} V(s) d\nu_h(s),$$

therefore $\|w_h\|_2 \leq \|\theta_h\|_2 + H \cdot \|\nu_h(\mathcal{S})\| \leq 1 + H\sqrt{d} \leq 2H\sqrt{d}$. The proof of the second part is similar by backward induction and the fact $V_h^\pi \leq H - h + 1$ for any $\pi$. $\qquad\square$

**Lemma H.10.** *For any pessimistic bonus design $\Gamma_h$, suppose $K > \max\{\mathcal{M}_1, \mathcal{M}_2, \mathcal{M}_3, \mathcal{M}_4\}$, then with probability $1 - \delta$, Algorithm 1 yields*

$$\left\| \mathcal{T}_h \widehat{V}_{h+1} - \widehat{\mathcal{T}}_h \widehat{V}_{h+1} \right\|_\infty \leq \widetilde{O}(\frac{H^2 \sqrt{d/\kappa}}{\sqrt{K}})$$

*Proof of Lemma H.10.* Suppose $w_h$ is the coefficient corresponding to the $\mathcal{T}_h \widehat{V}_{h+1}$ (such $w_h$ exists by Lemma H.9), *i.e.* $\mathcal{T}_h \widehat{V}_{h+1} = \phi^\top w_h$, and recall $(\widehat{\mathcal{T}}_h \widehat{V}_{h+1})(s,a) = \phi(s,a)^\top \widehat{w}_h$, then:

$$
\left( \mathcal{T}_h \widehat{V}_{h+1} \right)(s,a) - \left( \widehat{\mathcal{T}}_h \widehat{V}_{h+1} \right)(s,a) = \phi(s,a)^\top \left( w_h - \widehat{w}_h \right)
$$

$$
= \phi(s,a)^\top w_h - \phi(s,a)^\top \widehat{\Lambda}_h^{-1} \left( \sum_{\tau=1}^K \phi(s_h^\tau, a_h^\tau) \cdot \left( r_h^\tau + \widehat{V}_{h+1}\left(s_{h+1}^\tau\right) \right) / \widehat{\sigma}_h^2(s_h^\tau, a_h^\tau) \right)
$$

$$
= \underbrace{\phi(s,a)^\top w_h - \phi(s,a)^\top \widehat{\Lambda}_h^{-1} \left( \sum_{\tau=1}^K \phi(s_h^\tau, a_h^\tau) \cdot \left( \mathcal{T}_h \widehat{V}_{h+1} \right)(s_h^\tau, a_h^\tau) / \widehat{\sigma}_h^2(s_h^\tau, a_h^\tau) \right)}_{(i)}
$$

$$
+ \underbrace{\phi(s,a)^\top \widehat{\Lambda}_h^{-1} \left( \sum_{\tau=1}^K \phi(s_h^\tau, a_h^\tau) \cdot \left( r_h^\tau + \widehat{V}_{h+1}\left(s_{h+1}^\tau\right) - \left( \mathcal{T}_h \widehat{V}_{h+1} \right)(s_h^\tau, a_h^\tau) \right) / \widehat{\sigma}_h^2(s_h^\tau, a_h^\tau) \right)}_{(ii)}.
$$

$$
(33)
$$

For term (i), it is bounded by $\frac{2\lambda H^3 \sqrt{d}/\kappa}{K}$ with probability $1 - \delta$ by Lemma C.2.

For term (ii), by Cauchy inequality it is bounded by

$$
\|\phi(s,a)\|_{\widehat{\Lambda}_h^{-1}} \cdot \left\| \sum_{\tau=1}^K \phi(s_h^\tau, a_h^\tau) \cdot \left( r_h^\tau + \widehat{V}_{h+1}\left(s_{h+1}^\tau\right) - \left( \mathcal{T}_h \widehat{V}_{h+1} \right)(s_h^\tau, a_h^\tau) \right) / \widehat{\sigma}_h^2(s_h^\tau, a_h^\tau) \right\|_{\widehat{\Lambda}_h^{-1}}
$$

$$
\leq \frac{2H}{\sqrt{\kappa K}} \left\| \sum_{\tau=1}^K \phi(s_h^\tau, a_h^\tau) \cdot \left( r_h^\tau + \widehat{V}_{h+1}\left(s_{h+1}^\tau\right) - \left( \mathcal{T}_h \widehat{V}_{h+1} \right)(s_h^\tau, a_h^\tau) \right) / \widehat{\sigma}_h^2(s_h^\tau, a_h^\tau) \right\|_{\widehat{\Lambda}_h^{-1}}
$$

$$
\leq \frac{2H}{\sqrt{\kappa K}} \cdot \widetilde{O}(H\sqrt{d}) = \widetilde{O}(\frac{H^2 \sqrt{d/\kappa}}{\sqrt{K}}),
$$

where the first inequality is by Lemma H.6 (with $\phi' = \phi/\widehat{\sigma}_h$ and $\|\phi/\widehat{\sigma}_h\| \leq \|\phi\| \leq 1 := C$) and the third inequality uses $\sqrt{a^\top \cdot A \cdot a} \leq \sqrt{\|a\|_2 \|A\|_2 \|a\|_2} = \|a\|_2 \sqrt{\|A\|_2}$ with $a$ to be either $\phi$ or $w_h$. Moreover, $\lambda_{\min}(\tilde{\Lambda}_h^p) \geq \kappa / \max_{h,s,a} \widehat{\sigma}_h(s,a)^2 \geq \kappa/H^2$ implies $\left\| (\tilde{\Lambda}_h^p)^{-1} \right\| \leq H^2/\kappa$. The second inequality comes from Lemma H.2 with $R = H$ since $|\eta_\tau| = |(r_h^\tau + \widehat{V}_{h+1}\left(s_{h+1}^\tau\right) - (\mathcal{T}_h \widehat{V}_{h+1})(s_h^\tau, a_h^\tau))/\widehat{\sigma}_h(s_h^\tau, a_h^\tau)| \leq H$ and $|x_\tau| = |\phi(s_h^\tau, a_h^\tau)/\widehat{\sigma}_h(s_h^\tau, a_h^\tau)| \leq 1$.

The final result is obtained by absorbing the term (i) via the condition $K > \max\{\mathcal{M}_1, \mathcal{M}_2, \mathcal{M}_3, \mathcal{M}_4\}$. □

**Lemma H.11.** *Suppose random variables* $\|X\|_\infty \leq 2H$, $\|Y\|_\infty \leq 2H$, *then*

$$
|\mathrm{Var}(X) - \mathrm{Var}(Y)| \leq 8H \cdot \|X - Y\|_\infty.
$$

*Proof of Lemma H.11.*

$$
|\mathrm{Var}(X) - \mathrm{Var}(Y)| = |\mathbb{E}[X^2] - \mathbb{E}[Y^2] - (\mathbb{E}[X]^2 - \mathbb{E}[Y]^2)| = |\mathbb{E}[(X+Y)(X-Y)] - (\mathbb{E}[X+Y])(\mathbb{E}[X-Y])|
$$

$$
\leq \mathbb{E}[|X+Y| \cdot |X-Y|] + 4H \cdot \|X - Y\|_\infty
$$

$$
\leq 4H \mathbb{E}[|X-Y|] + 4H \cdot \|X - Y\|_\infty = 8H \cdot \|X - Y\|_\infty.
$$

□

