# OpenReview forum: "Near-optimal Offline Reinforcement Learning with Linear Representation: Leveraging Variance Information with Pessimism"
_ICLR.cc/2022/Conference — ICLR 2022 Poster_

### Official Review · Reviewer_VQKc · 2021-10-19

**Correctness:** 4
**Technical Novelty And Significance:** 3
**Empirical Novelty And Significance:** Not applicable
**Recommendation:** 8
**Confidence:** 4

**Main Review:**

The paper is overall clear and easy to follow. Incorporating variance information is a promising direction that can date back to the analysis of UCBVI, which can generally obtain better results.

I have some minor comments:

1. Perhaps it’s better to make some remarks in the algorithm on which step use which parts of data, that can be much more clear compared with current presentation.

2. There are some minor typos in the paper, e.g. in Section 3.3 “the proof can be found in Appendix D” rather than “can be bound in Appendix D”, for Theorem 3.5 the sum term should be in the denominator. Please go through the paper again to eliminate this kind of typo.

3. In the current presentation, it’s not clear where the improvement in Section 3.3 comes from. It’s acceptable due to the space constraint, but I hope there can be some intuitive interpretation on that.

4. Is the lower bound of Theorem 3.5 matches the upper bound of Theorem 3.3 up to logarithm factors and higher order term? I think so, but the authors don’t provide sufficient discussion on this. And I also want to ask if there’s a separation for tabular setting and linear setting, i.e. the worst case of linear setting is beyond the tabular setting?

I hope the authors can continuously improve on these aspects, which will further improve the readability.


**Summary Of The Paper:**

This paper considers offline policy optimization with linear function approximation that incorporates the variance information into consideration, which eventually leads to a better statistical error and fine-grained statistical analysis.

**Summary Of The Review:**

The overall idea is clear and easy to follow, but it can be much better if the authors can polish the paper again and provide more discussion.

---

> ### Author Response · Authors · 2021-11-21
> **Response for the reviewer VQKc:**
>
> We appreciate the reviewer for the expert comments and for understanding the value of this paper! Here are the detailed responses.
>
> **--- "eliminate this kind of typo" ---**
>
> Thank you for carefully reading the paper! We have fixed these typos you listed and proofread the rest of the paper to catch other typos too. We believe the paper is now a lot more readable.
>
>
> **--- "it’s not clear where the improvement in Section 3.3 comes from" ---**
>
> Thank you for the insightful question. The improvement comes from the linear representation of $\langle \phi(s,a), |\ldots|\rangle$ structure in $\Gamma^I_h(s,a)$, therefore when taking expectation $\mathbb{E}_\pi$ over
>
> $\Gamma^I_h(s,a)$, we can obtain the structure of $\mathbb{E}_\pi[\phi]$
>
>
> by exchange the order of inner product $\langle \phi ,\cdot \rangle$ and expectation $\mathbb{E}_\pi$.
>
>
> For the $\Gamma_h(s,a)$ in Theorem 3.2, one can only obtain the form
>
> $\mathbb{E}_{\pi} \sqrt{\phi^\top \Lambda^{-1}_h \phi}$ after taking expectation, which makes the bound weaker.
>
> **--- "Is the lower bound of Theorem 3.5 matches the upper bound of Theorem 3.3 up to logarithm factors and higher order term?" ---**
>
> Thanks for the expert question! Actually, we admit our lower bound does not match the higher order term. The reason is due to the hard problem instances construction in E.1 have stochastic transitions, which makes our lower bound can only match the dominate term in the upper bound (up to the logarithmic factors) and could not have the additional $1/K$ convergence for the deterministic transitions. Including both terms in the lower bound is an open question and to be honest we do not know the answer yet.
>
>
> **--- "I also want to ask if there’s a separation for tabular setting and linear setting" ---**
>
> Thank you for the very insightful question! When reduced the tabular case, there is a separation between tabular and linear case (roughly) by a factor of $\sqrt{d}$ (e.g. by comparing (4) with [Yin and Wang (2021b)]). We believe this separation is caused by the definition of Linear MDP (Definition 2.1) since $||\nu(\mathcal{S})||_2\leq \sqrt{d}$ and $\int_\mathcal{S}P_h(s'|s,a)ds'=\langle \phi(s,a),\nu_h(\mathcal{S})\rangle\leq ||\phi(s,a)||_2\cdot||\nu_h(\mathcal{S})||_2$ and could have order $\sqrt{d}$ without further constraint (instead of the maximum value $1$). We feel this phenomenon is analogous in the bandit setting, i.e. $\Omega(\sqrt{\frac{A}{K}})$ for best-arm idenfication in MAB problems and $\Omega(\sqrt{\frac{d^2}{K}})$ for stochastic linear bandits (e.g. [1]).
>
> [1] Bandit Algorithms, Tor Lattimore and Csaba Szepesvari.
>
> We thank the reviewer again for your expert comments.

---

### Official Review · Reviewer_of24 · 2021-10-29

**Correctness:** 3
**Technical Novelty And Significance:** 3
**Empirical Novelty And Significance:** Not applicable
**Recommendation:** 6
**Confidence:** 2

**Main Review:**

Strength:
* The paper is well organized, theoretically grounded
* The results seem of reasonable significance.
* The related work in this area is well explained.


Weakness:

*There is a lack of empirical evaluation carried out, to showcase how the lower bound estimation
* It's hard to follow for those who are not familiar with this sub-field.


* Minor typo in Introduction(section 1): "stat" -> "state"

**Summary Of The Paper:**

The given work proposes the introduction of a pessimistic lower bound for reinforcement learning agents.The estimated variances are used to re-weight the Bellman residual learning objective, such that the samples with higher uncertainty are correspondingly penalized.



**Summary Of The Review:**

Overall, the paper is well motivated, aiming to solve an important problem in the area of offline reinforcement learning. The work is theoretically well grounded, studying offline reinforcement learning for time-inhomogeneous episodic linear Markov decision processes. While the paper is reasonably well organized, it's quite hard to grasp the fundamental concepts especially for those not an expert in this sub-field.

---

> ### Author Response · Authors · 2021-11-21
> **Response for the reviewer of24:**
>
> We appreciate the reviewer for the encouraging comments of our paper! We have fix the typo you mentioned.
>
> **--- "There is a lack of empirical evaluation" ---**
>
> We provide a simulation study these days and it is included in Appendix F of our revision. We implement PEVI and VAPVI and shows VAPVI outperform PEVI. We do not compare with [Xie et al.] or [Zanette et al.] since their algorithms are either information-theoretical or rely on a computation oracle (which make their algorithms not really implementable). Thank you again for your positive view.

---

### Official Review · Reviewer_UKxA · 2021-10-30

**Correctness:** 4
**Technical Novelty And Significance:** 3
**Empirical Novelty And Significance:** 3
**Recommendation:** 6
**Confidence:** 4

**Main Review:**

Strength:
The paper is clearly written. Rigorous theory is backed up by neat intuitions. The presentation is very fluent and readers are able to capture the high-level intuitions smoothly.
This work contributes to the study of value-basic pessimistic RL by closing the gap in suboptimality bounds. It also related RL to OPE by showing the variance-aware technique in OPE also applies to policy learning. I believe these are important insights towards a complete picture of pessimism.

Weakness / questions:
The current setting of the paper is well studied and complete, but it might be a little restricted. For example, a nice point of the PEVI paper (Jin et al. 2021b) is that the suboptimality directly depends on the dataset, not restricting to behavior policy. I am interested whether such variance-incorporated ideas can be used for fully data-dependent constructions. This might also be interesting for a better understanding of what a static dataset can provide.
Invertibility of covariance matrix: the feasibility algorithm of PEVI is actually not influenced by singular covariance matrix, as long as that is in line to the actual optimal policy, or the policy in comparison (i.e., both behavior and optimal policy miss some subspace). I am curious what happens when the covariance matrix is not invertible, so that Lambda_h^* does not exist. It might depend on the relationship between \mu to optimal policy? Not expecting an exact result, but some discussions might be interesting.


**Summary Of The Paper:**

This paper studies an extension of value-based pessimistic offline RL in linear settings, where variance information is included in the pessimistic penalty to provide a refined quantification of uncertainty. Although a little restricted to the behavior policy setting, careful treatment of uncertainty leads to improved suboptimality bound than the literature, nearly closes the gap in statistical limits of such approach, bringing insights into pessimistic offline RL.

**Summary Of The Review:**

In general, this paper is clearly written and elegantly presented. The combination of rigorous theory and high-level intuitions make it fluent to follow. This work provides a refined treatment of pessimistic penalty functions, which leads to tighter upper bounds and nearly matching lower bounds. Although complete for the current setting, I think the paper can be improved by adding some broader considerations to more general cases to bring more insights beyond the restrictions of fixed behavior policy and well-behaved covariance matrix.

---

> ### Author Response · Authors · 2021-11-21
> **Response for the reviewer UKxA:**
>
> We appreciate the reviewer for the positive comments and for understanding the value of this paper. Here are the detailed responses.
>
> **--- "PEVI paper (Jin et al. 2021b) is that the suboptimality directly depends on the dataset, not restricting to behavior policy. I am interested whether such variance-incorporated ideas can be used for fully data-dependent constructions." ---**
>
> Thanks for the sharp observation! yes, you are right. Indeed, our variance-aware result could also work directly on the dataset, without specifying the behavior policy. The reason is: in Theorem 3.2,
> $$\Lambda_{h}=\sum_{k=1}^{K} {\phi\left(s_{h}^{k}, a_{h}^{k}\right) \cdot \phi\left(s_{h}^{k}, a_{h}^{k}\right)^{\top}}/\sigma^2_{\hat{V}} +\lambda I_{d}$$
> is calculated by data only, and our bound is also a fully data-dependent bound. Therefore, it works for general static data and we just need to replace $\Sigma^p_h:=E_{\mu,h}[\phi(s,a)\phi(s,a)^\top]$ with $E_{\mathcal{D},h}[\phi(s,a)\phi(s,a)^\top]$ in Assumption 2.2.
>
>
> **--- "Invertibility of covariance matrix: the feasibility algorithm of PEVI is actually not influenced by singular covariance matrix, as long as that is in line to the actual optimal policy. I am curious what happens when the covariance matrix is not invertible?" ---**
>
> Thank you for the insightful comment! Yes, in general it is possible to assume condition $sup_{x \in {R}^{d}} \frac{x \Sigma_{\pi^{\star} }x^{\top}}{x \Sigma_{\mu} x^{\top}}<\infty$ if one only wants to learn the optimal policy (see the discussion after Assumption 2.2 in page 4). However, our theorem statements (Theorem 3.2, Theorem 3.3) wish to complete with any policy $\pi$ simultaneously, hence we make the assumption on $\Sigma_{\mu}$ instead. When $\Sigma_{\mu}$ is not fully covered, we can still learn in the subspan of $\Sigma_{\mu}$ that are explored. Indeed, we add the discussion section (Appendix G.1) in the revision to discuss this. Roughly speaking, we can learn the problem with the truncated features $\phi'_h|{1:d'}$ and avoid the singularity with the $d'$-dimensional subspace. If some optimal policy $\pi^\star$ falls into this subspace (i.e. only explore this $d'$-dimensional subspace), then the result is still near-optimal. If no optimal policy is coverd by this $d'$-dimensional subspace, then we need to suffer a constant suboptimality due to the agnostic state-action region.

---

> > ### Author Response · Authors · 2021-11-27
> > **Further response to reviewer UKxA:**
> >
> > Dear reviewer,
> >
> > To further respond to the broader considerations you mentioned in the initial review, we wish to mention that, as a by-product, the variance-aware framework can also be applied for linear mixture MDPs. We promise we will add the discussion of this part in the next revision (since at this moment we cannot update the draft).
> >
> > The authors

---

### Official Review · Reviewer_exon · 2021-11-02

**Correctness:** 2
**Technical Novelty And Significance:** 2
**Empirical Novelty And Significance:** 2
**Recommendation:** 5
**Confidence:** 3

**Main Review:**

Overall comments :

- This paper focuses on the problem of understanding statistical limit of offline RL algorithms, under linear representations. The authots propose a variance aware approach for offline RL, which leads to the pessimistic value estimation. The idea of pessimism has been extensively studied for offline RL recently.
- The core idea of the algorithm is to use the variance term, to re-weight the Bellman updates - and provide improved bounds for offline RL). The paper exploits the reweighted Bellman update, which leads to the improved bounds.
- Recent works has studied the statistical limits for offline RL (Wang et al). This paper can be seen as a follow-up along those lines, studying the limits for offline RL under linear MDP assumptions. As pointed out, even though several works already address this (e.g Jin et al 2021), they all use the standard Bellman update leading to the bound for offline RL. In contrast, this paper uses the pessimistic value, based on the variance, and argues that the re-weighting is helpful for improving bounds. The core idea is to modify the LSVI update with a variance weighting term and an additional pessimism bonus.
- The paper is well written, and section 3 is quite easy to follow. Most importantly, it is great to see example 3.1 where the authors clearly state how existing results that equally weighs the samples has dependency on the H factor that this work is trying to avoid. More importantly, the comparisons with previous works are useful to get an understanding of the contributions of this work.



Few weaknesses of the work  and negative comments :
- I am not an expert in following the proof details of this work; therefore the comments below are not on the technical details of the proof.
- My overall impression of a work like this is that - the variance penalty term and the re-weighting of the Bellman update does not necessarily give better insights about offline RL, other than the fact that it helps improve the bound. This is similar to any other empirical works, which can re-weight the Bellman update and claiming to have improved results.
- What is the key novelty in the paper that leads to the improved bound? Is there a specific technical novelty introduced in the proof that leads to the improved result? Can the authors point this out exactly?
- Otherwise, simply re-weighting the update and following similar analysis as in the vast majority of works studying offline RL theory is not a significant contribution and does not lead to new insights. The Bellman re-werighting idea has been well studied and exploited in many previous empirical works in RL.
- How does the variance term help for offline RL? To me, it seems like the idea is a nice trick to include the variance term in the update. I do not see any bottlenecks to try this out empirically in any existing offline RL algorithm on any benchmark task like D4RL? I believe this update will be quite easy to plug in, and it would be nice to examine the significance of the variance and re-weighting term.



**Summary Of The Paper:**

This paper proposes a re-weighting of the Bellman update for offline RL, using the variance term, that leads to an improved bound for offline RL. The paper is well written and easy to follow, with the key improvement (theoretically) easy to see. The core idea of the algorithm is a simple trick to re-weight the update and the analysis follows as in other offline RL works examining the statistical significance for offline value estimation.




**Summary Of The Review:**

I do not think without experiments, the work has enough technical and theoretical novelty (in terms of proof details) for acceptance. Simply improving the bound, by re-weighting the update and following similar analysis as other offline RL works do not, in my opinion, pass the criterion. I may be wrong, as I am not an expert in following the technical proofs of the paper.

---

> ### Author Response · Authors · 2021-11-21
> **Response for the reviewer exon:**
>
> We thank the reviewer for providing detailed feedback. We have read your comments carefully and below are our detailed responses.
>
>
> **--- "the variance penalty term and the re-weighting of the Bellman update does not necessarily give better insights about offline RL, other than the fact that it helps improve the bound" ---**
>
> Indeed, re-weighting the least square objective can improve the bound reveals the key insight of RL: Reinforcement learning is a **heteroscedastic** problem. This is due to: for different $(s,a)$, the transition $P(\cdot|s,a)$ could incur different variances (especially for continuous problems where there are infinite many transitions to learn). Please also see **Motivation** paragraph in page 5 for more discussions. Previous papers ([Jin et al., Xie et al., Zanatte et al.]) put equal weights for the training samples, which "implicitly" cast RL as a **homoscedastic** problem and this is not precise. As a result, their theoretical guarantee is suboptimal to ours.
>
> **--- "What is the key novelty in the paper that leads to the improved bound?" ---**
>
> Thanks for the excellent question! The key novelty that provides near-optimality is our variance-weighted design can fit appropriately with the newly developed tool *Bernstein inequality for self-normalized martingale* by [Zhou et al. 2021]. Especially, previous offline RL methods (e.g. PEVI in [Jin et al. 2021b]) can apply this technical tool as well, but they will end up with no improvement compared to simply applying the Hoeffding style tool. The reason behind the scenes is that they do not incorporate variance information in their algorithm design, as opposed to us. Combing variance-aware design with the new tool, we can further improve over existing results and they are near-optimal.
>
> **--- "I do not think without experiments..." ---**
>
> Thanks. First of all, this is a theoretical-orientated paper. We wish to understand the statistical limit of offline reinforcement learning and provide improved theoretical guarantees over existing theoretical papers.
>
> Second, evaluating VAPVI empirically in a task like D4RL seems not very ideal. The main reason is that continuous control problems in D4RL are usually complex and might not admit linear representations. Hence, one might not expect VAPVI to work well in those tasks when the linear MDP structure is violated.
>
> Lastly, to show our method indeed outperforms previous methods when the linear MDP structure is satisfied, we spend these days running a simulation study by comparing our VAPVI with PEVI. We do not compare with [Xie et al.] or [Zanette et al.] since their algorithms are either information-theoretical or rely on a computation oracle (which make their algorithms not really implementable). In the synthetic linear MDP, our result outperforms PEVI and the details can be found in the updated draft (Appendix F).
>
>
>
> We sincerely appreciate it if the reviewers could kindly consider improving the scores if we successfully addressed the concerns. If not, we are very happy to answer any further questions you have.

---

> > ### Author Response · Authors · 2021-11-27
> > **Further response to reviewer exon**
> >
> > Dear reviewer,
> >
> > May I ask if we successfully address your concern? If there are some issues with it, please feel free to let us know about it.
> >
> > Thank you,
> > Authors

---

### Decision · Program_Chairs · 2022-01-20

**Decision:**

Accept (Poster)

**Comment:**

In this paper, the authors motivate the paper well by the gap between the upper bound of the popular offline RL algorithm and the lower bound of the offline RL. By exploiting the special linear structure, the authors designed a variance-aware pessimistic value iteration, in which the variance estimation is used for reweighting the Bellman loss. Finally, the upper bound of the proposed algorithm in terms of the algorithm quantity is proposed, which is more refined to reflexing the problem-dependency. These results are interesting to the offline RL theoretical community.

As the reviewers suggested, several improvements can be made to further refinement, e.g.,

- The intuition about the self-normalization in the algorithm exploited to improve the upper bound should be introduced.
- The discussion in Sec 3.3t about the insight of the improvement of the upper bound is not sufficient.
- The extra computational cost about the variance should be discussed.